# Measuring Multimodal Mathematical Reasoning with the MATH-Vision Dataset

**Ke Wang**[1,*]    **Junting Pan**[1,4,*,†]    **Weikang Shi**[1,*]    **Zimu Lu**[1]

**Houxing Ren**[2]    **Aojun Zhou**[1]    **Mingjie Zhan**[3,*,‡]    **Hongsheng Li**[1,4,‡]

[1]CUHK MMLab, [2]Shanghai AI Lab, [3]SenseTime Research, [4]CPII under InnoHK
*Joint first author, †Project lead, ‡Directional lead

## Abstract

Recent advancements in Large Multimodal Models (LMMs) have shown promising results in mathematical reasoning within visual contexts, with models exceeding human-level performance on existing benchmarks such as MathVista. However, we observe significant limitations in the diversity of questions and breadth of subjects covered by these benchmarks. To address this issue, we present the `MATH-Vision` (`MATH-V`) dataset, a meticulously curated collection of 3,040 high-quality mathematical problems with visual contexts sourced from real math competitions. Spanning 16 distinct mathematical disciplines and graded across 5 levels of difficulty, our dataset provides a comprehensive and diverse set of challenges for evaluating LMMs' mathematical reasoning abilities. Through extensive experimentation, we unveil a notable performance gap between current LMMs and human performance on `MATH-V`, underscoring the imperative for further advancements in LMMs. Moreover, our detailed categorization allows for a thorough error analysis of LMMs, offering valuable insights to guide future research and development. The dataset is released at MathLLMs/MathVision

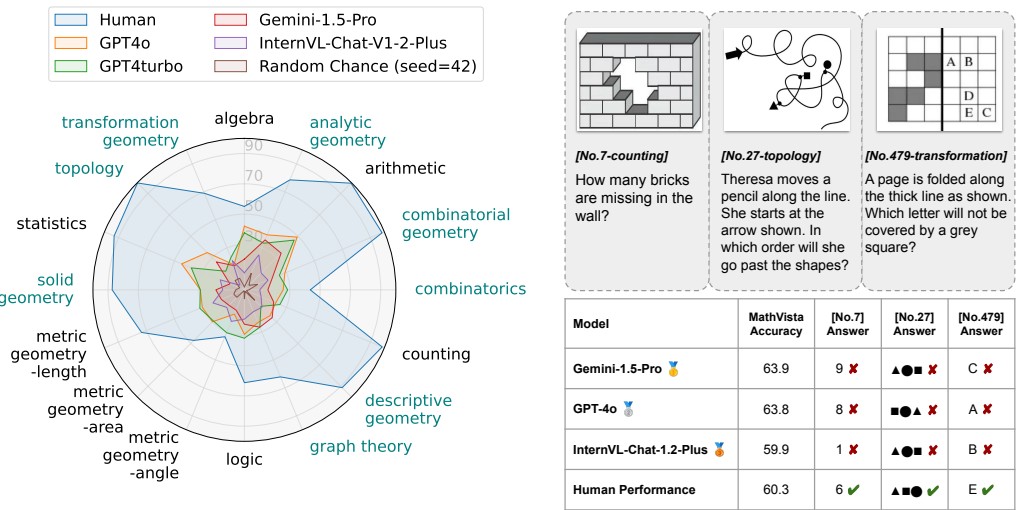

(a) Zero-shot Accuracy by Subjects          (b) "Easy" problems failed by LMMs

Figure 1: (a) Zero-shot accuracies of four prominent Large Multimodal Models (LMMs), random chance, and human performance are evaluated on our proposed `MATH-V` across 16 subjects. Teal means newly introduced subjects. (b) Examples of easy problems in `MATH-V` failed by top-performing LMMs on MathVista. The three questions come from tests designed for elementary school students.

38th Conference on Neural Information Processing Systems (NeurIPS 2024) Track on Datasets and Benchmarks.

# 1 Introduction

The latest advancements in Large Language Models (LLMs) [1, 2, 3, 4, 5] and Large Multimodal Models (LMMs) [6, 7, 8, 9, 10, 11] have demonstrated remarkable capabilities, achieving impressive performance on a diverse array of language and visual-language tasks [12, 13, 14, 15, 16, 17, 18, 19, 20]. As a result of these strides, recent LMMs like GPT-4 [1, 6], InternLM-XComposer [21], InternVL [22] and Gemini [7, 23] are now outperforming average human capabilities in multimodal mathematical reasoning. Notably, on MathVista [24], the most widely used benchmark for evaluating LMMs' mathematical reasoning in visual contexts, the current leading model achieves a score of 63.9, surpassing the human average of 60.3. While the impressive performance on MathVista suggests that LMMs have surpassed average human capabilities in multimodal mathematical reasoning, our recent experiment on human math tests reveals a different conclusion. As illustrated in Figure 1.b, these problems, which are straightforward for humans, remain challenging for LMMs.

In light of this discrepancy and to gain a deeper understanding of the limitations of current math-oriented multimodal datasets, we conducted a thorough examination of existing benchmarks. Our analysis revealed two key issues:

Firstly, we observed that the range of subjects of existing benchmarks remains limited. For example, geometry problems are one of the most common types of math problems with visual contexts, as evidenced by the large body of geometry problems and benchmarks (*i.e.,* Geometry3k [25], GeoQA+[26], GEOS [27], and UniGeo [28]). However, despite the abundance of problems, the diversity is quite limited. Surprisingly, we found that they predominantly concentrate on plane geometry, neglecting the rich diversity of other geometry subjects such as solid geometry, which deals with the volume and surface area of 3D objects. Additionally, we found that topics such as Graph Theory, Topology and Transformation Geometry are also rarely present in current benchmarks.

Secondly, while the visual data in existing datasets are quite diverse and are collected from different sources, the questions are quite limited in question types and contents. For example, we identified that among the 375 questions on "abstract scene" from MathVista, the majority can be categorized into just three types of questions: (1) Object Counting, (2) Length Measurement with a given reference, and (3) Time Inquiry based on a clock. Questions are mostly recognition-driven and seldom require in-depth mathematical reasoning. Similar problems also exist in image contexts of "function plots", "synthetic scenes", "natural images", etc. This tendency towards monotonous question patterns is commonly observed, where annotators are first given a contextual image and then asked to create math-related questions according to the image.

The limited subject range and question variety of current multimodal mathematical reasoning benchmarks can create a false impression of LMMs' capabilities, potentially misguiding the development of these models. Therefore, aiming to more comprehensively evaluate the mathematical reasoning skills of LMMs, we introduce the `MATH-Vision` (`MATH-V`) dataset. This dataset comprises 3,040 high-quality mathematical problems within visual contexts across 12 grades, carefully selected from 19 real-world math competitions. To ensure a high-quality standard, **all data within our dataset were cross-validated and filtered by multiple expert annotators**, guaranteeing that each problem has a unique and correct answer. The dataset is well-balanced, featuring 1,532 problems in an open-ended format and 1,508 in a multiple-choice format. Specifically, we introduce three major updates in our `MATH-V`:

1. **Expanded Categories**: In `MATH-V`, we introduce 8 new math categories, such as descriptive geometry (dimension understanding), graph theory (relations modeling), and topology (invariant under continuous deformation) and subdivide plane geometry into three categories (angle, area and length). These categories are important areas of multimodal mathematical reasoning and are currently not covered in existing benchmarks.

2. **Holistic Data Approach**: All data in `MATH-V`, including both questions and images, are jointly collected from real-world math competitions. This approach eliminates the need for additional annotation and avoids the monotonous question patterns. This difference in question annotation is reflected in the average question length, which is 42.3 for `MATH-V`, significantly surpassing the average of 15.6 of MathVista. Longer question lengths offer numerous advantages, particularly in facilitating richer mathematical reasoning by providing additional context. Further qualitative comparisons can be found in Appendix G.1.

3. **Difficulty Levels**: To better understand model performance, we categorize all problems in `MATH-V` by difficulty levels ranging from 1 to 5. This classification is rigorously verified by a team of experts to ensure accuracy and reliability. We hope that this categorization can facilitate better analysis of mathematical problem-solving skills across various levels of difficulty.

We conducted extensive experiments with our dataset to provide insights into the mathematical reasoning abilities of current LMMs on human math tests. As depicted in Figure 1.a, our findings reveal a substantial gap in mathematical reasoning within visual contexts between LMMs and humans, with scores of 30.39 and 68.82, respectively. Our `MATH-V` benchmark poses a significant challenge for LMMs but is relatively easy for humans to achieve a good score, as shown in Figure 1.b. This indicates that current top-tier LMMs still have many areas (*e.g.,* transformation geometry, topology, etc.) with significant room for improvement in terms of mathematical reasoning in visual contexts to approach the performance of average humans.

Given that `MATH-V` primarily consists of unseen data for existing LMMs , it generally exhibits lower scores compared to other benchmarks with available training data that could have been used during LMMs' development [24, 29, 30]. We believe that solving these tasks is fundamental for applications in architecture, design, and engineering (*e.g.,* solid geometry for architecture). Therefore, our benchmark provides a crucial foundation for future research. Furthermore, `MATH-V` could perfectly complement current benchmarks, together offering a more robust evaluation of the mathematical reasoning capabilities of LMMs.

In conclusion, the contributions of this study are listed as follows:

- We carefully audited existing datasets and identified significant problems that are relevant but not known to the community, such as missing math subjects and a significant number of questions with repetitive patterns.

- We present a new `MATH-V` benchmark for pushing advances in multimodal mathematical reasoning. `MATH-V` comprises 3,040 problems with visual context, selected from real-world math competitions. All problems are classified into 5 difficulty levels and 16 mathematical disciplines.

- Finally, leveraging the fine-grained categorization, we conducted a comprehensive analysis of current LMMs, suggesting new directions for improvement in future research.

## 2 Related Works

### 2.1 Mathematical Reasoning Benchmarks

Various benchmark datasets [31, 32, 33, 34, 35] have been proposed to assess models' mathematical reasoning capabilities. However, current multimodal benchmarks [29, 36, 37, 38] primarily focus on visual recognition tasks, and multimodal mathematical reasoning benchmarks are less common. Early multimodal reasoning benchmarks, such as Geometry3k [25], GeoQA+[26], GEOS [27], and UniGeo [28], are very limited in subjects, primarily focused on plane geometry. Among recent multimodal benchmarks, MMMU [39] stands out, though it is limited to multiple-choice questions and only a small fraction of its content is mathematical. Another recent benchmark, MathVista [24], evaluates mathematical reasoning abilities within diverse visual contexts. However, it lacks a detailed classification of mathematical subdomains and emphasizes visual abilities rather than mathematical reasoning. In contrast, `MATH-V` contains math problems rigorous in both mathematical reasoning and visual recognition, categorized by clear difficulty levels and covering 16 representative subjects.

### 2.2 LMMs for Math

Recently, AlphaGeometry [40] has shown impressive performance in solving challenging geometry problems, though it lacks the ability to process images and must rely on text descriptions of images. Current math-specific multimodal models, such as G-LLaVA [41], UniMath [42], MatCha [43], and UniChart [44], are primarily focused on plane geometry or chart-based problems and still lag behind general multimodal models such as GPT-4-turbo [1]. To facilitate better evaluation of the mathematical reasoning abilities of multimodal models, we introduce our benchmark, featuring high-quality problems demanding expertise in both mathematical reasoning and visual recognition.

### 2.3 Multimodal Foundation Models

With the success of LLMs, MiniGPT-4 [45], LLaMA-Adapter [46], and LLaVA [47, 10] have attempted to construct multimodal models from text-only models. Subsequently, various large multimodal models such as GPT-4V [6], Gemini [7, 23], Qwen-VL [11], InternLM-XComposer-VL [21, 8], and SPHINX [9] have been created, demonstrating impressive performance on benchmarks like MMMU [39] and MathVista [24]. Notably, some open-sourced models have even received scores higher than GPT-4-turbo on these benchmarks. However, on our benchmark, GPT-4-turbo performs considerably better than open-source models, indicating the lack generalization abilities of some models, especially on questions very different from those in fine-tuning datasets.

## 3 Dataset

### 3.1 Overview

We introduce `MATH-Vision` (`MATH-V`), a carefully curated benchmark designed to evaluate the multi-modal mathematical reasoning capabilities of foundation models across a wide range of mathematical tasks with visual contexts. We particularly emphasize the challenges posed by the need for both expert-level visual perception and deliberate reasoning with subject-specific knowledge. This challenge is exemplified in our tasks, which require the processing of various heterogeneous image types and necessitate a model's proficiency in utilizing domain-specific knowledge to deeply comprehend both text and images, and to reason accordingly. This extends far beyond basic visual perception and calls for an advanced approach that combines sophisticated multimodal analysis with mathematical reasoning. The proposed benchmark encompasses 16 subjects over 5 levels of difficulty, including 8 new subjects including *Analytic Geometry, Combinatorial Geometry, Combinatorics, Descriptive Geometry, Graph Theory, Solid Geometry, Topology, Transformation Geometry* and subdividing plane geometry into three categories: *angle, area, length*. The questions in our benchmark were manually collected from 19 competitions. The division of difficulty levels is primarily based on the recommended grades of these competitions, with details provided in Appendix H.2. `MATH-V` consists of 3,040 questions, and we also provide a smaller test set of 304 questions for quick evaluation. Detailed coverage, statistics and sources of `MATH-V` are presented in Appendix B.

### 3.2 Data Collection

**Data collection** Our benchmark collection comprises two stages. In the first stage, we gather a list of various mathematical competitions, both international and national. The selection is based on the criterion that visual inputs should be frequently utilized in the competition questions to provide significant insights, and the questions should be categorizable into different levels, such as AMC8, AMC10, AMC12, and AIME. Competitions like the International Mathematical Olympiad (IMO) are excluded due to the excessive difficulty of the questions and the rare use of visual inputs. Consequently, we select 19 competitions from various countries. In the second stage, we filter through online resources to identify problems where questions are interspersed with images. We adhere strictly to copyright and licensing regulations, ensuring that we avoid data from sites that prohibit copying and redistribution. Given the concerns surrounding data contamination in LMMs, we prioritize selecting questions that lack readily available answers, such as those found in separate documents or pages. For questions formatted as PDFs, we use the Mathpix API to extract text and images. This approach allows us to curate a diverse collection of 9k questions from various sources.

**Data curation** To improve the quality of our data, we undergo a four-stage data curation process with the help of 10 annotators (senior college students from science-related majors). In the first stage, we verify the alignment of text questions and images, as Mathpix might return them in an incorrect order. We also eliminate questions with missing text or incorrect images and those with private information or offensive content. All results are cross-verified between different annotators. 3,352 math-VQA data are left after this stage. In the second stage, we use lexical overlap and Levenshtein distance to identify potential duplicate questions. These suspected duplicates are then reviewed to identify and remove any duplication. As a result of these steps, we obtain the final 3,040 math-VQA data for our benchmark. In the third stage, we categorize the problems into 16 different subjects. Our annotators are divided into three groups, each independently labeling all the questions, and we also utilized GPT-4V [6] and Gemini Pro [7] for subject advice. Annotations from various groups and

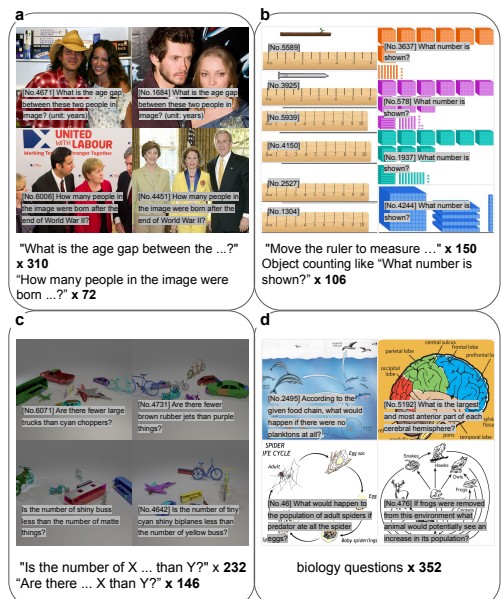

Figure 2: Questions in (a), (b), and (c) all emphasize visual recognition over mathematical reasoning. (a) requires model to identify the persons, which involves niche domain knowledge that is not related to mathematical reasoning. (d) highlights non-mathematical problems in MathVista.

| | MathVista | MATH-V |
|---|---|---|
| Main source | existing VQA & MathQA datasets | all from real exams |
| Template questions | Yes | No |
| Math reasoning types | 7 | 16 |
| Newly annotated questions | 736 | 3040 |
| Unique questions | 4746 | 3040 |
| Average question length | 15.6 words | 42.3 words |

Table 1: Statistics of key differences between MathVista and our MATH-V. MATH-V comprises high-quality math questions from real examinations or competitions. In contrast, MathVista primarily features images from existing VQA datasets with template questions.

models are consolidated through a majority voting system involving three expert annotators. In the final stage, we assign difficulty levels for each questions with reference to the grade requirements of the original competitions.

### 3.3 Comparison with Existing Benchmarks

**Comparing with MathVista** MathVista [24] is a comprehensive multimodal mathematical reasoning benchmark derived from 28 existing math-related multimodal datasets and 3 newly collected ones. However, within MathVista, approximately 20 datasets exhibit a trend where questions are annotated post-image collection by annotators, resulting in a relatively limited variability of questions, as shown in Figure 2. In contrast to MathVista, all math-VQA data including both questions and images in MATH-V are newly collected from real-world math competitions. This difference in obtaining question annotations is reflected in the average question length, which stands at 42.3 for MATH-V, significantly surpassing the 15.6 average of MathVista. A longer question length offers numerous advantages, particularly in facilitating richer mathematical reasoning by providing additional context. Further qualitative comparisons of math-VQA data between our dataset and MathVista are provided in Appendix G.1. Moreover, for MATH-V we also provide 16 fine-grained categorizations of different math subjects, which can gain deeper insights into individuals' strengths and weaknesses in specific areas of mathematics, compared to MathVista's 7 math reasoning types. The detailed data of the comparison can be found in Table 1. Finally, all problems within our datasets are labeled with one of the five levels of difficulty, while MathVista has only three levels, with a considerable portion of problems (37.7%) classified under the *not applicable* category.

**Comparing with MMMU** Different from MathVista and our MATH-V, MMMU [39] is designed to evaluate the multi-discipline multimodal understanding and reasoning abilities of LMMs with college-level problems. In terms of Math, it spans eight distinctive subjects with a total of 500 questions, whereas subjects like Operation Research, Graph Theory, and Group Theory require sophisticated domain knowledge such as "DFS search" and "Cayley diagram". In contrast, our MATH-V focus on mathematics problems from elementary school to high school and can test the general visual perception and mathematical reasoning abilities of LMMs. Our dataset is 6 times larger than MMMU-Math, providing a significantly larger and more diverse set of challenges for analysis and evaluation.

Table 2: Comparison of model performances across various mathematical subjects. Subjects: Alg: algebra, AnaG: analytic geometry, Ari: arithmetic, CombG: combinatorial geometry, Comb: combinatorics, Cnt: counting, DescG: descriptive geometry, GrphT: graph theory, Log: logic, Angle: metric geometry - angle, Area: metric geometry - area, Len: metric geometry-length, SolG: solid geometry, Stat: statistics, Topo: topology, TransG: transformation geometry. The first and second highest accuracy of LMMs are marked in red and blue, respectively.

| Model | Overall | Alg | AnaG | Ari | CombG | Comb | Cnt | DescG | GrphT | Log | Angle | Area | Len | SolG | Stat | Topo | TransG |
|---|---|---|---|---|---|---|---|---|---|---|---|---|---|---|---|---|---|
| Random Chance (seed=42) | 5.86 | 2.0 | 11.9 | 4.3 | 9.4 | 3.0 | 1.5 | 9.6 | 1.1 | 5.9 | 0.6 | 7.0 | 7.6 | 5.7 | 3.5 | 8.7 | 8.3 |
| LLMs (Text Only) | | | | | | | | | | | | | | | | | |
| GPT-4 (CoT) | 8.16 | 12.8 | 10.7 | 15.7 | 4.9 | 10.7 | 10.5 | 1.9 | 5.6 | 8.4 | 8.1 | 6.2 | 8.7 | 8.6 | 3.5 | 4.4 | 4.8 |
| LLMs (Text + Image Caption) | | | | | | | | | | | | | | | | | |
| ChatGPT-3.5 (CoT) | 9.74 | 10.7 | 20.0 | 18.6 | 10.1 | 7.7 | 17.9 | 16.4 | 10.0 | 13.5 | 6.4 | 5.8 | 6.5 | 9.4 | 12.1 | 4.4 | 10.7 |
| GPT-4 (CoT) | 13.10 | 16.5 | 20.2 | 34.3 | 10.4 | 17.9 | 19.4 | 7.7 | 11.1 | 10.1 | 9.8 | 9.6 | 9.1 | 13.5 | 13.8 | 8.7 | 12.5 |
| Open-source LMMs (Text + Image) | | | | | | | | | | | | | | | | | |
| LLaVA-v1.5-7B | 8.52 | 7.0 | 7.1 | 10.7 | 7.1 | 4.8 | 10.5 | 7.7 | 10.0 | 9.2 | 15.6 | 10.2 | 9.8 | 5.3 | 8.6 | 4.4 | 4.8 |
| SPHINX (V2) | 9.70 | 6.7 | 7.1 | 12.9 | 7.5 | 7.7 | 6.0 | 9.6 | 16.7 | 10.1 | 11.0 | 11.8 | 12.5 | 8.2 | 8.6 | 8.7 | 6.0 |
| ShareGPT4V-7B | 10.53 | 5.5 | 3.6 | 12.9 | 10.1 | 4.8 | 7.5 | 11.5 | 14.4 | 10.9 | 16.2 | 11.8 | 12.3 | 9.8 | 15.5 | 17.4 | 11.3 |
| LLaVA-v1.5-13B | 11.12 | 7.0 | 14.3 | 14.3 | 9.1 | 6.6 | 6.0 | 13.5 | 5.6 | 13.5 | 10.4 | 12.6 | 14.7 | 11.5 | 13.8 | 13.0 | 10.7 |
| ShareGPT4V-13B | 11.88 | 7.5 | 15.5 | 16.4 | 10.7 | 8.9 | 9.0 | 11.5 | 8.9 | 7.6 | 11.6 | 13.0 | 17.4 | 10.3 | 8.6 | 8.7 | 12.5 |
| SPHINX-MoE | 14.18 | 7.8 | 17.9 | 14.3 | 15.6 | 9.5 | 11.9 | 12.5 | 15.6 | 12.6 | 16.2 | 15.6 | 17.8 | 13.5 | 12.1 | 8.7 | 16.1 |
| InternLM-XComposer2-VL | 14.54 | 9.3 | 15.5 | 12.1 | 15.3 | 11.3 | 10.5 | 14.4 | 22.2 | 19.3 | 19.7 | 15.6 | 15.0 | 11.9 | 15.5 | 26.1 | 15.5 |
| InternVL-Chat-V1-2-Plus | 16.97 | 11.3 | 25.0 | 15.7 | 16.9 | 10.1 | 11.9 | 16.4 | 15.6 | 19.3 | 22.5 | 16.4 | 22.5 | 14.3 | 17.2 | 4.4 | 20.8 |
| Closed-source LMMs (Text + Image) | | | | | | | | | | | | | | | | | |
| Qwen-VL-Plus | 10.72 | 11.3 | 17.9 | 14.3 | 12.7 | 4.8 | 10.5 | 15.4 | 8.9 | 14.3 | 11.6 | 6.4 | 10.0 | 14.3 | 6.9 | 8.7 | 11.31 |
| Qwen-VL-Max | 15.59 | 10.7 | 19.1 | 20.0 | 16.9 | 12.5 | 17.9 | 16.4 | 12.2 | 21.0 | 19.8 | 11.5 | 19.8 | 11.5 | 20.7 | 13.0 | 17.3 |
| Gemini Pro | 17.66 | 15.1 | 10.7 | 20.7 | 20.1 | 11.9 | 7.5 | 20.2 | 21.1 | 16.8 | 19.1 | 19.0 | 20.0 | 14.3 | 13.8 | 17.4 | 20.8 |
| Gemini-1.5 Pro | 19.24 | 20.3 | 35.7 | 34.3 | 19.8 | 15.5 | 20.9 | 26.0 | 26.7 | 22.7 | 14.5 | 14.4 | 16.5 | 18.9 | 10.3 | 26.1 | 17.3 |
| GPT-4V | 22.76 | 27.3 | 32.1 | 35.7 | 21.1 | 16.7 | 13.4 | 22.1 | 14.4 | 16.8 | 22.0 | 22.2 | 20.9 | 23.8 | 24.1 | 21.7 | 25.6 |
| GPT-4turbo | 30.26 | 37.7 | 33.3 | 46.4 | 25.0 | 28.6 | 25.3 | 15.4 | 27.8 | 31.9 | 30.6 | 29.0 | 31.9 | 28.7 | 37.9 | 17.4 | 23.2 |
| GPT-4o | 30.39 | 42.0 | 39.3 | 49.3 | 28.9 | 25.6 | 22.4 | 24.0 | 23.3 | 29.4 | 17.3 | 29.8 | 30.1 | 29.1 | 44.8 | 34.8 | 17.9 |
| Human Performance | | | | | | | | | | | | | | | | | |
| Human | 68.82 | 55.1 | 78.6 | 99.6 | 98.4 | 43.5 | 98.5 | 91.3 | 62.2 | 61.3 | 33.5 | 47.2 | 73.5 | 87.3 | 93.1 | 99.8 | 69.0 |

Table 3: Influence of Chain-of-Thought on model performances across various mathematical subjects

| Model | CoT | Overall | Alg | AnaG | Ari | CombG | Comb | Cnt | DescG | GrphT | Log | Angle | Area | Len | SolG | Stat | Topo | TransG |
|---|---|---|---|---|---|---|---|---|---|---|---|---|---|---|---|---|---|---|
| GPT-4V | ✗ | 22.76 | 27.3 | 32.1 | 35.7 | 21.1 | 16.7 | 13.4 | 22.1 | 14.4 | 16.8 | 22.0 | 22.2 | 20.9 | 23.8 | 24.1 | 21.7 | 25.6 |
| | ✓ | 23.98 | 26.7 | 26.2 | 38.6 | 22.1 | 24.4 | 19.4 | 27.9 | 23.3 | 25.2 | 17.3 | 21.4 | 23.4 | 23.8 | 25.9 | 4.4 | 25.6 |
| | | +1.22 | -0.58 | -5.95 | +2.86 | +0.98 | +7.73 | +5.97 | +5.76 | +8.89 | +8.4 | -4.63 | -0.8 | +2.45 | - | -1.72 | -17.39 | - |
| Gemini Pro | ✗ | 17.66 | 15.1 | 10.7 | 20.7 | 20.1 | 11.9 | 7.5 | 20.2 | 21.1 | 16.8 | 19.1 | 19.0 | 20.0 | 14.3 | 13.8 | 17.4 | 20.8 |
| | ✓ | 14.34 | 13.9 | 19.1 | 27.9 | 14.0 | 9.5 | 11.9 | 18.3 | 14.4 | 14.3 | 15.6 | 13.2 | 13.1 | 10.3 | 12.1 | 13.0 | 17.9 |
| | | -3.32 | -1.16 | +8.34 | +7.15 | -6.17 | -2.38 | +4.48 | -1.92 | -6.67 | -2.52 | -3.47 | -5.8 | -6.9 | -4.09 | -1.72 | -4.35 | -2.97 |
| Qwen-VL-Max | ✗ | 15.59 | 10.7 | 19.1 | 20.0 | 16.9 | 12.5 | 17.9 | 16.4 | 12.2 | 21.0 | 19.8 | 11.5 | 19.8 | 11.5 | 20.7 | 13.0 | 17.3 |
| | ✓ | 12.63 | 10.4 | 22.6 | 20.0 | 12.3 | 5.4 | 13.4 | 19.2 | 13.3 | 18.5 | 12.1 | 10.4 | 12.3 | 10.7 | 19.0 | 8.7 | 14.3 |
| | | -2.96 | -0.29 | +3.57 | - | -4.54 | -7.14 | -4.48 | +2.88 | +1.11 | -2.52 | -1.15 | -3.8 | -7.57 | -0.82 | -1.72 | -4.34 | -2.97 |
| InternLM-XComposer2-VL | ✗ | 14.54 | 9.3 | 15.5 | 12.1 | 15.3 | 11.3 | 10.5 | 14.4 | 22.2 | 19.3 | 19.7 | 15.6 | 15.0 | 11.9 | 15.5 | 26.1 | 15.5 |
| | ✓ | 14.37 | 10.4 | 14.3 | 14.3 | 15.9 | 14.3 | 7.5 | 16.4 | 26.7 | 19.3 | 15.0 | 15.2 | 11.4 | 13.1 | 8.6 | 26.1 | 18.5 |
| | | -0.17 | +1.15 | -1.19 | +2.15 | +0.65 | +2.98 | -2.99 | +1.93 | +4.45 | - | -4.62 | -0.4 | -3.56 | +1.22 | -6.9 | - | +2.97 |

Specific to subjects, our datasets not only offer better coverage of subjects but also have more problems per subject on average (190 v.s. 63). Note that in 3/8 subjects (Linear Algebra, Group Theory, and Logic) covered by MMMU-Math, there are less than 10 problems. More qualitative comparisons can be found in Appendix G.2.

## 4 Experiments

We conduct experiments to evaluate models' performance on MATH-V. Our findings indicate that accuracy remains low across even the most advanced models. We assess a variety of models, including LLMs and LMMs, and compare both closed- and open-source models, observing that closed-source models perform better. We also did a detailed error analysis of GPT-4V, the first widely used LMM, delving into its error distribution and providing corresponding qualitative examples.

### 4.1 Experimental Setup

**Models.** We conducted experiments on (a) LLMs: ChatGPT-3.5 [15] and GPT-4 [1], augmented with image captions generated by GPT-4V, (b) Open-source LMMs: LLaVA-v1.5 [47], SPHINX [9], ShareGPT-4V [48], InternLM-XComposer2-VL [8] and InternVL-Chat-V1-2-Plus [49], (c) Closed-source LMMs: Qwen-VL [11], Gemini Pro [7], Gemini 1.5 Pro [23] and the GPT-4 series.

**Implementation details.** Our evaluations are carried out in a zero-shot setting, both with and without Chain-of-Thought prompting [50] on several prominent models, to determine the models' ability

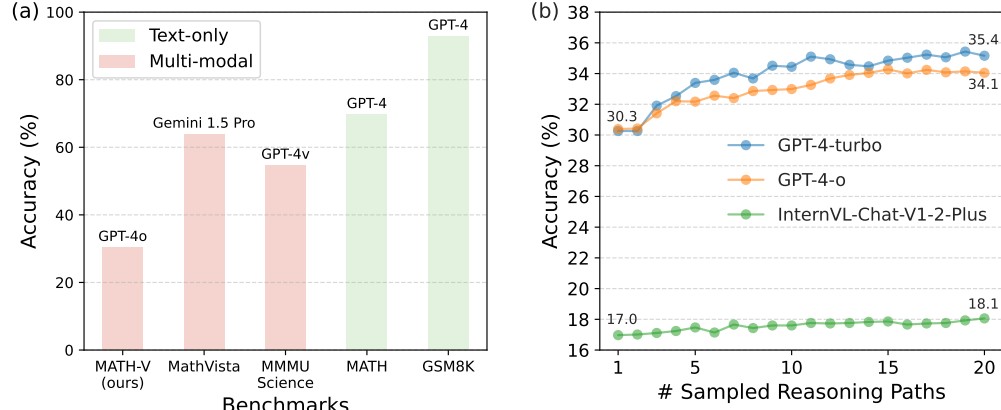

Figure 3: (a) State-of-the-art (SOTA) performance and corresponding model of several mathematical benchmarks in **zero-shot** settings. Our `MATH-V` benchmark is markedly lower than existing text-only and multi-modal mathematical benchmarks. Results of other benchmarks are either from their official website or Papers With Code. (b) Although current models achieve relatively low scores on `MATH-V`, employing 20 times majority voting can still increase accuracy by up to 5%, indicating that models have the potential to address these questions effectively.

to produce accurate answers without the need for fine-tuning or few-shot demonstrations on our benchmark. For human performance, we recruited 100 annotators possessing a high school degree or higher, and they were asked to solve the questions in `MATH-V` independently, thereby obtaining the human performance baseline. Details can be found in Appendix K.3. For other models, we utilize the default prompts provided for multiple-choice or open-ended questions, when available. In cases where models do not offer prompts for the task types present in the `MATH-V` dataset, we engage in prompt engineering and select the most effective prompt for use in the zero-shot configuration during the main experiments. Detailed information about our prompts is available in the Appendix E.

## 4.2 Experimental Results

In this section, we compare the performance of several best-performing Large Language Models (LLMs) and Large Multimodal Models (LMMs) on `MATH-V`, as detailed in Table 2 and Table 3.

**Robustness of** `MATH-V` comes from three aspects. First, to provide a more rigorous evaluation of the models' generalization capabilities, `MATH-V` remains entirely unseen by the LMMs. This differs from existing benchmarks, which may include training data from the same domain. This stricter evaluation is reflected by the lower SOTA scores on `MATH-V` compared to those on mainstream mathematical reasoning benchmarks, as shown in Figure 3.a. Furthermore, our dataset's multiple-choice questions all feature five options, while existing benchmarks typically offer fewer options or simple yes/no questions. This reduces the chances of correct guesses, as evidenced by a much lower random chance score (5.86 for `MATH-V` vs. 17.9 for MathVista). Lastly, our holistic data collection approach jointly collects images and corresponding questions from real exams rather than relying on additional annotation. The lower performance of text-only models like GPT-4 on `MATH-V` further demonstrates the dependency of our questions on the associated images. This approach not only results in longer and more diverse questions but also requires models to possess more sophisticated visual recognition and mathematical reasoning capabilities.

**Considerable room for improvement.** As shown in Table 2, both closed- and open-source models have achieved significantly lower scores on our dataset compared to humans, with the highest scores being 30.39 and 16.97, respectively, while the average human score is 68.82, highlighting the considerable room for improving LMMs' capabilities. As depicted in Figure 3.b, with majority voting of 20 reasoning paths, models like GPT-4-turbo can achieve a 5% increase in accuracy, indicating that these models have the potential to address these questions. Moreover, in less than eight months' evolution from GPT-4V to GPT-4o, there was a notable increase in accuracy on `MATH-V`, from 22.76% to 30.39%, consistent with significant accuracy gains observed in other benchmarks such as MathVista and MMMU. Other open-source models, such as InternVL-Chat-V1-2-Plus, have achieved

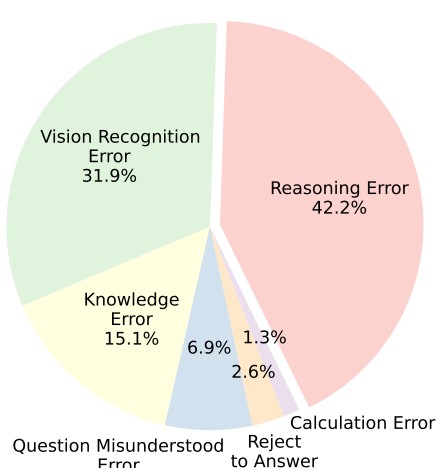

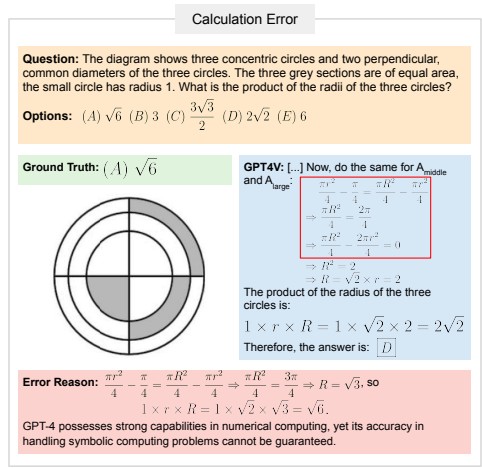

Figure 4: Error distribution of 232 GPT-4V wrong results on the testmini subset of `MATH-V`.

Figure 5: A typical calculation error. Complete answers and more examples can be found in the Appendix I.

enhanced performance on `MATH-V` by employing a stronger language model and scaled fine-tuning datasets. These advancements indicate that LMMs can continue to make progress on `MATH-V` through innovations in model architecture, enhancements in training data, etc.

**Comparison among different subjects.** The dataset covers a broad spectrum of mathematical subjects, from algebra to transformation geometry, each presenting unique complexities. For example, even the high-performing GPT-4o struggles to achieve more than 20% accuracy in transformation geometry. This contrasts with its relatively higher accuracy in arithmetic, where it achieves 49.3%. These disparities across subjects highlight the model's strengths in numerical calculations and its weaknesses in understanding spatial position transformations.

**Disparity of closed- and open-source models.** There is a notable performance disparity between closed- and open-source models, different from the results on MathVista. Closed-source models, including GPT-4, GPT-4-turbo, GPT-4V, Gemini-1.5 Pro, and Gemini Pro, all outperform the best open-source model. For instance, GPT-4 achieved the highest overall accuracy at 30.39%, excelling in subjects like algebra (42.0%), analytic geometry (39.3%), and arithmetic (49.3%). In contrast, the highest-performing open-source model, InternVL-Chat-V1-2-Plus, only reached an overall accuracy of 16.97%. However, results from MathVista show that the accuracy of InternVL-Chat-V1-2-Plus is 59.9%, which is 10% higher than that of GPT-4V, which scored 49.9%. This discrepancy may be attributed to the use of the training set of MathVista during the supervised fine-tuning stage of InternVL-Chat-V1-2-Plus, as training data for both models is unavailable for our dataset.

**Failure of CoT prompting on some models.** Chain-of-Thought (CoT) prompting did not consistently enhance performance across models. While GPT-4V benefited from CoT (22.76% to 23.98%), other models like Gemini Pro and Qwen-VL-Max saw a decrease in performance. We found many LMMs tend to generate very short responses like "The answer is ..." without reasoning steps, even when required to solve the question step by step. This suggests that the step-by-step reasoning capabilities of LMMs are still not mature and cannot be fully exploited by CoT prompting.

### 4.3 Error Analysis

**Quantitative and qualitative analysis on GPT-4V.** In this section, we meticulously analyzed the causes of errors in GPT-4V through both quantitative and qualitative approaches, detailed in Figure 4 and Figure 5. The most prevalent cause of errors, accounting for 42.2%, was identified as Reasoning Error. This indicates a significant challenge in the model's logical processing and inferential reasoning capabilities. Following closely, Vision Recognition Error was the second most common issue, responsible for 31.9% of errors. This highlights difficulties in accurately interpreting visual data, suggesting that the vision encoder remains a bottleneck in multi-modal models. Knowledge Error,

making up 15.1% of the total errors, pointed to gaps between the model's information database and its application. Furthermore, Calculation Error accounted for the smallest percentage of errors at 1.3%, indicating that GPT-4V is relatively robust in numerical and computational tasks, which historically have been challenging for language models. Notably, all errors in the Calculation Error category related to symbolic computation, as shown in Figure 5, rather than numerical computation, suggesting a potential area for future improvement. Detailed examples of these errors can be found in Appendix I.

**Qualitative analysis on GPT-4 with image captions.** The integration of image captions generated by GPT-4V enables GPT-4 to reach an accuracy of 13.10%, comparable to that of SPHINX and Qwen-VL. In disciplines such as algebra, analytic geometry, arithmetic, combinatorics, and counting, GPT-4 with image captions outperforms Gemini Pro and all open-source models except InternVL-Chat-V1-2-Plus. Conversely, in other subjects, its performance notably lags behind these models. Through qualitative analysis of these categories, we observed that subjects like algebra and analytic geometry are more likely to receive relatively accurate descriptions from GPT-4V, unlike other disciplines. This discrepancy underscores the need for LMMs to enhance their capabilities in fine-grained image understanding and description. Remarkably, in some areas (e.g., combinatorics and counting), GPT-4 with image captions even surpasses GPT-4V. This success can be attributed to GPT-4's reasoning capabilities, which evidently contribute to a better understanding and solving of problems. Examples supporting this qualitative analysis are provided in the Appendix J.

## 5 Conclusion and Limitations

In this paper, we propose a benchmark dataset, called `MATH-V`, for evaluating the mathematical reasoning abilities of multimodal models. `MATH-V` consists of 3,040 math reasoning problems, each accompanied by images and texts. We have carefully evaluated a variety of open-source and closed-source models using our `MATH-V` and conducted a systematic analysis. Our benchmark reveals a significant gap between human performance and that of current multimodal models in the task of multimodal reasoning, and facilitates more effective evaluation for future studies in this field.

However, our work does have certain limitations. First, all of the questions in our dataset are in English. Therefore, the dataset cannot evaluate models' capabilities to reason in languages other than English. Also, the images in our benchmark are all abstract, with no pictures taken in real life. These limitations are important and warrant further investigation.

## 6 Acknowledgements

This project is funded in part by National Key R&D Program of China Project 2022ZD0161100, by the Centre for Perceptual and Interactive Intelligence (CPII) Ltd under the Innovation and Technology Commission (ITC)'s InnoHK, by General Research Fund of Hong Kong RGC Project 14204021. Hongsheng Li is a PI of CPII under the InnoHK. This work was conducted using content provided by Association Kangourou Sans Frontières – AKSF, `https://www.aksf.org`. We are grateful to Meike Akveld and Robert eretschläger for their support of this project.

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

# Appendix

## A  Ethics Statement

All questions of MATH-V are from publicly available sources. Although we do not commercialize MATH-V, we should like to demonstrate that we are far from the boundary for action or infringement.

**Legal compliance.** MATH-V problems are created by the the Mathematical Association of America (MAA), Kangaroo of Mathematics in Austria (KMA) and United Kingdom Maths Trust (UKMT). For decades, the MAA has not protected its problem IP even from separate organizations which sell MAA problems, such as AoPS. Courts have ruled that this implies the IP rights are permanently forfeited. Both of the KMA and UKMT are leading charities that advances the education of young people in mathematics, and MATH-V only contains free test questions published on their official website.

For MATH-V, we abide by Fair Use §107: "the fair use of a copyrighted work, including such use by ⋯ scholarship, or research, is not an infringement of copyright", where fair use is determined by "the purpose and character of the use, including whether such use is of a commercial nature or is for nonprofit educational purposes" and "the effect of the use upon the potential market for or value of the copyrighted work."

**Dataset intended usage and license.** We document the MATH-V dataset within the paper and note that the MATH-V and code for results reproduction will be publicly available. The MATH-V dataset is not supposed, though the risk exists, to be used to train models for cheating. We intend for researchers to use this dataset to better evaluate LMMs' mathematical reasoning capabilities and consequently facilitate future studies in this area. We bear all responsibility in case of violation of rights. The MATH-V data and our open-source code are under an MIT license.

## B  Dataset Statistics

| Statistic | Number |
|---|---|
| Total questions | 3,040 |
|  - multiple-choice questions | 1,532 (50.4%) |
|  - Free-form questions | 1,508 (49.6%) |
|  - Questions in testmini | 304 (10.0%) |
| Multiple-choice questions | 1,532 |
|  - Proportion of answers A | 299(19.5%) |
|  - Proportion of answers B | 311(20.3%) |
|  - Proportion of answers C | 320(20.9%) |
|  - Proportion of answers D | 326(21.2%) |
|  - Proportion of answers E | 278(18.1%) |
| Unique number of images | 3,472 |
| Unique number of answers | 370 |
| Levels | 5 |
| Subjects | 16 |
| Source competitions | 19 |
|  - Math Kangaroo | 1,499 |
|  - UK (Grey,Pink,Junior,...) | 535 |
|  - AMC (8,10,12), AIME | 759 |
| Maximum question length | 280 |
| Minimum question length | 5 |
| Average question length | 42.3 |

Table 4: Key statistics of MATH-V. The unit of question length is words.

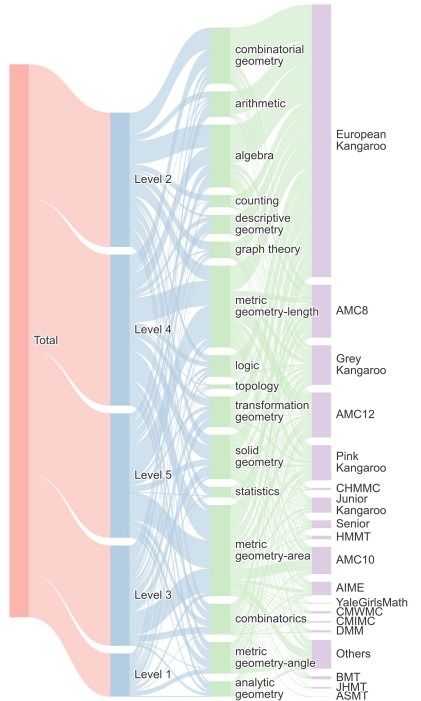

Figure 6: Levels, subjects and sources distribution of MATH-V.

## C  Data Contamination

We have also considered the issue of data contamination:

Firstly, we primarily collect questions from the official websites of various competitions. Each file typically contains 15 to 30 questions and is kept separate from the answers or solutions. Given the concerns surrounding data contamination in large language models (LLMs), we prioritize selecting questions that do not have readily available answers, such as those found on separate documents or pages.

Secondly, the three main series of closed-source models, GPT-4, Gemini, and Claude, all evaluate their models using several contests in their technical reports, including the AMC, which is a primary source of our problems. Therefore, we assume that these models adhere to the common practice of not using leaked data for training. However, as these are closed-source models, this assumption is our best estimation.

Thirdly, among open-source models, the most widely used base model is LLaVA, which utilizes a subset of the CC-3M dataset for pretraining and LAION/CC/SBU BLIP captions for instruction tuning. We compared our dataset with the training dataset and found no overlap between the two.

## D  Testmini

In this section, we show the effectiveness of the testmini subset in mirroring the performance trends observed in the full test set across a diverse range of mathematical subjects. The detailed results are presented in Table 5. We find that the model performances, when evaluated on both the entire set and testmini, exhibit close parallelism in trend. For instance, the performance delta between the full test set and testmini for the Qwen-VL-Plus model is a mere 0.19%, indicating a high degree of similarity in model behavior across the two datasets. Similarly, the GPT4V model also exhibits a negligible performance difference of -0.39%, further corroborating the testmini subset's capability to accurately reflect the full test set's characteristics. Furthermore, even for LLaVA-v1.5-13B and Share-GPT4V-7B, two models with the largest performance variations, where differences of 1.98% and 1.97% are observed respectively, the deviation remains within 2%, underscoring the testmini subset's reliability for performance evaluation. These findings suggest that models' performance on the testmini subset should be indicative of their performance on the full test set, thereby providing a reliable means for preliminary evaluation and comparison of models across various mathematical subjects.

## E  Evaluation Details

We conduct all experiments on A100 80G GPUs.

### E.1  Prompts for Response Generation

In this section, we present details of the prompts used for the evaluation of response generation models. The prompts are designed to ensure that the models generate responses that are consistent with the input question and the image. The prompts used for the evaluation of response generation models are shown in Table 6.

### E.2  Prompt for Caption Generation

The prompt for caption generation is as follows: *Here is a diagram of a math problem, please describe the diagram in complete detail so that your description can be used to replace the diagram for math problem-solving.*

### E.3  Models

We present the source of the models used in the evaluation in Table 7. The models are sourced from OpenAI, Google, Alibaba, and Hugging Face. The models from OpenAI, Google and Alibaba are accessed from their respective platforms through API calls.

| Model | Overall | Alg | AnaG | Ari | CombG | Comb | Cnt | DescG | GrphT | Log | Angle | Area | Len | SolG | Stat | Topo | TransG |
|---|---|---|---|---|---|---|---|---|---|---|---|---|---|---|---|---|---|
| Random Chance | 5.86 | 2.0 | 11.9 | 4.3 | 9.4 | 3.0 | 1.5 | 9.6 | 1.1 | 5.9 | 0.6 | 7.0 | 7.6 | 5.7 | 3.5 | 8.7 | 8.3 |
| Testmini | 7.57 | 5.3 | 31.6 | 10.5 | 15.8 | 0.0 | 0.0 | 5.3 | 5.3 | 5.3 | 5.3 | 5.3 | 10.5 | 10.5 | 0.0 | 5.3 | 5.3 |
| Δ | +1.71 | +3.3 | +19.7 | +6.2 | +6.4 | -3.0 | -1.5 | -4.4 | +4.2 | -0.6 | +4.7 | -1.7 | +3.0 | +4.8 | -3.5 | -3.4 | -3.1 |
| LLMs (Text Only) | | | | | | | | | | | | | | | | | |
| GPT-4 (CoT) | 8.16 | 12.8 | 10.7 | 15.7 | 4.9 | 10.7 | 10.5 | 1.9 | 5.6 | 8.4 | 8.1 | 6.2 | 8.7 | 8.6 | 3.5 | 4.4 | 4.8 |
| Testmini | 6.58 | 5.3 | 10.5 | 15.8 | 0.0 | 21.1 | 10.5 | 5.3 | 0.0 | 5.3 | 10.5 | 5.3 | 0.0 | 5.3 | 5.3 | 5.3 | 0.0 |
| Δ | -1.58 | -7.5 | -0.2 | +0.1 | -4.9 | +10.4 | 0.0 | +3.4 | -5.6 | -3.1 | +2.4 | -0.9 | -8.7 | -3.3 | +1.8 | +0.9 | -4.8 |
| LLMs (Text + Image Caption) | | | | | | | | | | | | | | | | | |
| ChatGPT-3.5 (CoT) | 9.74 | 10.7 | 20.0 | 18.6 | 10.1 | 7.7 | 17.9 | 16.4 | 10.0 | 13.5 | 6.4 | 5.8 | 6.5 | 9.4 | 12.1 | 4.4 | 10.7 |
| Testmini | 10.20 | 10.5 | 26.3 | 5.3 | 0.0 | 10.5 | 21.1 | 15.8 | 10.5 | 0.0 | 10.5 | 0.0 | 5.3 | 21.1 | 5.3 | 10.5 | 5.3 |
| Δ | +0.46 | -0.2 | +6.3 | -13.3 | -10.1 | +2.8 | +3.2 | -0.6 | +0.5 | -13.5 | +4.1 | -5.8 | -1.2 | +11.7 | -6.8 | +6.1 | -5.4 |
| GPT-4 (CoT) | 13.10 | 16.5 | 20.2 | 34.3 | 10.4 | 17.9 | 19.4 | 7.7 | 11.1 | 10.1 | 9.8 | 9.6 | 9.1 | 13.5 | 13.8 | 8.7 | 12.5 |
| Testmini | 12.50 | 15.8 | 10.5 | 31.6 | 5.3 | 15.8 | 31.6 | 10.5 | 15.8 | 15.8 | 0.0 | 5.3 | 5.3 | 0.0 | 21.1 | 10.5 | 5.3 |
| Δ | -0.6 | -0.7 | -9.7 | -2.7 | -5.1 | -2.1 | +12.2 | +2.8 | +4.7 | +5.7 | -9.8 | -4.3 | -3.8 | -13.5 | +7.3 | +1.8 | -7.2 |
| Open-source LMMs (Text + Image) | | | | | | | | | | | | | | | | | |
| LLaVA-v1.5-7B | 8.52 | 7.0 | 7.1 | 10.7 | 7.1 | 4.8 | 10.5 | 7.7 | 10.0 | 9.2 | 15.6 | 10.2 | 9.8 | 5.3 | 8.6 | 4.4 | 4.8 |
| Testmini | 10.20 | 0.0 | 10.5 | 15.8 | 5.3 | 5.3 | 15.8 | 10.5 | 10.5 | 15.8 | 21.1 | 15.8 | 15.8 | 5.3 | 10.5 | 0.0 | 5.3 |
| Δ | +1.68 | -7.0 | +3.4 | +5.1 | -1.8 | +0.5 | +5.3 | +2.8 | +0.5 | +6.6 | +5.5 | +5.6 | +6.0 | 0.0 | +1.9 | -4.4 | +0.5 |
| SPHINX (V2) | 9.70 | 6.7 | 7.1 | 12.9 | 7.5 | 7.7 | 6.0 | 9.6 | 16.7 | 10.1 | 11.0 | 11.8 | 12.5 | 8.2 | 8.6 | 8.7 | 6.0 |
| Testmini | 9.21 | 5.3 | 10.5 | 10.5 | 0.0 | 21.1 | 10.5 | 10.5 | 15.8 | 15.8 | 15.8 | 5.3 | 10.5 | 0.0 | 5.3 | 5.3 | 10.5 |
| Δ | -0.49 | -1.4 | +3.4 | -2.4 | -7.5 | +13.4 | +4.5 | +0.9 | -0.9 | +0.4 | +4.8 | -6.5 | -2.0 | -8.2 | -3.3 | -3.4 | +4.5 |
| ShareGPT4V-7B | 10.53 | 5.5 | 3.6 | 12.9 | 10.1 | 4.8 | 7.5 | 11.5 | 14.4 | 10.9 | 16.2 | 11.8 | 12.3 | 9.8 | 15.5 | 17.4 | 11.3 |
| Testmini | 12.50 | 5.3 | 0.0 | 10.5 | 21.1 | 5.3 | 5.3 | 26.3 | 15.8 | 15.8 | 15.8 | 10.5 | 21.1 | 15.8 | 15.8 | 10.5 | 5.3 |
| Δ | +1.97 | -0.2 | -3.6 | -2.4 | +11.0 | +0.5 | -2.2 | +14.8 | +1.4 | +4.9 | -0.4 | -1.3 | +8.8 | +6.0 | +0.3 | -6.9 | -6.0 |
| LLaVA-v1.5-13B | 11.12 | 7.0 | 14.3 | 14.3 | 9.1 | 6.6 | 6.0 | 13.5 | 5.6 | 13.5 | 10.4 | 12.6 | 14.7 | 11.5 | 13.8 | 13.0 | 10.7 |
| Testmini | 13.10 | 10.4 | 5.3 | 15.8 | 5.3 | 10.5 | 10.5 | 26.3 | 5.3 | 15.8 | 5.3 | 15.8 | 31.6 | 10.5 | 15.8 | 15.8 | 10.5 |
| Δ | +1.98 | +3.4 | -9.0 | +1.5 | -3.8 | +3.9 | +4.5 | +12.8 | -0.3 | +2.3 | -5.1 | +3.2 | +16.9 | -1.0 | +2.0 | +2.8 | -0.2 |
| ShareGPT4V-13B | 11.88 | 7.5 | 15.5 | 16.4 | 10.7 | 8.9 | 9.0 | 11.5 | 8.9 | 7.6 | 11.6 | 13.0 | 17.4 | 10.3 | 8.6 | 8.7 | 12.5 |
| Testmini | 13.49 | 15.8 | 21.1 | 10.5 | 5.3 | 15.8 | 10.5 | 15.8 | 10.5 | 15.8 | 5.3 | 10.5 | 36.8 | 21.1 | 5.3 | 10.5 | 5.3 |
| Δ | +1.61 | +8.3 | +5.6 | -5.9 | -5.4 | +6.9 | +1.5 | +4.3 | +1.6 | +8.2 | -6.3 | -2.5 | +19.4 | +10.8 | -3.3 | +1.8 | -7.2 |
| SPHINX-MoE | 14.18 | 7.8 | 17.9 | 14.3 | 15.6 | 9.5 | 11.9 | 12.5 | 12.6 | 10.5 | 12.6 | 16.2 | 15.6 | 17.8 | 13.5 | 12.1 | 16.1 |
| Testmini | 13.49 | 10.5 | 15.8 | 15.8 | 15.8 | 10.5 | 10.5 | 10.5 | 15.8 | 10.5 | 15.8 | 15.8 | 15.8 | 15.8 | 10.5 | 10.5 | 15.8 |
| Δ | -0.69 | +2.7 | -2.1 | +1.5 | +0.2 | +1.0 | -1.4 | -2.0 | +3.2 | 0.0 | +3.2 | -0.4 | +0.2 | -2.0 | -2.3 | -1.6 | -0.3 |
| InternLM-XComposer2-VL | 14.54 | 9.3 | 15.5 | 12.1 | 15.3 | 11.3 | 10.5 | 14.4 | 22.2 | 19.3 | 19.7 | 15.6 | 15.0 | 11.9 | 15.5 | 26.1 | 15.5 |
| Testmini | 15.79 | 10.5 | 15.8 | 10.5 | 15.8 | 10.5 | 10.5 | 15.8 | 21.1 | 21.1 | 21.1 | 15.8 | 15.8 | 10.5 | 15.8 | 26.3 | 15.8 |
| Δ | +1.25 | +1.2 | +0.3 | -1.6 | +0.5 | -0.8 | 0.0 | +1.4 | -1.1 | +1.8 | +1.4 | +0.2 | +0.8 | -1.4 | +0.3 | +0.2 | +0.3 |
| InternVL-Chat-V1-2-Plus | 16.97 | 11.3 | 25.0 | 15.7 | 16.9 | 10.1 | 11.9 | 16.4 | 15.6 | 19.3 | 22.5 | 16.4 | 22.5 | 14.3 | 17.2 | 4.4 | 20.8 |
| Testmini | 18.42 | 21.1 | 26.3 | 10.5 | 21.1 | 5.3 | 10.5 | 10.5 | 15.8 | 10.5 | 31.6 | 31.6 | 26.3 | 10.5 | 36.8 | 5.3 | 21.1 |
| Δ | +1.45 | +9.8 | +1.3 | -5.2 | +4.2 | -4.9 | -1.4 | -5.8 | +0.2 | -8.8 | +9.0 | +15.2 | +3.8 | -3.8 | +19.6 | +0.9 | +0.2 |
| Closed-source LMMs (Text + Image) | | | | | | | | | | | | | | | | | |
| Qwen-VL-Plus | 10.72 | 11.3 | 17.9 | 14.3 | 12.7 | 4.8 | 10.5 | 15.4 | 8.9 | 14.3 | 11.6 | 6.4 | 10.0 | 14.3 | 6.9 | 8.7 | 11.31 |
| Testmini | 10.53 | 26.3 | 10.5 | 15.8 | 15.8 | 10.5 | 21.1 | 5.3 | 10.5 | 10.5 | 10.5 | 5.3 | 10.5 | 10.5 | 5.3 | 5.3 | 0.0 |
| Δ | -0.19 | +15.0 | -7.4 | -3.8 | +3.1 | +5.7 | +10.6 | -10.1 | +1.6 | -3.8 | -1.1 | -1.1 | +0.5 | -3.8 | -1.6 | -3.4 | -11.31 |
| Qwen-VL-Max | 15.59 | 10.7 | 19.1 | 20.0 | 16.9 | 12.5 | 17.9 | 16.4 | 12.2 | 21.0 | 13.3 | 14.2 | 19.8 | 11.5 | 20.7 | 13.0 | 17.3 |
| Testmini | 16.1 | 10.5 | 21.1 | 21.1 | 15.8 | 15.8 | 15.8 | 15.8 | 10.5 | 21.1 | 10.5 | 15.8 | 21.1 | 10.5 | 21.1 | 15.8 | 15.8 |
| Δ | +0.51 | -0.2 | +2.0 | +1.1 | -1.1 | +3.3 | -2.1 | -0.6 | -1.7 | +0.1 | -2.8 | +1.6 | +1.3 | -1.0 | +0.4 | +2.8 | -1.5 |
| Gemini Pro | 17.66 | 15.1 | 10.7 | 20.7 | 20.1 | 11.9 | 7.5 | 20.2 | 21.1 | 16.8 | 19.1 | 19.0 | 20.0 | 14.3 | 13.8 | 17.4 | 20.8 |
| Testmini | 17.11 | 15.8 | 10.5 | 21.1 | 21.1 | 10.5 | 5.3 | 21.1 | 21.1 | 15.8 | 21.1 | 21.1 | 21.1 | 15.8 | 15.8 | 15.8 | 21.1 |
| Δ | -0.55 | +0.7 | -0.2 | +0.4 | +1.0 | -1.4 | -2.2 | +0.9 | 0.0 | -1.0 | +2.0 | +2.1 | +1.1 | +1.5 | +2.0 | -1.6 | +0.3 |
| Gemini-1.5 Pro | 19.24 | 20.3 | 35.7 | 34.3 | 19.8 | 15.5 | 20.9 | 26.0 | 26.7 | 22.7 | 14.5 | 14.4 | 16.5 | 18.9 | 10.3 | 26.1 | 17.3 |
| Testmini | 21.05 | 31.6 | 36.8 | 26.3 | 31.6 | 10.5 | 10.5 | 31.6 | 26.3 | 21.1 | 5.3 | 10.5 | 15.8 | 5.3 | 10.5 | 26.3 | 36.8 |
| Δ | +1.81 | +11.3 | +1.1 | -8.0 | +11.8 | -5.0 | -10.4 | +5.6 | -0.4 | -1.6 | -9.2 | -3.9 | -0.7 | -13.6 | +0.2 | +0.2 | +19.5 |
| GPT-4V | 22.76 | 27.3 | 32.1 | 35.7 | 21.1 | 16.7 | 13.4 | 22.1 | 14.4 | 16.8 | 22.0 | 22.2 | 20.9 | 23.8 | 24.1 | 21.7 | 25.6 |
| Testmini | 22.37 | 26.3 | 31.6 | 36.8 | 21.1 | 15.8 | 10.5 | 21.1 | 15.8 | 15.8 | 21.1 | 21.1 | 21.1 | 26.3 | 26.3 | 21.1 | 26.3 |
| Δ | -0.39 | -1.0 | -0.5 | +1.1 | 0.0 | -0.9 | -2.9 | -1.0 | +1.4 | -1.0 | -0.9 | -1.1 | +0.2 | +2.5 | +2.2 | -0.6 | +0.7 |
| GPT-4o | 30.39 | 42.0 | 39.3 | 49.3 | 28.9 | 25.6 | 22.4 | 24.0 | 23.3 | 29.4 | 17.3 | 29.8 | 30.1 | 29.1 | 44.8 | 34.8 | 17.9 |
| Testmini | 29.93 | 42.1 | 42.1 | 57.9 | 42.1 | 21.1 | 21.1 | 31.6 | 36.8 | 26.3 | 5.3 | 31.6 | 21.1 | 10.5 | 36.8 | 36.8 | 15.8 |
| Δ | -0.46 | +0.1 | +2.8 | +8.6 | +13.2 | -4.5 | -1.3 | +7.6 | +13.5 | -3.1 | -12.0 | +1.8 | -9.0 | -18.6 | -8.0 | +2.0 | -2.1 |
| GPT-4turbo | 30.26 | 37.7 | 33.3 | 46.4 | 25.0 | 28.6 | 25.4 | 15.4 | 26.7 | 31.9 | 30.6 | 29.0 | 31.9 | 29.1 | 37.9 | 17.4 | 23.2 |
| Testmini | 29.93 | 36.8 | 36.8 | 36.8 | 31.6 | 36.8 | 31.6 | 15.8 | 36.8 | 36.8 | 31.6 | 31.6 | 31.6 | 21.1 | 26.3 | 10.5 | 26.3 |
| Δ | -0.33 | -0.9 | +3.5 | -9.6 | +6.6 | +8.2 | +6.2 | +0.4 | +10.1 | +4.9 | +1.0 | +2.6 | -0.3 | -8.0 | -11.6 | -6.9 | +3.1 |

Table 5: Comparison of model performances between the entire dataset and testmini subset.

# F More Results on Text-only LLMs and LMMs.

In Table 2, we augment the text-only LLM like ChatGPT-3.5 [15] and GPT-4 [1] with image captions generated by GPT-4V [6]. Considering current text-only math LLMs like MathCoder [51], WizardMath [52], and Mammoth [53], we further evaluate three 7B models on MATH-V using the same method applied to ChatGPT-3.5 and GPT-4, as shown in Table 8. We can find that MAmmoTH-7B-Mistral based on the current 7B base model with the strongest reasoning ability reached 12.4%, even exceeding LLaVA-v1.5-13B (11.12%) and Qwen-VL-Plus (10.72%). This not only shows that the reasoning ability of existing multi-modal models is weak, but also shows that our benchmark has higher requirements for reasoning capabilities.

As analyzed in [54, 55], the visual content is unnecessary for many questions in benchmarks like MathVista, MMMU, ScienceQA, etc. In Section 4.2, we have demonstrated that the low performance of text-only models like GPT-4 on MATH-V confirms the dependency of our questions on the associated images. To further validate the visual dependency of MATH-V, we evaluate some LMMs with vanilla (text and image) and text-only configurations. As depicted in Table 9, once the LMMs lose the image input, there is a decline in their performance, with a notable drop of 8.55% on InternVL-Chat-V1-2-

| Model | Prompt |
|---|---|
| GPT4-CoT (Text Only) | Please solve the problem step by step and put your answer in one "□". If it is a multiple choice question, only one letter is allowed in the "□". |
| ChatGPT 3.5-CoT (Text + Image Caption ) | Please first solve the problem step by step, then put your final answer or a single letter (if it is a multiple choice question) in one "□". |
| GPT4-CoT (Text + Image Caption ) | Please first solve the problem step by step, then put your final answer or a single letter (if it is a multiple multiple choice question) in one "□". |
| LLaVa-v1.5-7B | Answer the question using a single word or phrase. |
| SPHINX (V2) | Answer the question using a single word or phrase. |
| ShareGPT4V-7B | Answer the question using a single word or phrase. |
| LLaVa-v1.5-13B | Answer the question using a single word or phrase. |
| ShareGPT4V-13B | Answer the question using a single word or phrase. |
| SPHINX-MoE | Answer the question using a single word or phrase. |
| InternVL-Chat-V1-2-Plus | Please solve the problem and put your answer in one "□". If it is a multiple choice question, only one letter letter is allowed in the "□". |
| InternLM-XComposer2-VL | Please solve the problem and put your answer in one "□". If it is a multiple choice question, only one letter letter is allowed in the "□". |
| InternLM-XComposer2-VL -CoT | Please solve the problem step by step and put your answer in one "□". If it is a multiple choice question, only one letter is allowed in the "□". |
| Qwen-VL-Plus | Answer the question using a single word or phrase. |
| Qwen-VL-Max | Answer the question using a single word or phrase. |
| Qwen-VL-Max-CoT | Please solve the problem step by step and put your answer in one "□". If it is a multiple choice question, only one letter is allowed in the "□". |
| Gemini-1.5 Pro | Answer the question using a single letter or number or word. |
| Gemini Pro | Answer the question using a single letter or number or word. |
| Gemini Pro-CoT | Please first solve the problem step by step, then put your final answer or a single letter (if it is a multiple choice question) in one "□". |
| GPT4-V/o/turbo | Please solve the problem and put your answer in one "□". If it is a multiple choice question, only one letter is allowed in the "□". |
| GPT4V-CoT | Please solve the problem step by step and put your answer in one "□". If it is a multiple choice question, only one letter is allowed in the "□". |

Table 6: The prompts used in the evaluation for response generation.

| Model | Source | URL |
|---|---|---|
| ChatGPT 3.5 | gpt-3.5-turbo-0125 | `https://platform.openai.com/` |
| GPT4 | gpt-4-0125-preview | `https://platform.openai.com/` |
| GPT4V | gpt-4-1106-vision-preview | `https://platform.openai.com/` |
| GPT-4o | gpt-4o-2024-05-13 | `https://platform.openai.com/` |
| GPT-4-turbo | gpt-4-turbo-2024-04-09 | `https://platform.openai.com/` |
| Gemini Pro | gemini-pro-vision | `https://ai.google.dev/` |
| Gemini-v1.5 Pro | gemini-1.5-pro | `https://ai.google.dev/` |
| InternLM-XComposer2-VL | local checkpoint | `https://huggingface.co/internlm/internlm-xcomposer2-vl-7b` |
| InternVL-Chat-v1.2-Plus | local checkpoint | `https://huggingface.co/OpenGVLab/InternVL-Chat-V1-2-Plus` |
| Qwen-VL-Max | qwen-vl-max | `https://help.aliyun.com/zh/dashscope/developer-reference/vl-plus-quick-start` |
| Qwen-VL-Plus | qwen-vl-plus | `https://help.aliyun.com/zh/dashscope/developer-reference/vl-plus-quick-start` |
| SPHINX-MoE | local checkpoint | `https://huggingface.co/Alpha-VLLM/LLaMA2-Accessory/tree/main/finetune/mm/SPHINX/SPHINX-MoE` |
| SPHINX (V2) | local checkpoint | `https://huggingface.co/Alpha-VLLM/LLaMA2-Accessory/tree/main/finetune/mm/SPHINX/SPHINX-v2-1k` |
| LLaVa-v1.5-7B | local checkpoint | `https://huggingface.co/liuhaotian/llava-v1.5-7b` |
| LLaVa-v1.5-13B | local checkpoint | `https://huggingface.co/liuhaotian/llava-v1.5-13b` |
| ShareGPT4V-7B | local checkpoint | `https://huggingface.co/Lin-Chen/ShareGPT4V-7B` |
| ShareGPT4V-13B | local checkpoint | `https://huggingface.co/Lin-Chen/ShareGPT4V-13B` |

Table 7: The source of the models used in the evaluation.

| Model | Overall | Alg | AnaG | Ari | CombG | Comb | Cnt | DescG | GrphT | Log | Angle | Area | Len | SolG | Stat | Topo | TransG |
|---|---|---|---|---|---|---|---|---|---|---|---|---|---|---|---|---|---|
| | | | | | | LLMs (Text + Image Caption) | | | | | | | | | | | |
| ChatGPT-3.5 (CoT) | 9.74 | 10.7 | 20.0 | 18.6 | 10.1 | 7.7 | 17.9 | 16.4 | 10.0 | 13.5 | 6.4 | 5.8 | 6.5 | 9.4 | 12.1 | 4.4 | 10.7 |
| GPT-4 (CoT) | 13.10 | 16.5 | 20.2 | 34.3 | 10.4 | 17.9 | 19.4 | 7.7 | 11.1 | 10.1 | 9.8 | 9.6 | 9.1 | 13.5 | 13.8 | 8.7 | 12.5 |
| MathCoder-7B | 6.6 | 4.9 | 15.5 | 12.1 | 9.1 | 6.0 | 7.5 | 11.5 | 5.6 | 7.6 | 2.3 | 4.0 | 4.9 | 5.3 | 8.6 | 13.0 | 11.3 |
| WizardMath-7B | 4.5 | 3.5 | 7.1 | 5.0 | 8.4 | 0.6 | 10.5 | 10.5 | 4.4 | 5.0 | 1.7 | 3.6 | 2.7 | 4.5 | 3.5 | 13.0 | 5.4 |
| MAmmoTH-7B-Mistral | 12.4 | 11.6 | 16.7 | 15.7 | 11.4 | 10.1 | 11.9 | 14.4 | 14.4 | 8.4 | 8.7 | 12.8 | 11.4 | 11.5 | 19.0 | 21.7 | 16.6 |

Table 8: Performance of some text-only LLMs augmented with image captions on the entire dataset.

| Model | Overall | Alg | AnaG | Ari | CombG | Comb | Cnt | DescG | GrphT | Log | Angle | Area | Len | SolG | Stat | Topo | TransG |
|---|---|---|---|---|---|---|---|---|---|---|---|---|---|---|---|---|---|
| Random Chance | 7.57 | 5.3 | 31.6 | 10.5 | 15.8 | 0.0 | 0.0 | 5.3 | 5.3 | 5.3 | 5.3 | 5.3 | 10.5 | 10.5 | 0.0 | 5.3 | 5.3 |
| *LLMs (Text Only)* | | | | | | | | | | | | | | | | | |
| GPT-4 (CoT) | 6.58 | 5.3 | 10.5 | 15.8 | 0.0 | 21.1 | 10.5 | 5.3 | 0.0 | 5.3 | 10.5 | 5.3 | 0.0 | 5.3 | 5.3 | 5.3 | 0.0 |
| *LMMs* | | | | | | | | | | | | | | | | | |
| LLaVA-v1.5-7B | 10.20 | 0.0 | 10.5 | 15.8 | 5.3 | 5.3 | 15.8 | 10.5 | 10.5 | 15.8 | 21.1 | 15.8 | 15.8 | 5.3 | 10.5 | 0.0 | 5.3 |
| Text Only | 8.88 | 10.5 | 15.8 | 10.5 | 5.3 | 5.3 | 0.0 | 15.8 | 10.5 | 10.5 | 5.3 | 10.5 | 10.5 | 5.3 | 5.3 | 10.5 | 10.5 |
| Δ | -1.32 | +10.5 | +5.3 | -5.3 | – | – | -15.8 | +5.3 | – | -5.3 | -15.8 | -5.3 | -5.3 | – | -5.2 | +10.5 | +5.2 |
| LLaVA-v1.5-13B | 13.10 | 10.4 | 5.3 | 15.8 | 5.3 | 10.5 | 10.5 | 26.3 | 5.3 | 15.8 | 5.3 | 15.8 | 31.6 | 10.5 | 15.8 | 15.8 | 10.5 |
| Text Only | 10.86 | 15.8 | 10.5 | 10.5 | 0.0 | 10.5 | 0.0 | 31.6 | 0.0 | 15.8 | 5.3 | 10.5 | 26.3 | 10.5 | 10.5 | 5.3 | 10.5 |
| Δ | -2.24 | +5.4 | +5.2 | -5.3 | -5.3 | – | -10.5 | +5.3 | -5.3 | – | – | -5.3 | -5.3 | – | -5.3 | -10.5 | – |
| InternVL-Chat-V1-2-Plus | 18.42 | 21.1 | 26.3 | 10.5 | 21.1 | 5.3 | 10.5 | 10.5 | 15.8 | 10.5 | 31.6 | 31.6 | 26.3 | 10.5 | 36.8 | 5.3 | 21.1 |
| Text Only | 9.87 | 5.3 | 15.8 | 0.0 | 21.1 | 0.0 | 5.26 | 15.8 | 10.5 | 10.5 | 15.8 | 15.8 | 10.5 | 5.3 | 10.5 | 5.3 | 10.5 |
| Δ | -8.55 | -15.8 | -10.5 | -10.5 | – | -5.3 | -5.2 | +5.3 | -5.3 | – | -15.8 | -15.8 | -15.8 | -5.2 | -26.3 | – | -10.6 |

Table 9: Comparison of model performances between vanilla (text and image) and text-only LMMs on the testmini subset.

Plus, approaching the level of random chance. This highlights the strong visual dependency of our dataset's questions, further underscoring the advantages of our holistic data collection approach.

# G Comparison Details

## G.1 Comparing with MathVista

MathVista contains numerous similar questions with restricted question variability across a variety of source datasets. After eliminating questions with identical stem text, only 4,740 questions remain. For instance, among the 6,141 questions in MathVista, the question *"What is the age gap between these two people in the image?"* appears 270 times. Some images of this question are shown in Figure 7. Furthermore, there are many template questions within its collection that merely alter a few words. For instance, questions containing *"What is the age gap between"* appear 310 times. Another example is questions like *"Subtract all ... objects. How many objects are left?"*. Figure 8 illustrates this type of question involving very similar geometry recognition, which occurs 400 times. In Table 10, we present detailed statistics on similar questions.

We also provide some qualitative comparisons illustrating the different question lengths between MathVista and our dataset. Figure 9 presents the 3 main types of questions featuring an abstract scene in MathVista, constituting over 90% of the total questions, as discussed in Section. 1. We also showcase representative questions in MathVista that involve a function plot in Figure 10, which pose similar, straightforward, and concise inquiries regarding the depicted function graph. For comparison, we present examples of our `MATH-V` on analytic geometry in Figure 11, which feature longer questions and incorporate more complex function concepts, such as mutual symmetry and iterative functions. Moreover, Figure 12 and Figure 13 exhibit topology and graph theory-related questions in our dataset, two categories absent in MathVista, which require intricate visual recognition and mathematical reasoning.

## G.2 Comparing with MMMU

In this section, we showcase some examples of different subjects of MMMU demanding sophisticated college-level domain knowledge. Figure 14, Figure 15, and Figure 16 present the subject of Graph Theory, Group Theory, and Operation Research, respectively. Moreover, typical examples of Topology and Graph Theory in our `MATH-V` are displayed in Figure 13 and Figure 13. Although both sets of subjects are intricate mathematical disciplines, they exhibit notable differences in the nature of their posed questions. Questions in the MMMU dataset often involve advanced college-level mathematical concepts such as Kruskal's algorithm, alternating group structures, and DFS Spanning Tree. In contrast, the questions in our dataset predominantly comprise puzzles that are readily solvable by middle and high school students.

| Template | Proportion | Examples |
|---|---|---|
| **IconQA[56]** | | |
| Object Counting | 106/400 | What number is shown? |
| Perimeter Calculation | 51/400 | What is the perimeter of the square? |
| Length Measurement with a given reference | 150/400 | Move the ruler to measure the length of the line to the nearest centimeter. The line is about _ centimeters long. |
| Time Inquiry based on a clock | 67/400 | What time is shown? |
| **CLEVR-Math[57]** | | |
| Subtract ... How many objects are left? | 400/400 | Subtract all metallic cylinders. Subtract all gray things. How many objects are left? |
| **FigureQA[58]** | | |
| Is X the ...? | 176/400 | Is Khaki the minimum/maximum/ low median/high median/smoothest/ roughest? |
| Does X have the ...? | 86/400 | Does Brown have the maximum/ minimum area under the curve? (the highest/lowest value?) |
| Is X ... Y? | 117/400 | Is Hot Pink less/greater than Black? |
| Does X intersect Y? | 21/400 | Does Slate intersect Light Salmon? |
| **Super-CLEVEL[59]** | | |
| Are there ... X than Y? | 146/400 | Are there fewer/more tiny gray cars than small red sedans? |
| Is the number of X ... than Y? | 232/400 | Is the number of regular buss less/greater than the number of brown rubber minivans? |
| Others | 22/400 | How many cyan double buss have the same size as the matte aeroplane? |
| **KVQA[60]** | | |
| What is the age gap between the ...? | 310/382 | What is the age gap between these two people in image? |
| How many people in the image were born ...? | 72/382 | How many people in the image were born after the end of World War II? |
| **PlotQA[61]** | | |
| In how many years, is the X greater than Y? | 57/105 | In how many years, is the population in the largest city greater than 26 %? |
| What is the difference between ...? | 31/105 | What is the difference between two consecutive major ticks on the Y-axis? |
| Others | 17/105 | What is the average rating of statistical capacity in Bolivia per year? |

Table 10: Some source datasets of MathVista featuring similar questions. The table shows the template, proportion within the whole dataset and typical examples of them.

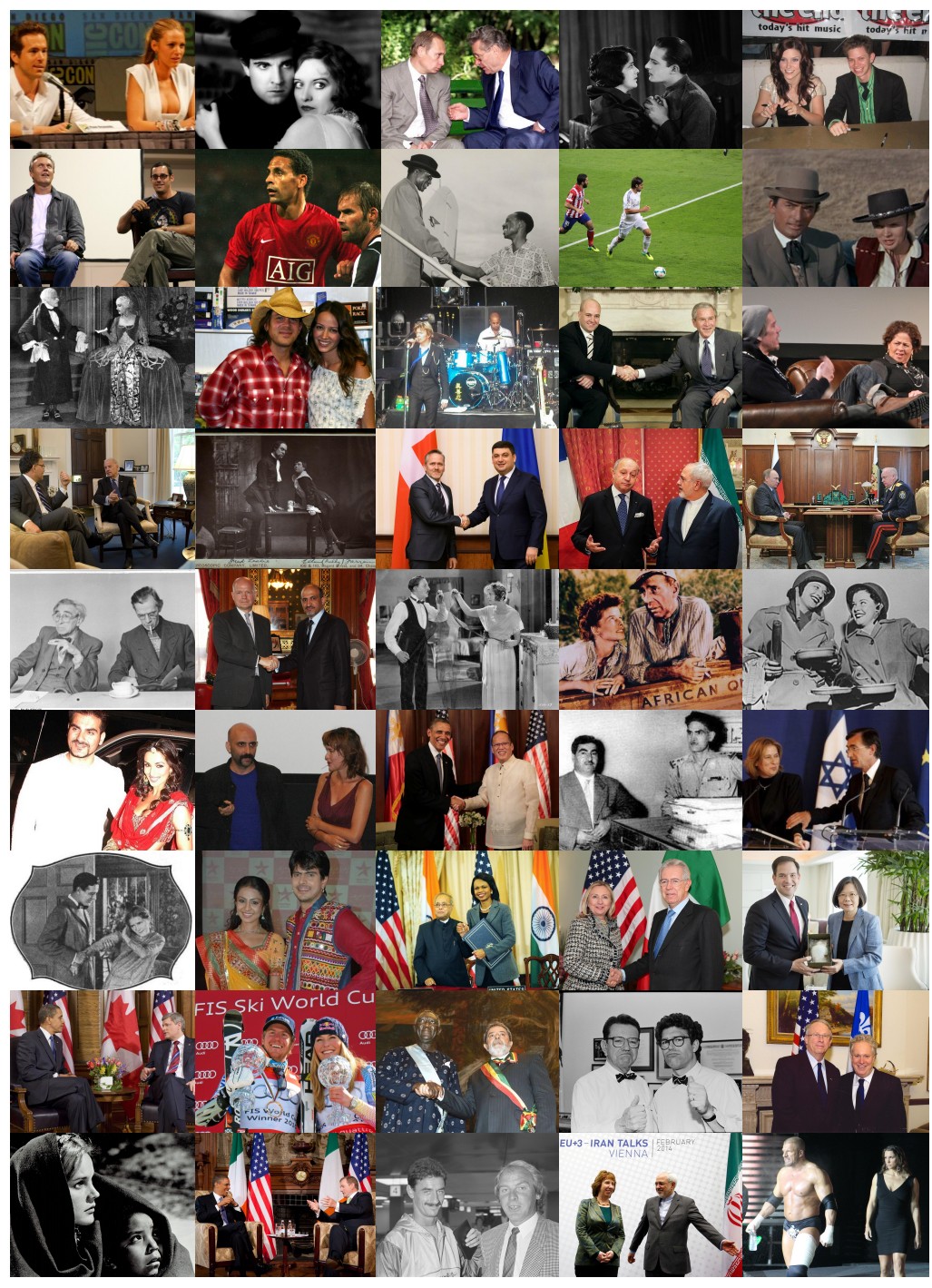

Figure 7: Some images with the question "What is the age gap between these two people in the image?" from MathVista. Questions containing "What is the age gap between" appear 310 times.

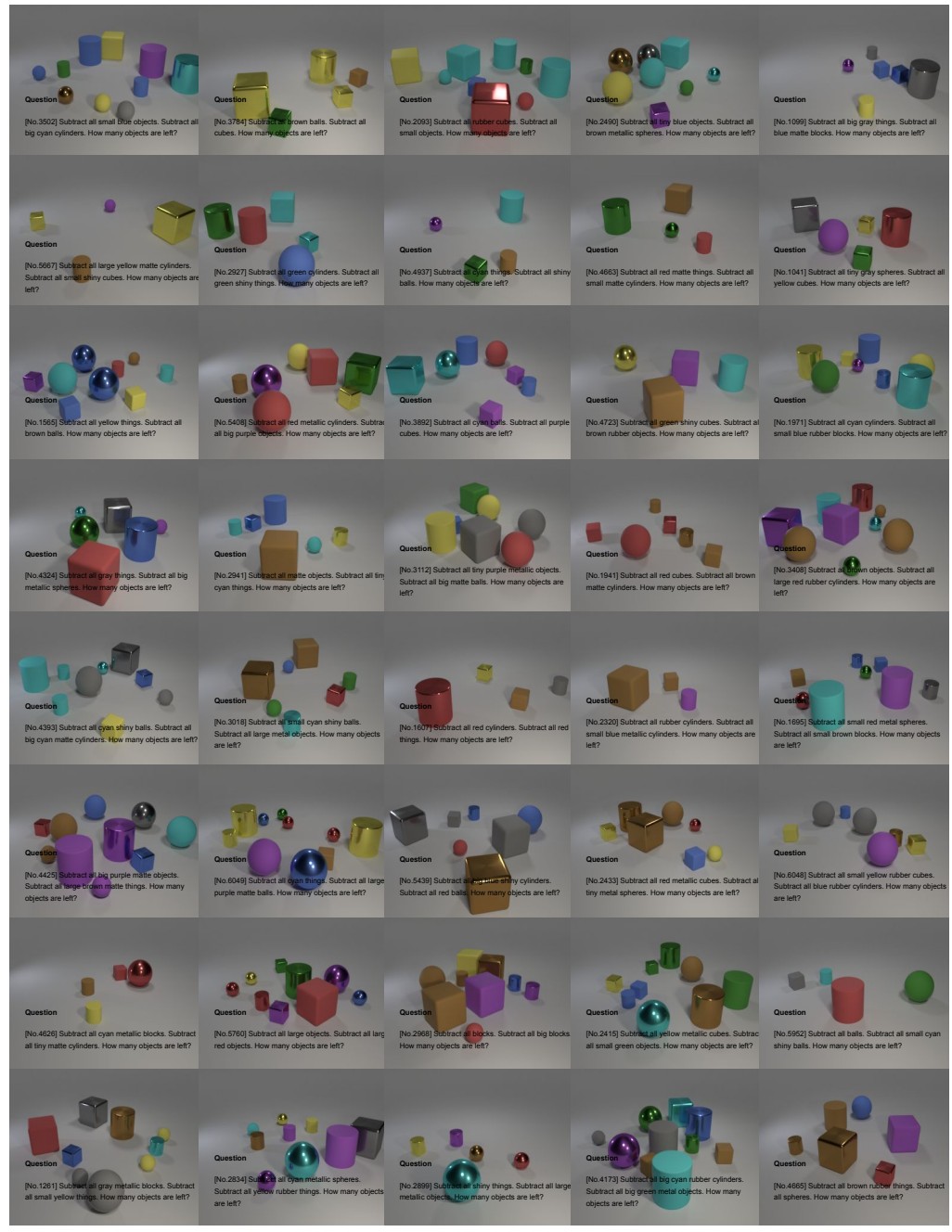

Figure 8: Problem involving very similar geometry recognition from MathVista. Questions like "Subtract all ... objects. How many objects are left?" appear 400 times.

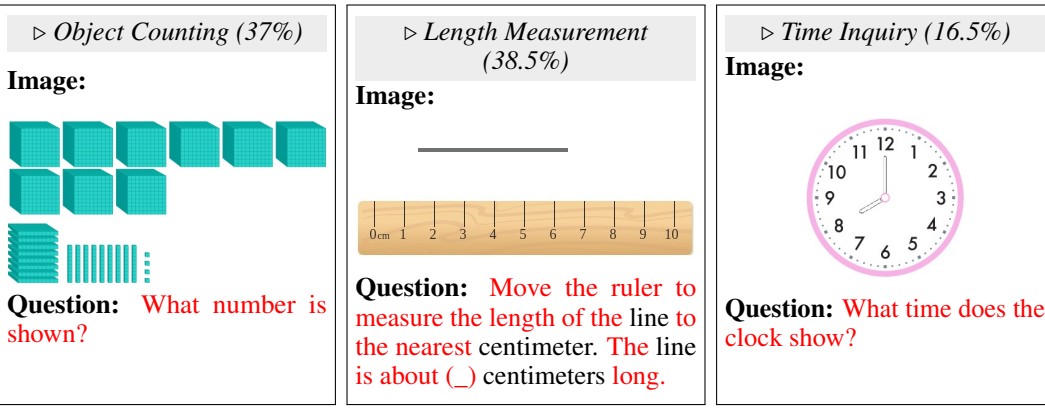

Figure 9: Three main types of questions among the 375 questions with an abstract scene in MathVista: Object Counting, Length Measurement, and Time Inquiry. Among the 200 sampled questions, the percentage of each category is in the figure. Text marked in red indicates that it belongs to a fixed template and is used in other similar questions.

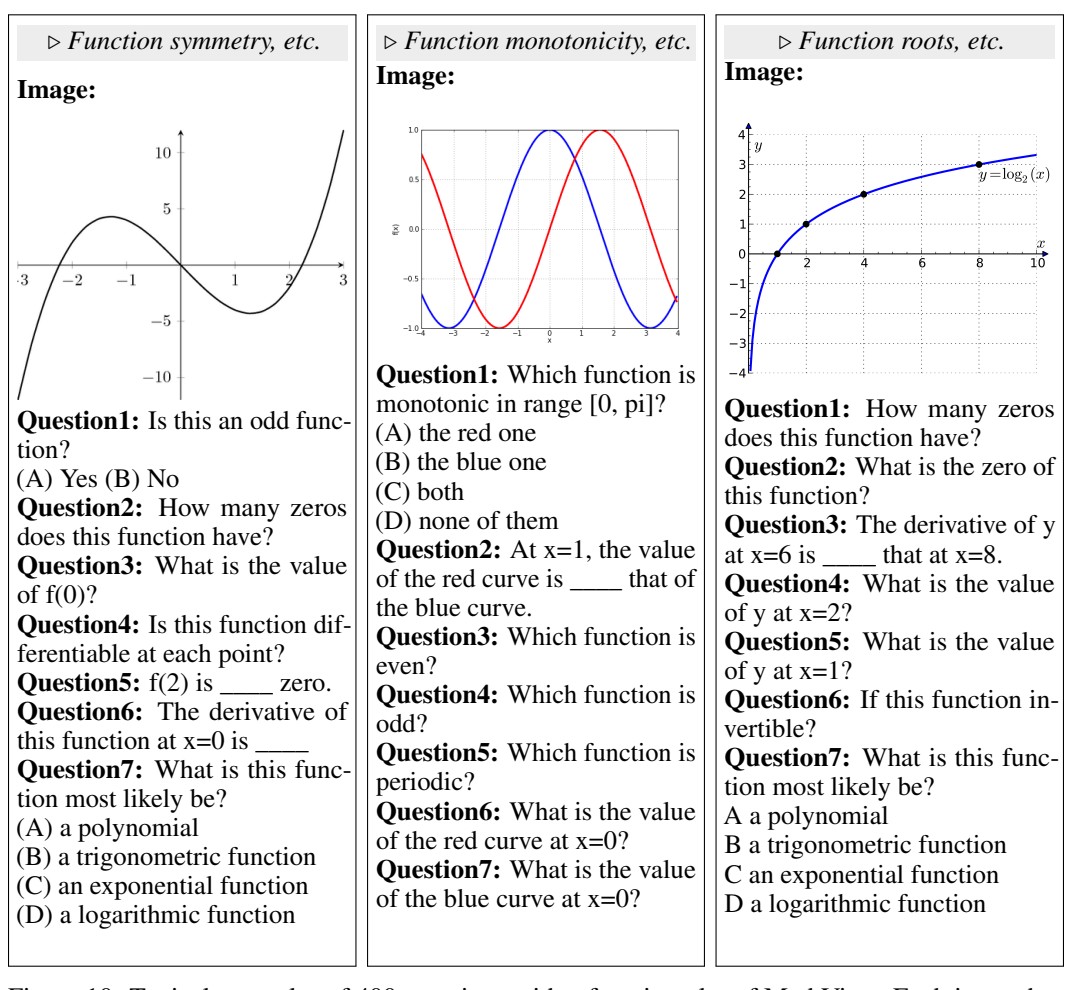

Figure 10: Typical examples of 400 questions with a function plot of MathVista. Each image has approximately seven questions about function symmetry, monotonicity, roots, periodicity, values, etc.

| ▷ *mutual symmetry of functions* | ▷ *quadratic function discriminant* | ▷ *find roots of iterative functions* |
|---|---|---|
| **Image:** 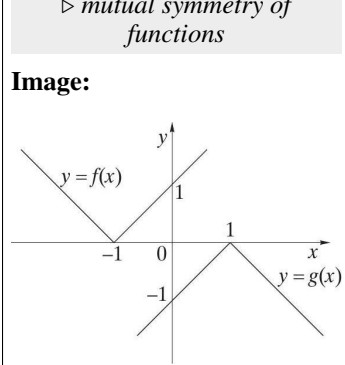 | **Image:** 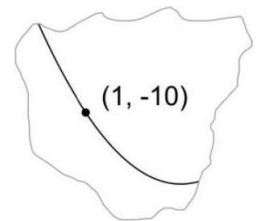 | **Image:** 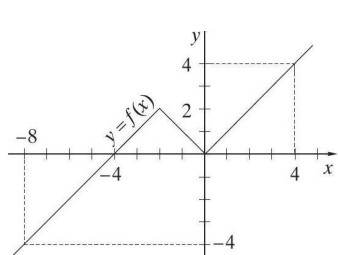 |
| **Question:** The figure shows graphs of functions $f$ and $g$ defined on real numbers. Each graph consists of two perpendicular halflines. Which is satisfied for every real number $x$? 
 (A) $f(x) = -g(x) + 2$ 
 (B) $f(x) = -g(x) - 2$ 
 (C) $f(x) = -g(x + 2)$ 
 (D) $f(x + 2) = -g(x)$ 
 (E) $f(x + 1) = -g(x - 1)$ | **Question:** In the (x,y)-plane the coordinate axes are positioned as usual. Point $A(1, -10)$ which is on the parabola $y = ax^2 + bx + c$ was marked. Afterwards the coordinate axis and the majority of the parabola were deleted. Which of the following statements could be false? 
 (A) $a > 0$ (B) $b < 0$ ... | **Question:** The graph of the function $f(x)$, defined for all real numbers, is formed by two half-lines and one segment, as illustrated in the picture. Clearly, -8 is a solution of the equation $f(f(x)) = 0$, because $f(f(-8)) = f(-4) = 0$. Find all the solutions of the equation $f(f(f(x))) = 0$. |

Figure 11: Three examples from `MATH-V` on analytic geometry, which are about mutual symmetry of functions, quadratic function discriminant, and roots of iterative functions. We omit some choices for brevity.

## H    More Dataset Details

### H.1    Question Distribution

All questions in `MATH-V` are written in English. As illustrated in Appendix B, the average number of words in a question is 42.3, with a maximum of 280 words and a minimum of 5 words. Figure 17 further elucidates the distribution of word counts, highlighting the diverse patterns of questions. We can see that the median word count and the average word count are close, indicating that the distribution is not skewed. The distribution of word counts is near a normal distribution, with more than 98.75% of questions having a word count of less than 100.

### H.2    Division of Levels

In this section, we detail the division of `MATH-V`'s difficulty levels. Our difficulty classification is divided into two steps. The first step involves roughly dividing the difficulty levels based on the grades of the contestants as specified in the competition. The second step entails manually checking and correcting any unreasonable divisions.

**Step 1.** The existing competition's grade requirements span grades 1 to 12. Because the questions for grades 1 and 2 are considered too simple, we classify them separately as level 1. The questions for grades 3, 4, 5, and 6 are similar in difficulty, so we classify them as level 2. Grades 7 and 8 are categorized as level 3, grades 9 and 10 as level 4, and grades 11 and above as level 5.

**Step 2.** We manually review the questions in each level and adjust the classification according to the actual difficulty of the questions. This step is necessary because, within the same exam, the difficulty range of the questions can be quite broad. For instance, the most difficult question in grade 4 might be more challenging than the easiest question in grade 5. In such cases, we make manual adjustments.

Figure 18.a shows a comparison of the level distribution between our `MATH-V` and the MATH dataset. We observe that both datasets have the fewest number of questions in level 1 and a similar number of

| ▷ *String Knot Probability Puzzle* | ▷ *Village Map Sketching Challenge* |

**String Knot Probability Puzzle**

**Image:**

`<image1>:`

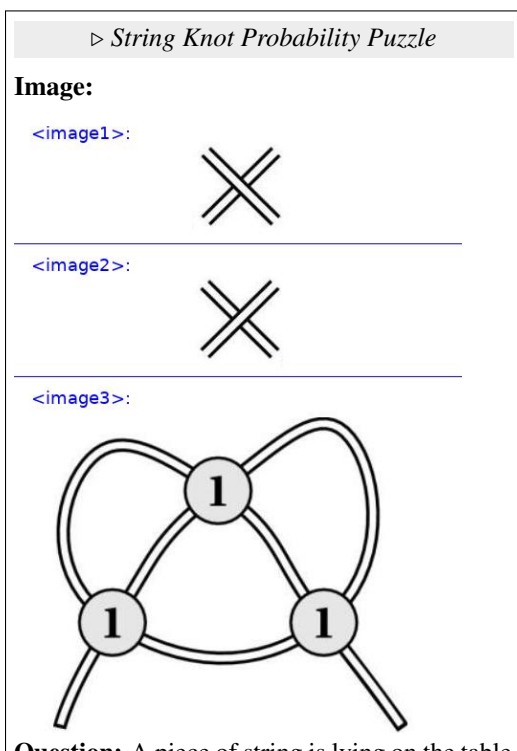

`<image2>:`

`<image3>:`

**Question:** A piece of string is lying on the table. It is partially covered by three coins as seen in the figure. Under each coin the string is equally likely to pass over itself like this: `<image1>` or like this: `<image2>`. What is the probability that the string is knotted after its ends are pulled?

**Village Map Sketching Challenge**

**Image:**

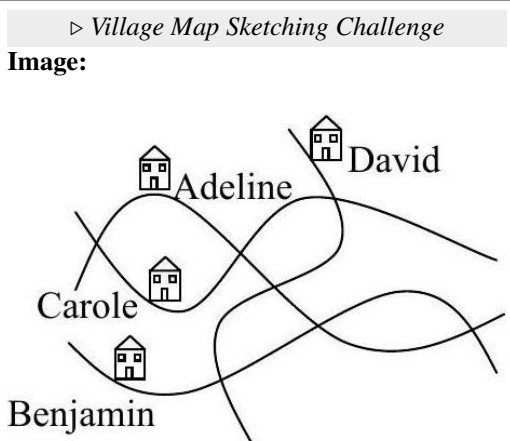

**Question:** During a rough sailing trip, Jacques tried to sketch a map of his village. He managed to draw the four streets, the seven places where they cross and the houses of his friends. The houses are marked on the correct streets, and the intersections are correct, however, in reality, Arrow Street, Nail Street and Ruler Street are all absolutely straight. The fourth street is Curvy Street. Who lives on Curvy Street?
(A) Adeline
(B) Benjamin
(C) Carole
(D) David
(E) It is impossible to tell without a better map

**▷ Borromaic Rings**

**Image:**

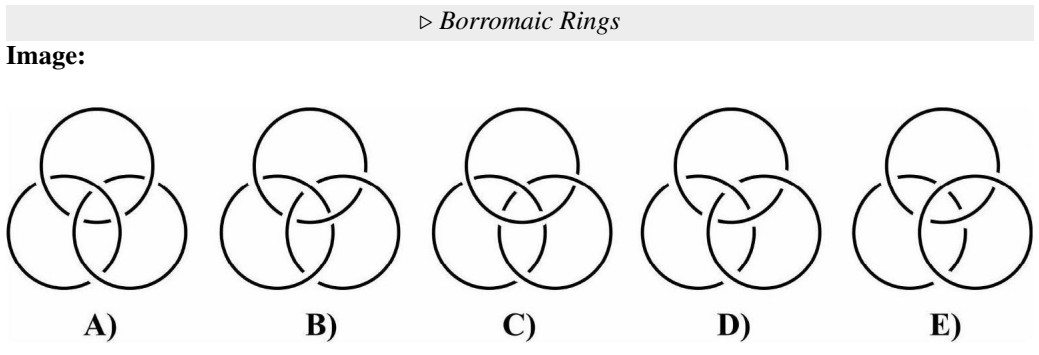

**Question:** The "Borromaic Rings" have an extraordinary property. Although no two are interlocked, they are strongly connected within each other. If one ring is cut through, the other two fall apart. Which of the following diagrams shows the picture of "Borromaic Rings"?

Figure 12: Three examples from `MATH-V` on topology. We omit some choices for brevity.

| ▷ *Adjacent Coloring Puzzle* | ▷ *Coloring Options* | ▷ *Optimal Line Additions* |
|---|---|---|
| **Image:** | **Image:** | **Image:** |
| 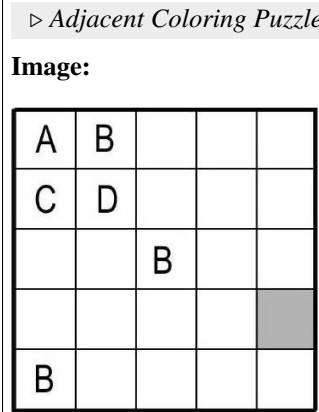 | 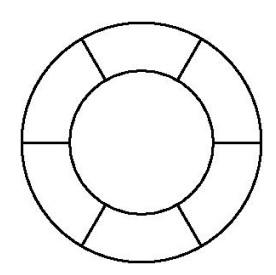 | 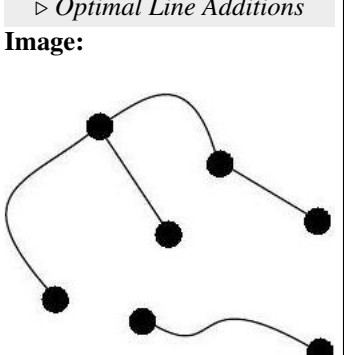 |
| **Question:** In the diagram we want to colour the fields with the colours A, B, C D so that adjacent fields are always in different colours. (Even fields that share only one corner, count as adjacent.) Some fields have already been coloured in. In which colour can the grey field be coloured in? | **Question:** The figure below shows a ring made of six small sections which you are to paint on a wall. You have four paint colors available and will paint each of the six sections a solid color. Find the number of ways you can choose to paint each of the six sections if no two adjacent section can be painted with the same color. | **Question:** The picture shows seven points and the connections between them. What is the least number of connecting lines that could be added to the picture so that each of the seven points has the same number of connections with other points? (Connecting lines are allowed to cross each other.) |

Figure 13: Three examples from `MATH-V` on graph theory. We omit some choices for brevity.

| ▷ *Kruskal's Algorithm* | ▷ *Prüfer Code* | ▷ *DFS Spanning Tree* |
|---|---|---|
| **Image:** | **Image:** | **Image:** |
| 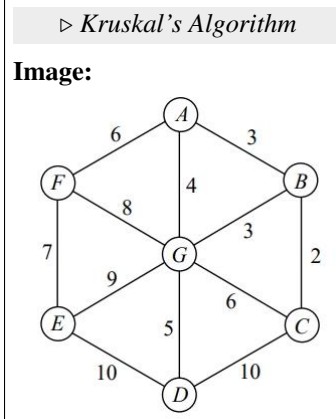 | 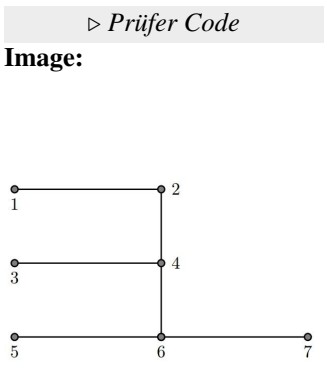 | 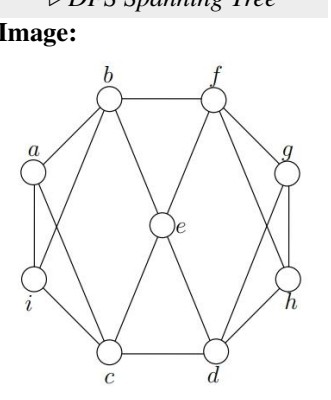 |
| **Question:** Using Kruskal's algorithm find a minimal spanning tree of <image 1>. Show the list of chosen edges only in alphabet order. Is 'AB, AF, BC, BG, DG, EF' the correct answer? 
 (A) Yes 
 (B) No | **Question:** Give the Prüfer code of <image 1> 
 (A) 2,4,4,6,6 
 (B) 2,3,3,6,6 
 (C) 2,2,4,6,6 
 (D) 2,4,4,6,7 | **Question:** Apply DFS Spanning Tree Algorithm to <image 1> with vertex pre-ordering a, b, c, e, i, h, g, d, f. Show the list of chosen edges only in alphabet order. 
 (A) ab,be,ci,cd,ce,dh,fg,gh 
 (B) ab,ai,be,bf,cd,ci,fg,fh |

Figure 14: Three examples from the Graph Theory subfield in the Math subset of MMMU.

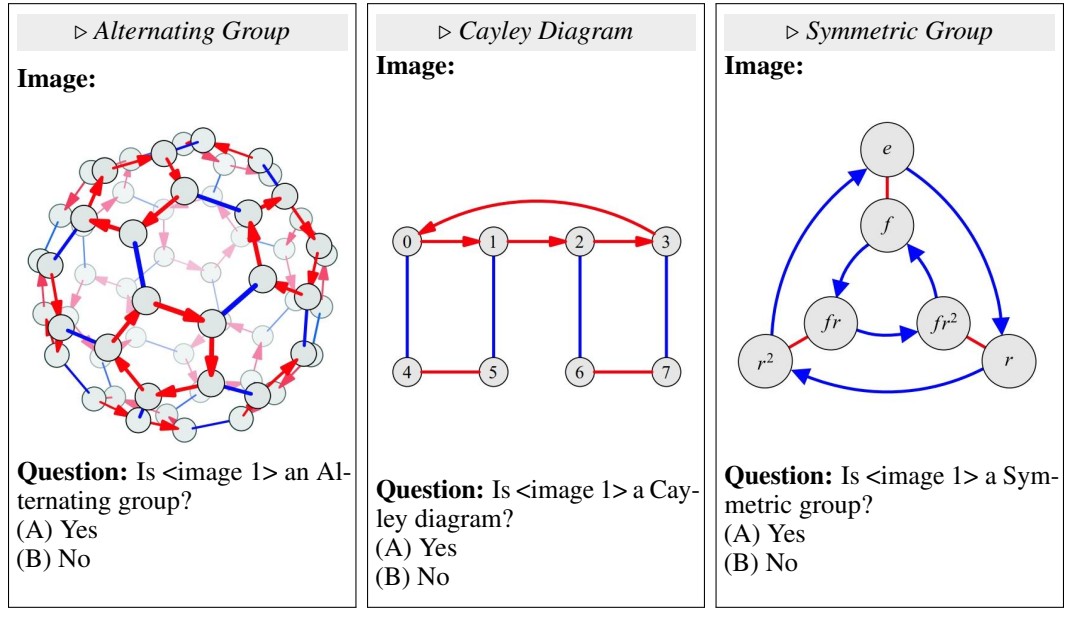

Figure 15: Three examples from the Group Theory subfield in the Math subset of MMMU.

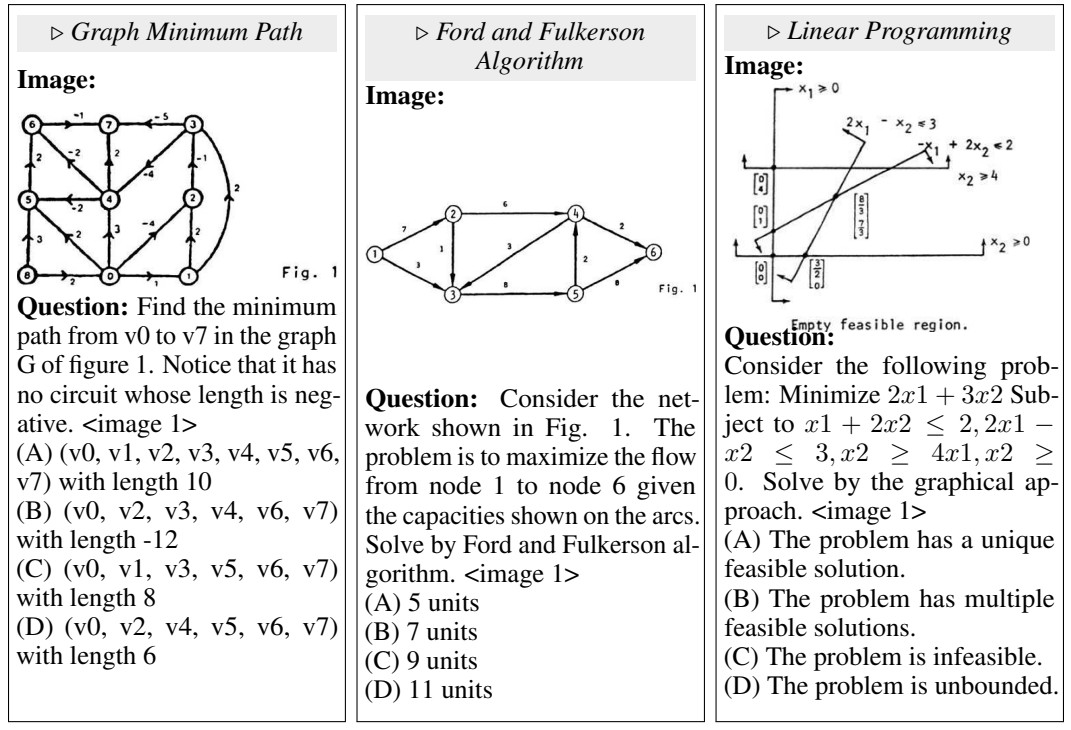

Figure 16: Three examples from the Operation Research subfield in the Math subset of MMMU.

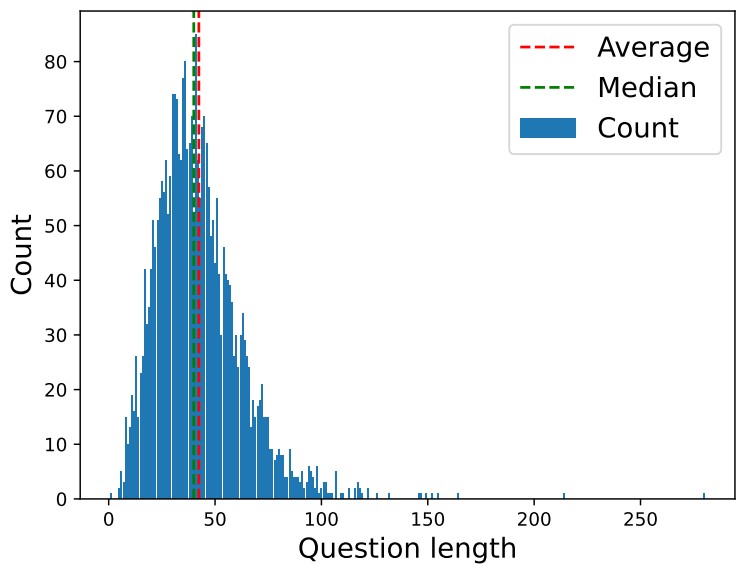

Figure 17: The distribution of the number of words per question in MATH-V.

questions in the other levels. Figure 18.b illustrates the models' accuracy across different levels. It is evident that the accuracy of the models decreases as the difficulty level increases, which aligns with the expectation that the difficulty of the questions escalates with the level.

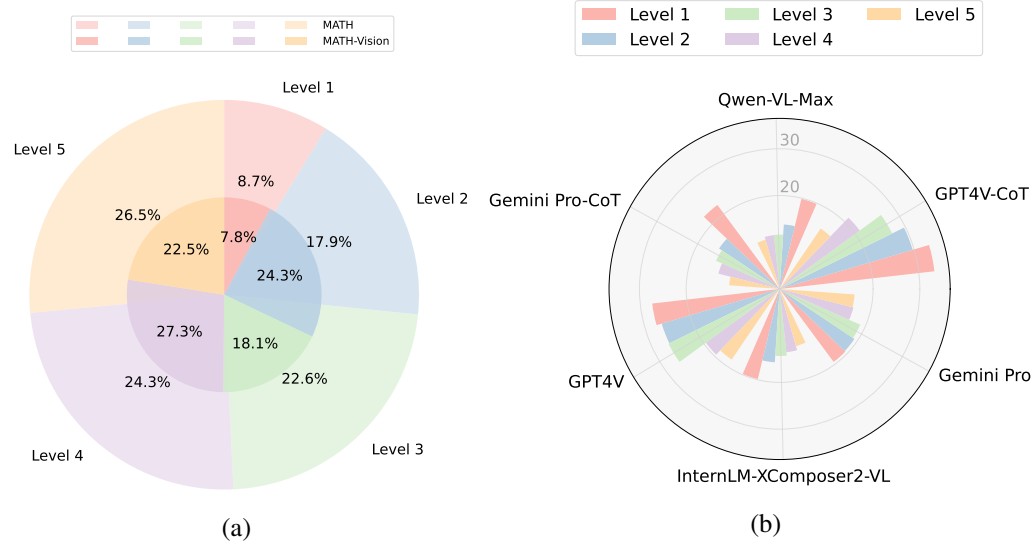

(a)                                                    (b)

Figure 18: (a) Comparison of the level distribution between our MATH-V and the MATH dataset [33]. (b) Accuracy on MATH-V on six typical LMMs across 5 difficulty levels.

## H.3 Images by Subjects

In this section, we demonstrate some images from the problems in our dataset. Figure 20, Figure 21, Figure 22, Figure 23, Figure 24, Figure 25, Figure 26, Figure 27, Figure 28, Figure 29, Figure 30, Figure 31, Figure 32, Figure 33, Figure 34 and Figure 35 show images from the problems under the category of Algebra, Analytic Geometry, Arithmetic, Combinatorial Geometry, Combinatorics, Counting, Descriptive Geometry, Graph Theory, Logic, Metric Geometry - Angle, Metric Geometry -

Area, Metric Geometry - Length, Solid Geometry, Statistics, Topology and Transformation Geometry, respectively.

We can see that the images are diverse among different categories, and the images in the same category have similar visual patterns. For example, the images in the category of Statistics are mainly various types of charts and tables, while the images in the category of Arithmetic and Counting, which are categories mainly for primary school students, are more colorful and have more cartoon characters or real-world objects, and, conversely, the images in the category of Analytic Geometry are more complex and abstract.

## H.4   Introduction of Subjects

**Algebra.** Algebra is a branch of mathematics that deals with symbols and the rules for manipulating these symbols. It is a unifying thread of almost all of mathematics and includes everything from solving simple equations to studying abstractions such as groups, rings, and fields. Algebra allows us to express relationships and solve problems where we don't initially know the values of certain quantities. It serves as the foundation for more advanced topics in mathematics and sciences, enabling the formulation and solving of equations and inequalities.

**Analytic Geometry.** Analytic Geometry, also known as coordinate geometry, merges algebra and geometry to discuss shapes and figures using a coordinate system. This branch of mathematics allows the representation of geometric shapes in a numerical way by translating them into equations and vice versa. It enables the study of geometric properties and relationships by means of algebraic equations, thus providing a powerful tool for solving geometric problems.

**Arithmetic.** Arithmetic is the most basic branch of mathematics, dealing with the study of numbers and the traditional operations on them: addition, subtraction, multiplication, and division. It forms the foundation upon which more advanced mathematical concepts are built. Arithmetic is used in everyday life for counting, measuring, and performing calculations. It is the first type of math that students are introduced to and remains fundamental for understanding more complex areas of mathematics.

**Combinatorial Geometry.** Combinatorial Geometry is a field of mathematics that studies geometric objects and their arrangement, with a particular focus on discrete and combinatorial aspects. It involves the investigation of geometric structures and their properties under various combinatorial processes, such as counting, arrangement, and partitioning. This branch often intersects with areas like graph theory and combinatorics, exploring problems related to the configuration of points, lines, and other geometric figures.

**Combinatorics.** Combinatorics is a branch of mathematics concerned with counting, arrangement, and combination of objects. It deals with questions of how many different ways such objects can be chosen, arranged, and combined under certain conditions. Combinatorics has applications in numerous areas, including computer science, optimization, and probability theory. It is fundamental in developing analytical skills and understanding complex mathematical concepts.

**Counting.** Counting is the most basic and fundamental concept in mathematics, involving the enumeration of objects in a set. It serves as the foundation for more advanced mathematical concepts, such as arithmetic and combinatorics. Counting is not only about determining the total number of items in a collection but also includes understanding and applying principles like one-to-one correspondence, the counting principle, permutations, and combinations.

**Descriptive Geometry.** Descriptive Geometry is the branch of geometry that allows the representation of three-dimensional objects in two dimensions, through the use of projections or perspective drawings. It is a method of visually communicating complex geometric shapes and is fundamental in fields such as engineering, architecture, and design. Descriptive geometry provides tools and techniques for solving spatial problems and for understanding the spatial relationships between objects.

**Graph Theory.** Graph Theory is a branch of mathematics that studies graphs, which are mathematical structures used to model pairwise relations between objects. A graph is made up of vertices (or nodes) and edges (or arcs) that connect them. Graph theory is used to study networks and their properties, such as paths, cycles, and connectivity, and has applications in computer science, biology, transportation, social sciences, and more.

**Logic.** Logic is the foundation of mathematical reasoning, focusing on the study of argument validity, structure, and form. It involves the analysis of propositions and their relationships to determine the truth values of statements. Logic is used to construct and analyze mathematical proofs, ensuring that conclusions follow logically from premises. It encompasses various systems, such as propositional logic, predicate logic, and modal logic, each with its own rules and applications.

**Metric Geometry - Angle.** Metric Geometry concerning angles focuses on the properties and measurements of angles within various geometric figures. It involves understanding the types of angles (acute, right, obtuse, and straight), their relationships, and how they can be measured and used to determine other geometric properties. This aspect of metric geometry is crucial in solving problems related to shapes, patterns, and designs in both two and three dimensions.

**Metric Geometry - Area.** Metric Geometry in the context of area deals with the measurement of the surface enclosed within geometric figures. It involves calculating the extent of two-dimensional shapes, such as squares, rectangles, triangles, and circles, using various formulas. Understanding area is essential in numerous practical applications, including architecture, land surveying, and crafting, where the determination of space is necessary.

**Metric Geometry - Length.** Metric Geometry concerning length involves the measurement of distance between points in geometric figures. It is the foundation for determining the perimeter of shapes, the distance around a figure, and the concept of metric spaces in more advanced mathematics. Length measurements are fundamental in various real-world applications, such as construction, fabrication, and navigation.

**Solid Geometry.** Solid Geometry is the branch of mathematics that deals with the study of three-dimensional figures, including spheres, cubes, cylinders, and pyramids, among others. It involves understanding the properties, measurements, and relationships of these figures, such as volume, surface area, and the angles between intersecting planes. Solid geometry is essential in fields such as engineering, architecture, and physics, where three-dimensional models are crucial.

**Statistics.** Statistics is the branch of mathematics that deals with collecting, analyzing, interpreting, and presenting data. It involves the use of theories of probability to make inferences about populations from samples. Statistics is applied in a wide range of disciplines, including economics, social sciences, biology, and engineering, to make decisions or predictions based on data analysis.

**Topology.** Topology is a branch of mathematics concerned with the properties of space that are preserved under continuous transformations, such as stretching and bending, but not tearing or gluing. It is often referred to as "rubber-sheet geometry" because of its focus on the intrinsic properties of spaces. Topology has applications in various areas, including quantum physics, computer science, and biology, and deals with concepts like continuity, compactness, and connectedness.

**Transformation Geometry.** Transformation Geometry involves the study of geometric figures and the changes they undergo through transformations such as translation, rotation, reflection, and dilation. It focuses on understanding how these transformations affect the properties and relations of shapes and figures. Transformation geometry is fundamental in understanding the symmetry, congruence, and similarity of geometric objects, and it has applications in areas such as computer graphics, robotics, and pattern recognition.

# I  Error Examples of GPT-4V

In this section, we present detailed examples of errors made by GPT-4V. We categorize these errors into six types: reasoning error, vision recognition error, knowledge error, question misunderstood error, refusal to answer, and calculation error.

## I.1  Reasoning Error (42.2%)

Reasoning errors occur when GPT-4V fails to apply logical steps correctly. In Figure 36, the model incorrectly substitutes an unknown variable with a known value, leading to an incorrect conclusion. Figure 37 shows the model making an illogical hypothesis about the relationships of train wagon numbers.

### I.2 Vision Recognition Error (31.9%)

Vision recognition errors highlight the model's struggles with interpreting visual data. Figure 38 depicts an instance where GPT-4V fails to read information from a simple bar chart. In Figure 39, the error occurs because the model mistakenly believes that 3, 6, and y are on the same side of the triangle. Figure 40 shows the model experiencing severe hallucinations, as there is no balance in the image at all.

### I.3 Knowledge Error (15.1%)

Knowledge errors are instances where GPT-4V lacks the necessary information to answer a question and provides information that is outdated or incorrect. Figure 41 and Figure 42 both showcase situations where the model did not know how to solve the problems.

### I.4 Question Misunderstood Error (6.9%)

These errors occur when GPT-4V misinterprets the user's query. Figure 43 demonstrates a failure to grasp the query's intent, resulting in an incorrect answer. Similarly, in Figure 44, the model mistakenly believes that the key below corresponds to the lock above, even though the question does not provide this information.

### I.5 Reject to Answer (2.6%)

Sometimes, GPT-4V refuses to provide an answer, which is categorized as a Reject to answer error. Figure 45 and Figure 46 illustrate instances where the model either cannot generate a response or chooses not to, citing various reasons.

### I.6 Calculation Error (1.3%)

Calculation errors are straightforward mistakes in numerical or symbolic calculations. The model only makes errors when performing symbolic calculations in our analysis. Both Figure 47 and Figure 48 show a simple symbolic computational mistake, undermining the model's reliability in symbolic calculations.

## J Examples of GPT-4 with Image Captions

In this section, we provide some qualitative examples of GPT-4 with image captions, in comparison to other models like GPT-4V, Gemini Pro, and Qwen-VL-Max.

### J.1 Correct Examples

Some typical correct examples ( Figure 49, Figure 50, Figure 51, Figure 52, Figure 53, Figure 54, Figure 55) of GPT-4 with image captions on subjects like algebra, arithmetic, analytic geometry, counting, and combinatorics. Image captions are either accurate or have some small hallucinations.

### J.2 Wrong Examples

Some typical wrong examples (Figure 56, Figure 57, Figure 58, Figure 59, Figure 60, Figure 61) of GPT-4 with image captions on other subjects. Image captions are usually inaccurate to describe the image.

## K Details of Human Annotators

We hire human annotators for three tasks: 1. Verify the alignment of text questions and images; 2. Categorize the problems into 16 different subjects; 3. Evaluate human performance on MATH-V.

### K.1 Alignment of Text Questions and Images

We hire ten annotators in stage one, and all of them are senior college students, all majoring in science-related disciplines. We pay them $0.35 per question, and they were asked to judge whether the text of the question matched the corresponding image and whether the image was complete. When the image is incomplete or incorrectly positioned, they are asked to re-capture the image from the original file.

### K.2 Subjects Classification

For task two, we have 10 annotators and three authors, to categorize the problems into 16 different subjects independently in three groups with our specially designed annotation tool, as illustrated in Figure 19. We ask them to read our definitions of the 16 subjects in advance as shown in Appendix H.4, and will randomly check their classification of certain topics. If there are mistakes in classification, we will promptly correct them and inform them. We pay $0.45 per question. Notably, we also utilize existing LMMs (such as Gemini and GPT-4V) for subject classification. Detailed instructions to annotators and prompts to models can be found in Table 11. Finally, we will combine the classification results of the three groups with the classification results returned by LMMs to give the final classification.

---

▷ *Instructions for annotators on how to label questions into 16 categories*

We are constructing a dataset about mathematical problems in visual contexts, and we have collected quantities of questions that require categorization into different mathematical subjects.
These categorizations are logic, algebra, counting, arithmetic, combinatorics, graph theory, topology, statistics, combinatorial geometry, solid geometry, metric geometry - angle, metric geometry - area, metric geometry - length, analytic geometry, descriptive geometry, and transformation geometry.
Here are the detailed definitions of them: ... (definitions in Appendix H.4)
If you are unsure about how to categorize the problem into these subjects, choose 'pending' instead.

---

▷ *The prompt for models to get subject advice*

What branch of mathematics does the problem belong to? Choose from the following: logic, algebra, counting, arithmetic, combinatorics, graph theory, topology, statistics, combinatorial geometry, solid geometry, metric geometry - angle, metric geometry - area, metric geometry - length, analytic geometry, descriptive geometry, transformation geometry, pending.
You are not supposed to solve the problem. Wrap your final answer, a word or a short phrase, in "□".

---

Table 11: Instructions for annotators on how to label questions into 16 categories and the prompt for models to get subject advice.

### K.3 Human Performance

To obtain average human performance on our benchmark, we recruit 100 annotators possessing a high school degree or higher. These annotators had not previously participated in the competitions where our data originated and were seeing the questions in our test set for the first time. We divided the 3040 questions of MATH-V into 100 subsets, each containing 30 or 31 questions. Subsequently, we randomly allocated two subsets from this pool to each student. Participants were tasked with completing both subsets within a 70-minute time frame, spread across two separate days. Note that this time limit closely aligns with that of the Kangaroo competition, where participants are assigned a similar duration for a test containing 30 problems. Each participant receives a one-time payment of $40 as compensation for their contribution.

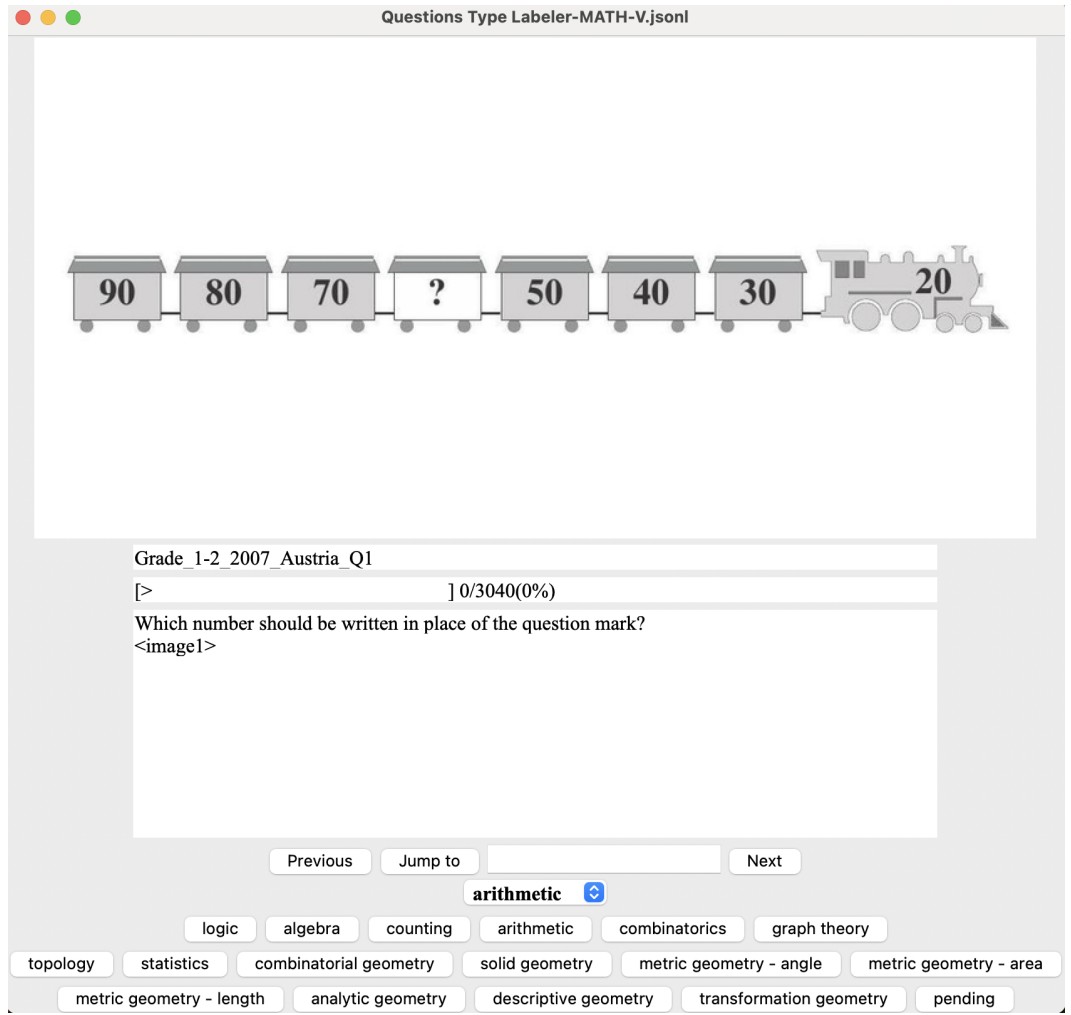

Figure 19: GUI for categorizing the problems into 16 different subjects.

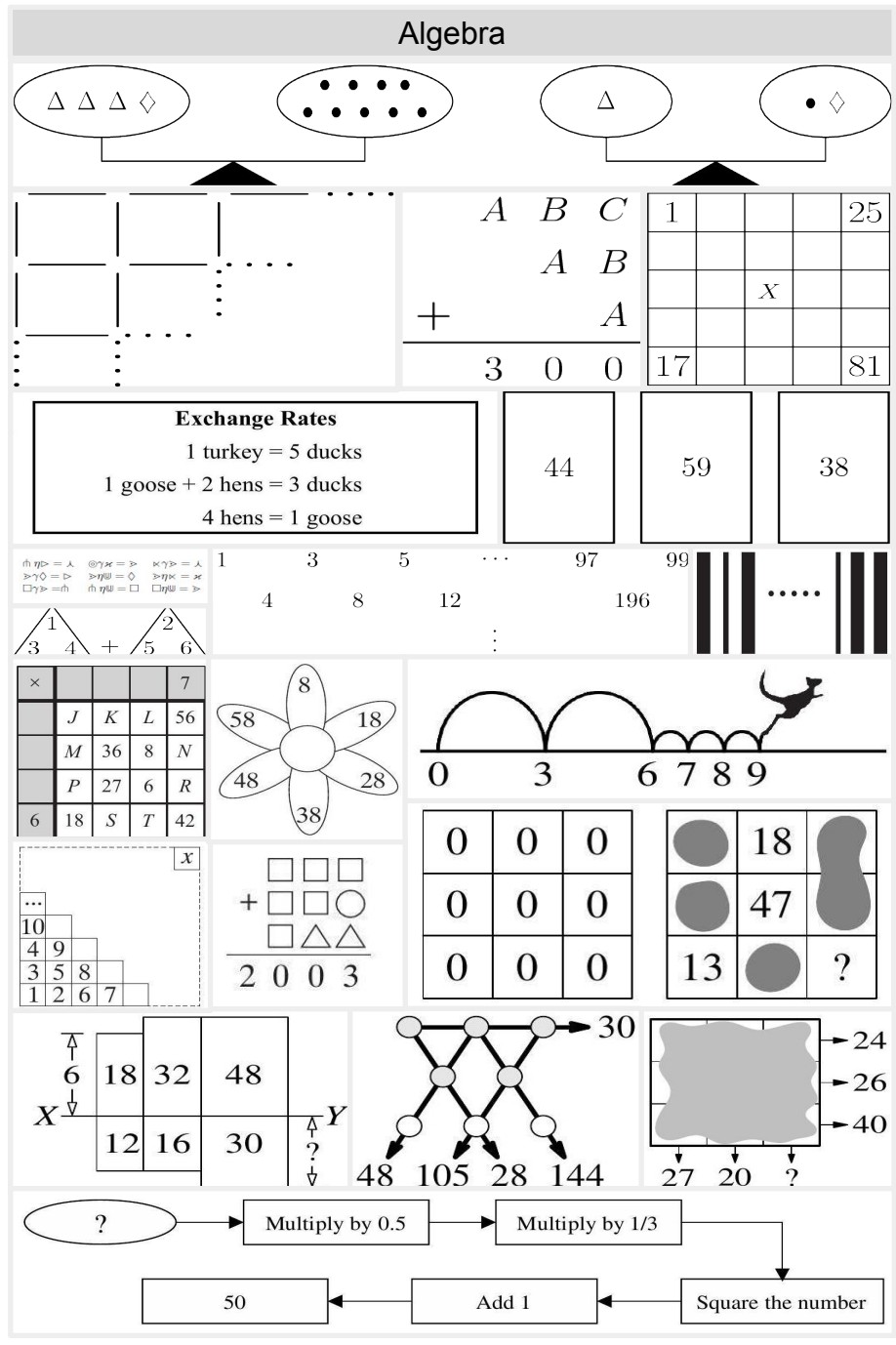

Figure 20: Some images from Algebra.

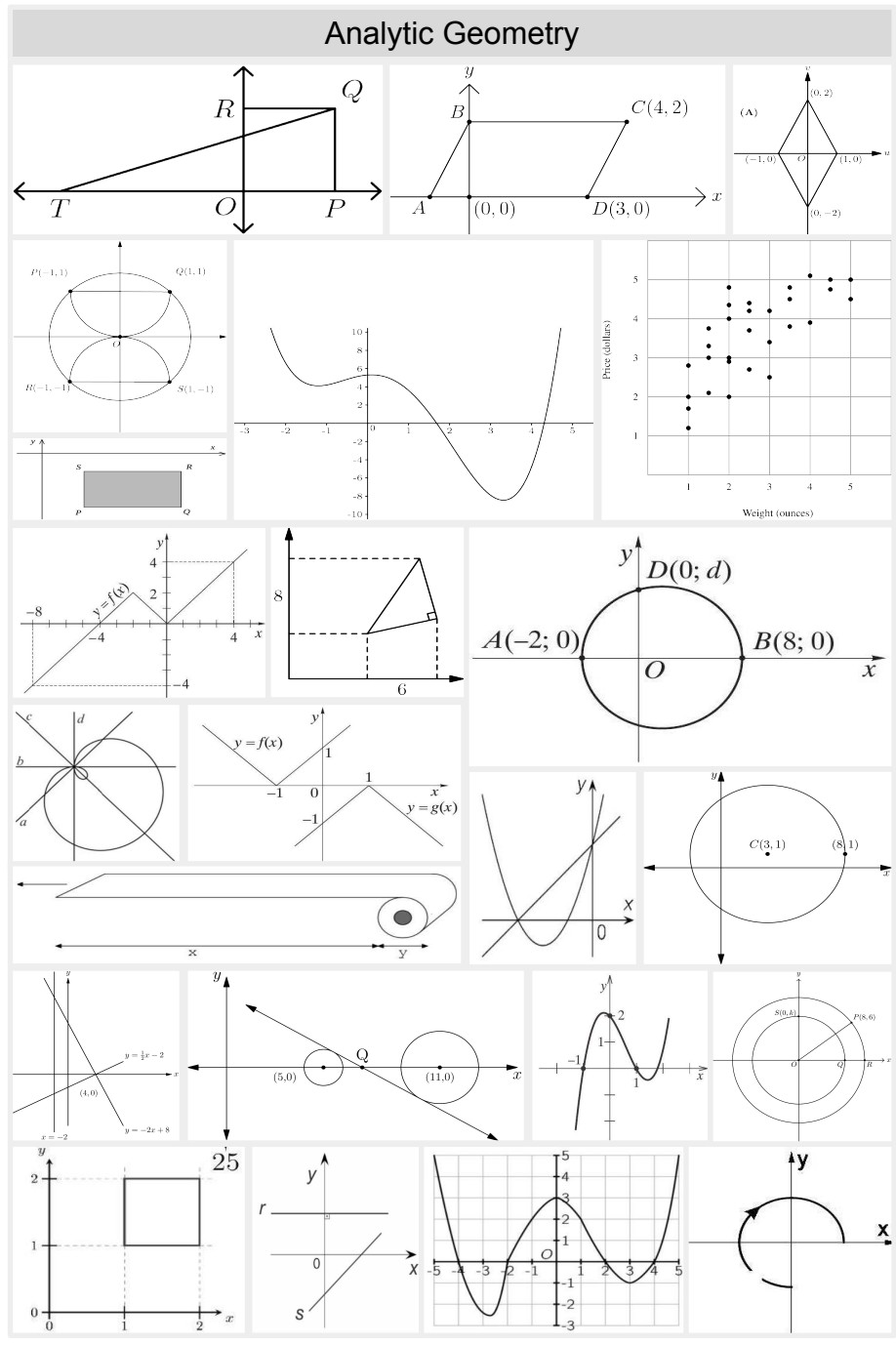

Figure 21: Some images from Analytic Geometry.

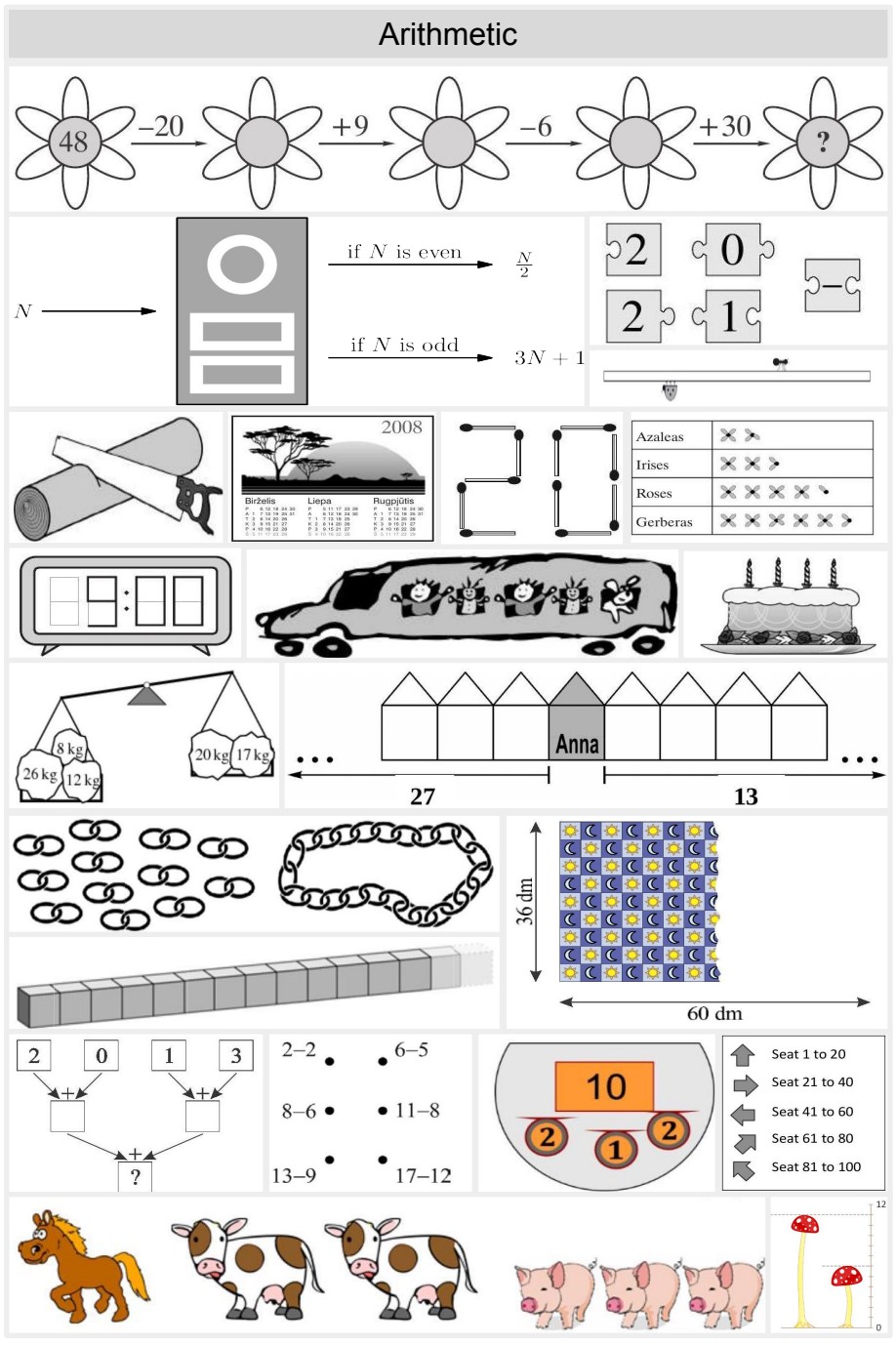

Figure 22: Some images from Arithmetic.

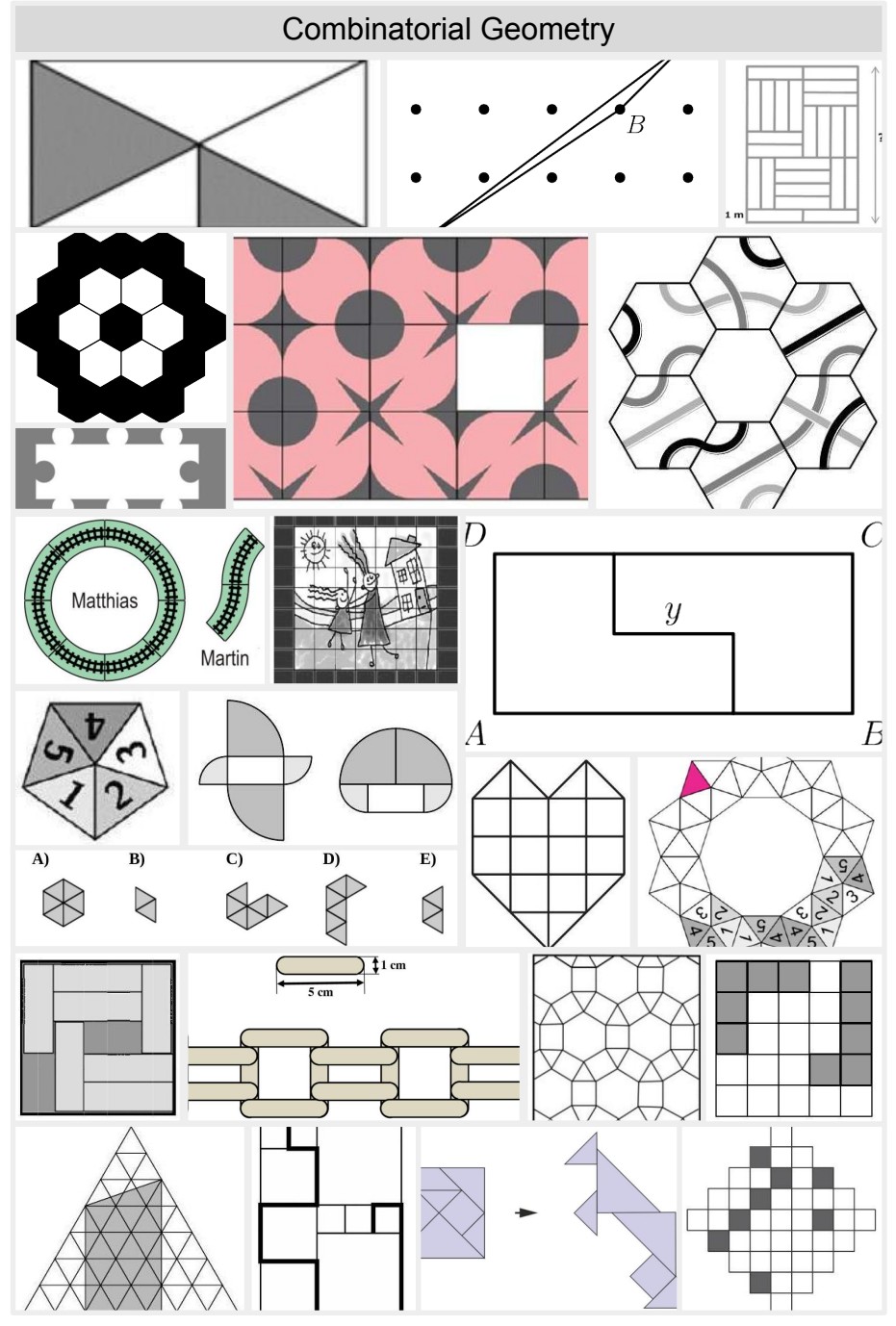

Figure 23: Some images from Combinatorial Geometry.

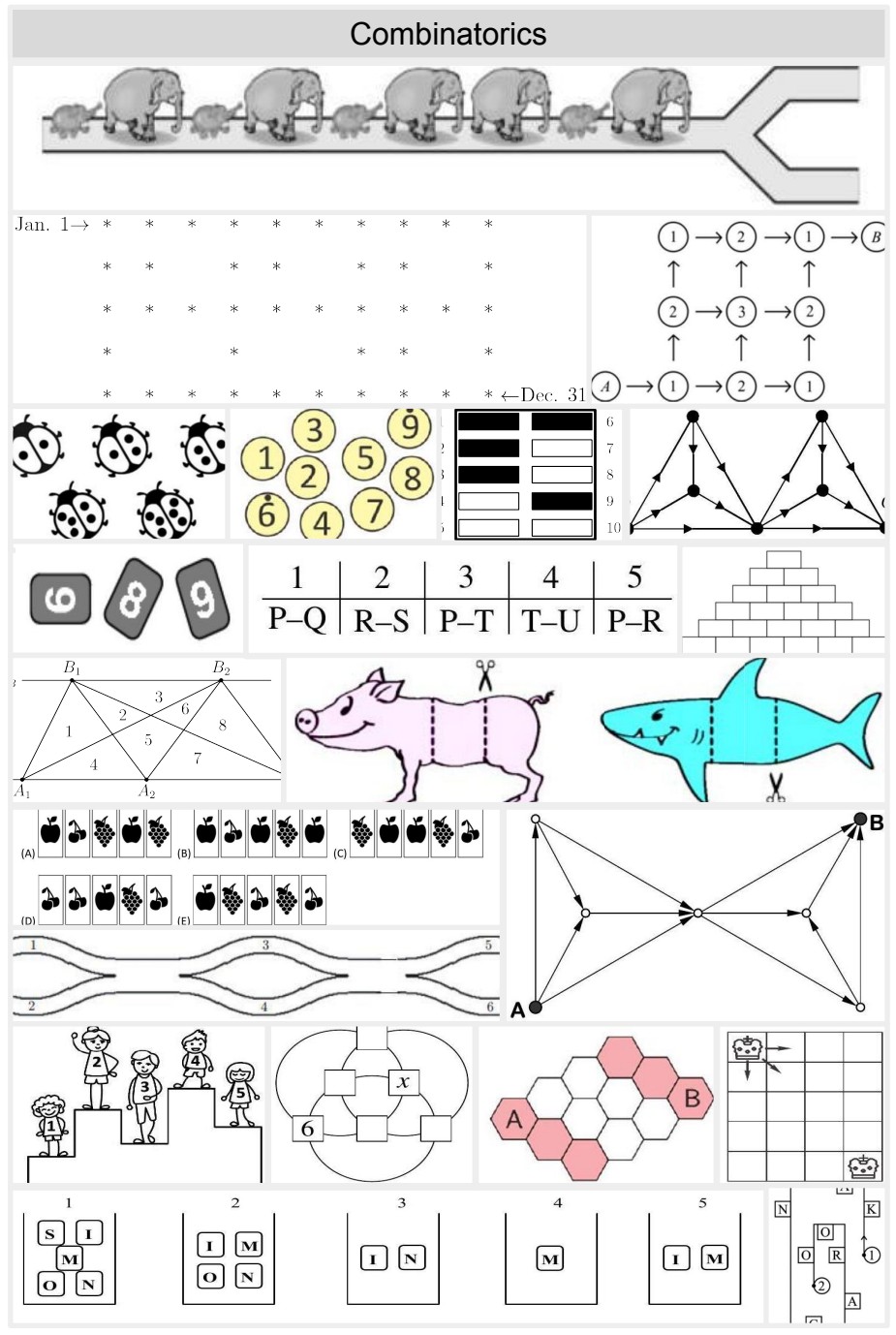

Figure 24: Some images from Combinatorics.

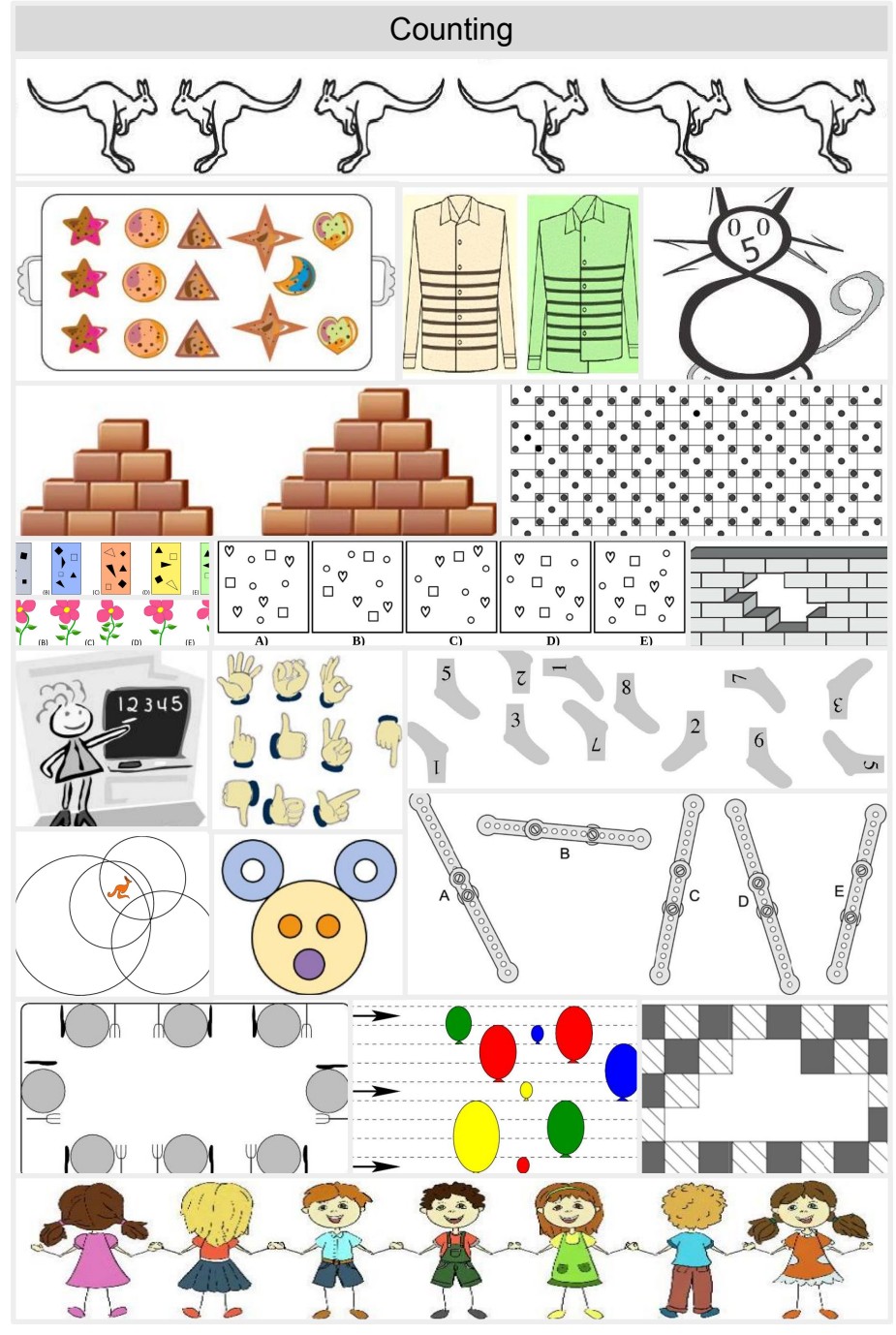

Figure 25: Some images from Counting.

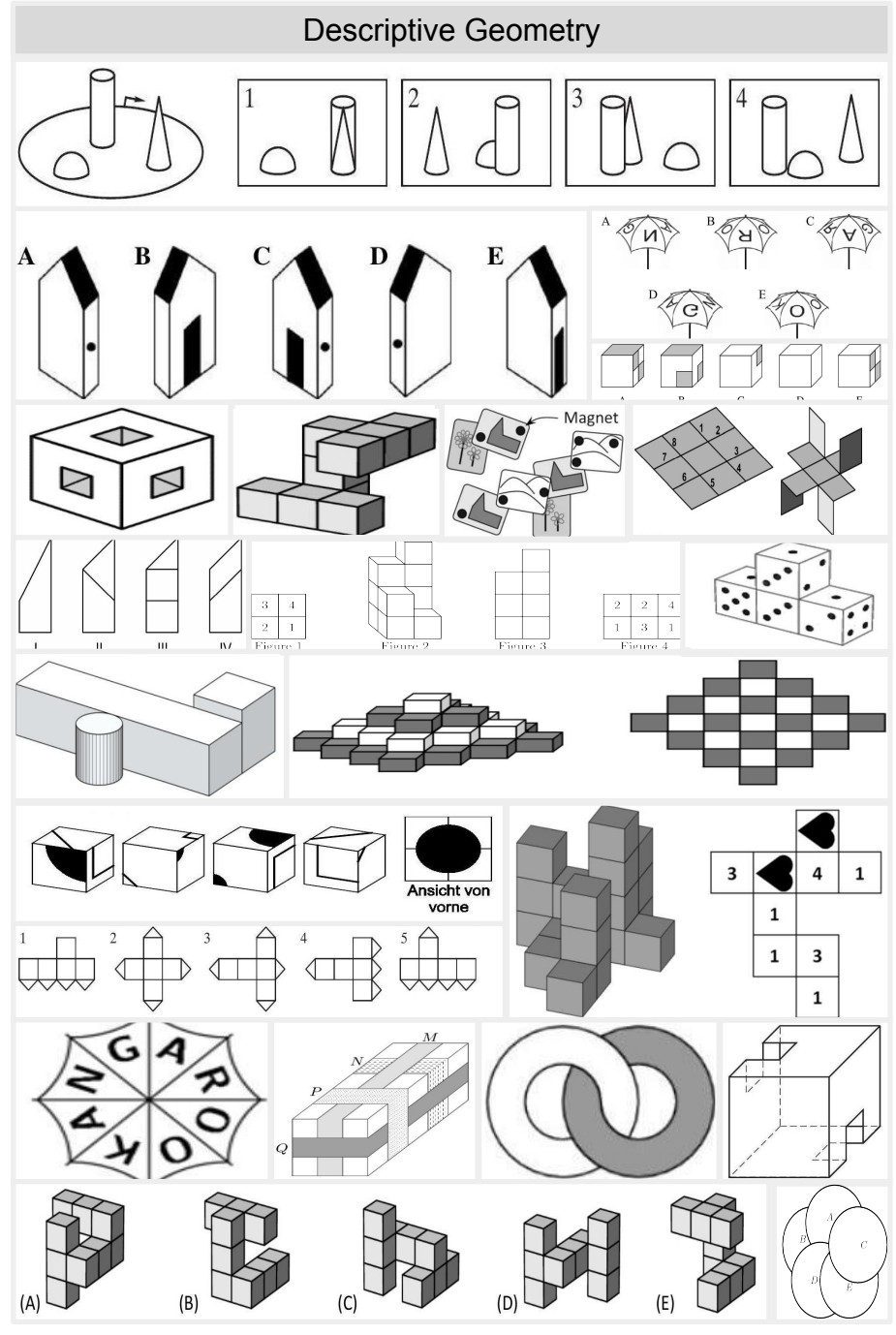

Figure 26: Some images from Descriptive Geometry.

# Graph Theory

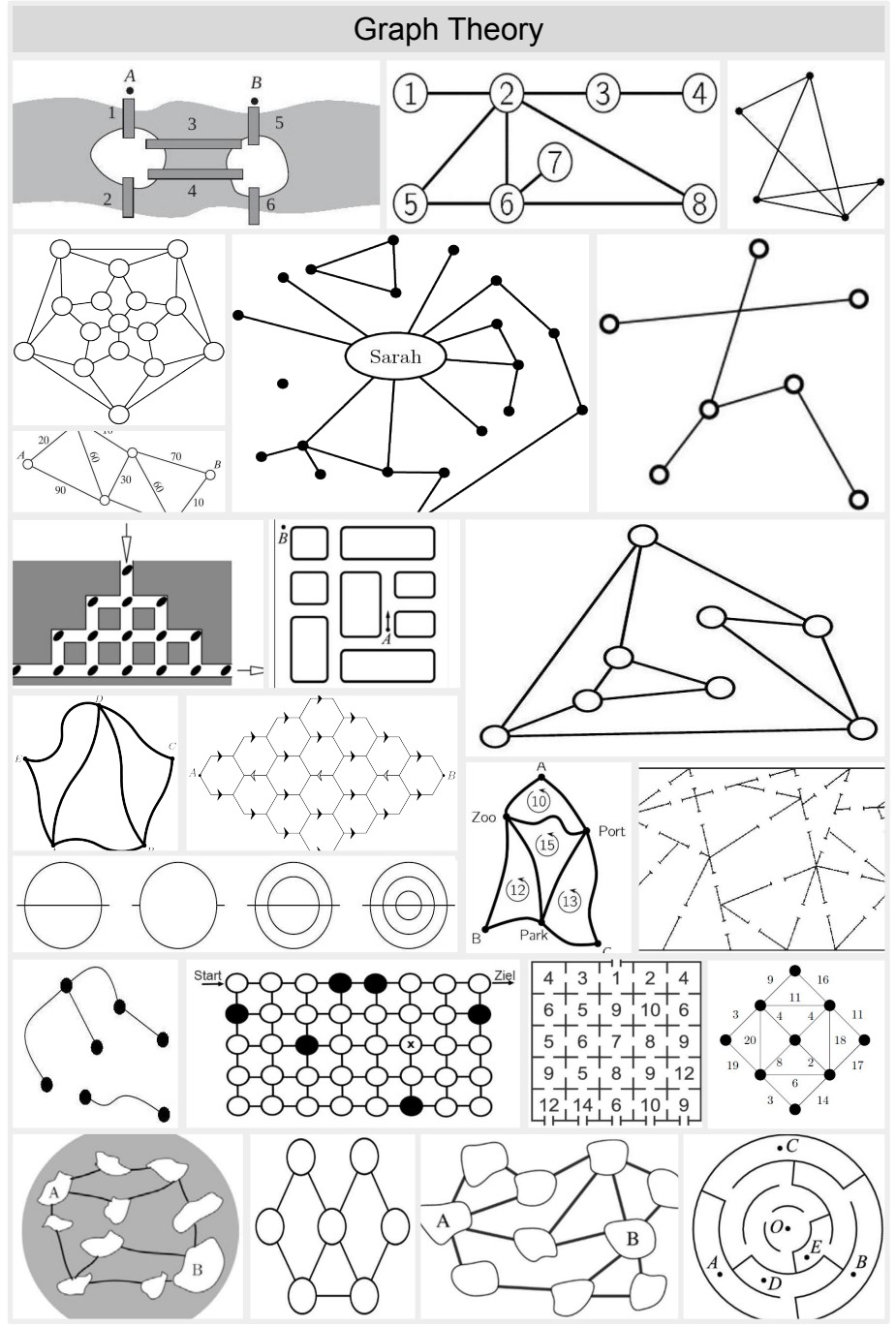

Figure 27: Some images from Graph Theory.

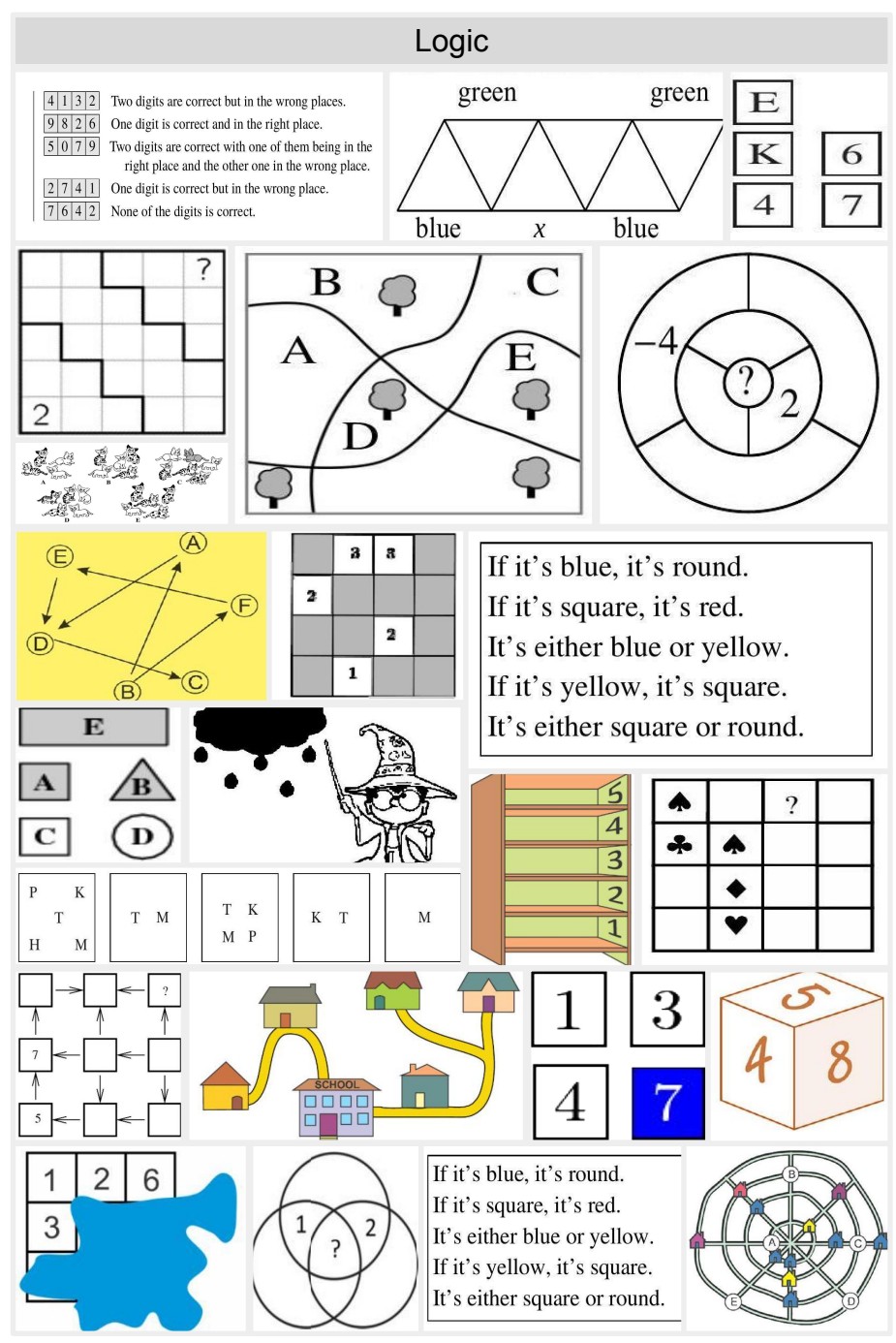

Figure 28: Some images from Logic.

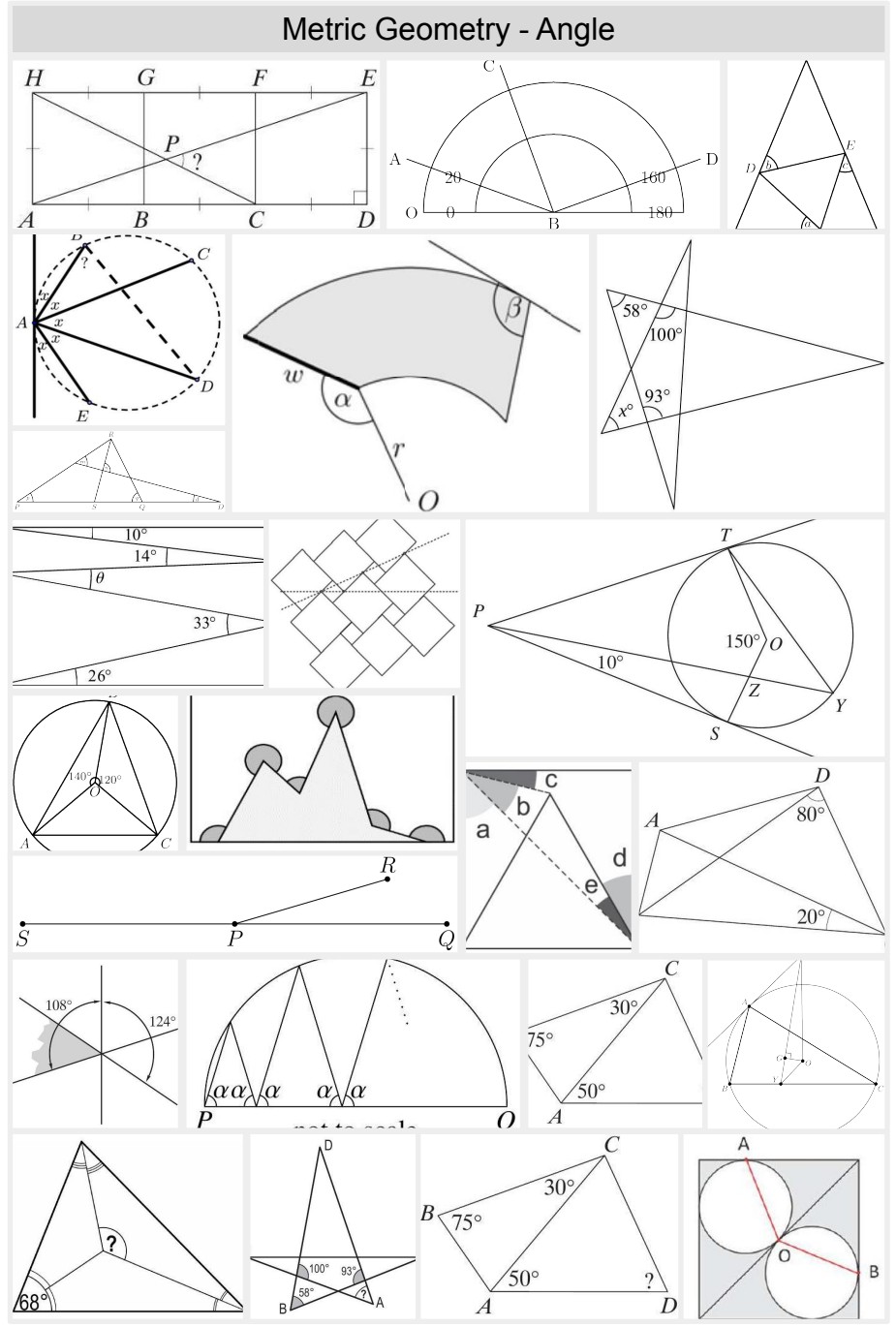

Figure 29: Some images from Metric Geometry - Angle.

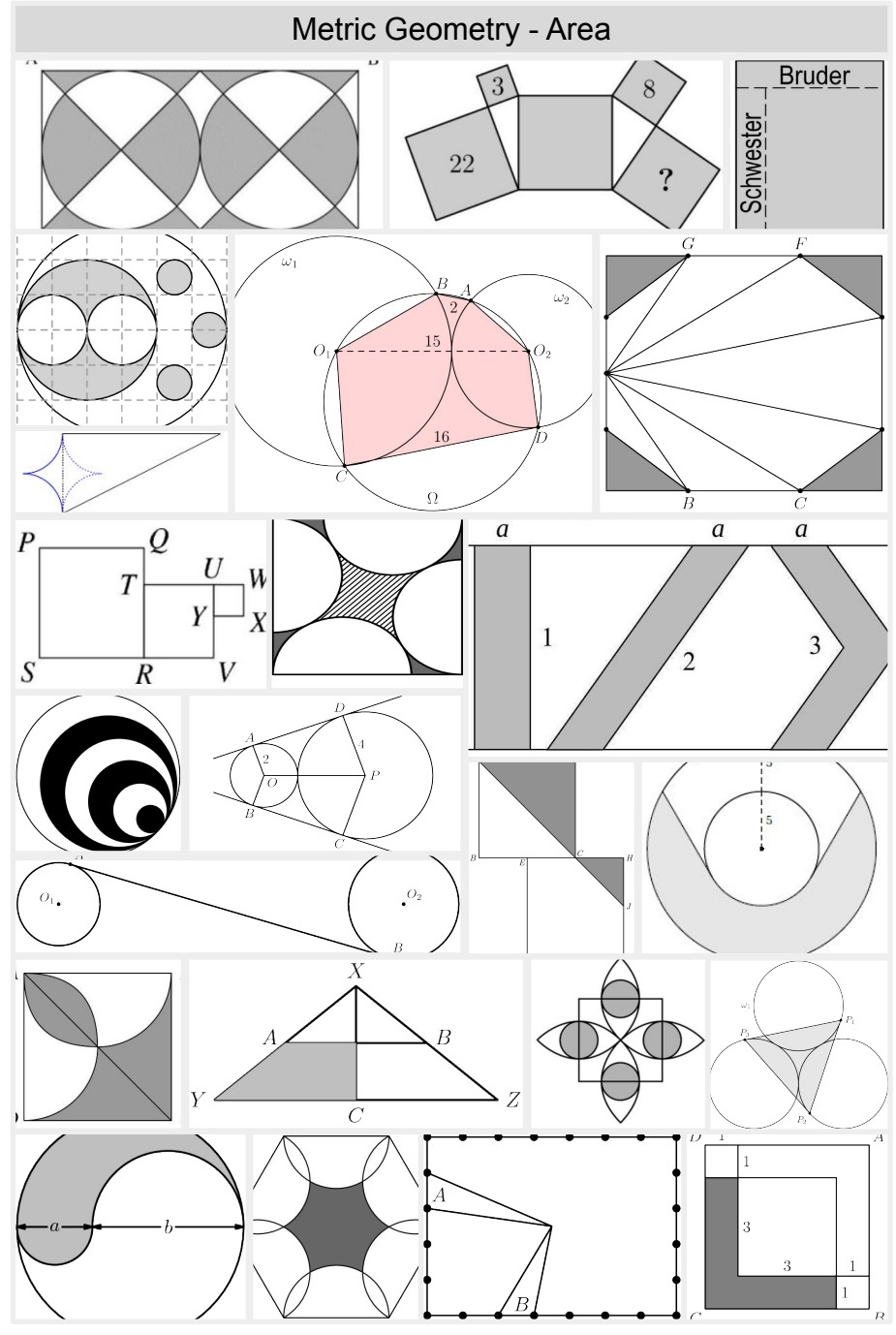

Figure 30: Some images from Metric Geometry - Area.

# Metric Geometry - Length

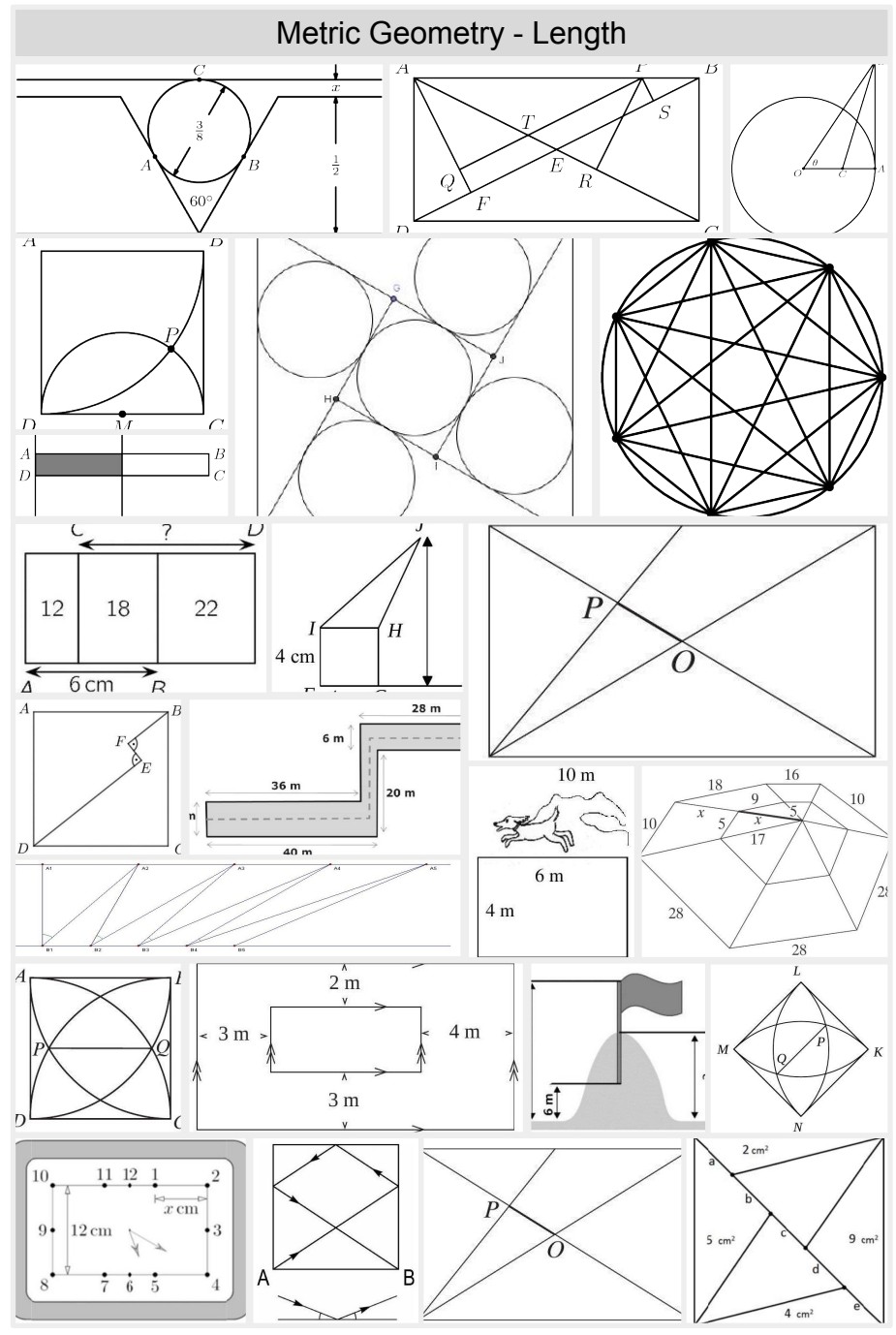

Figure 31: Some images from Metric Geometry - Length.

# Solid Geometry

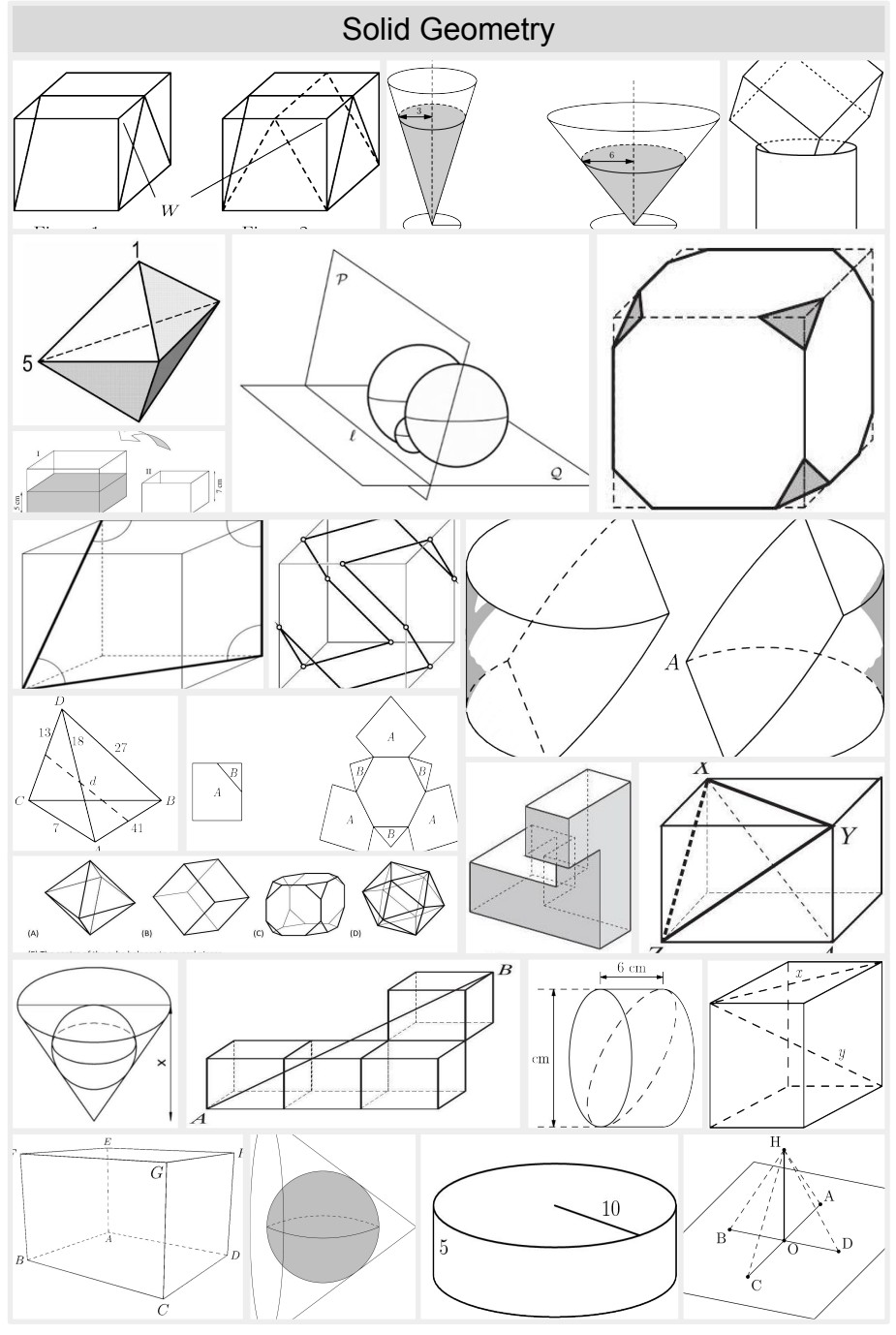

Figure 32: Some images from Solid Geometry.

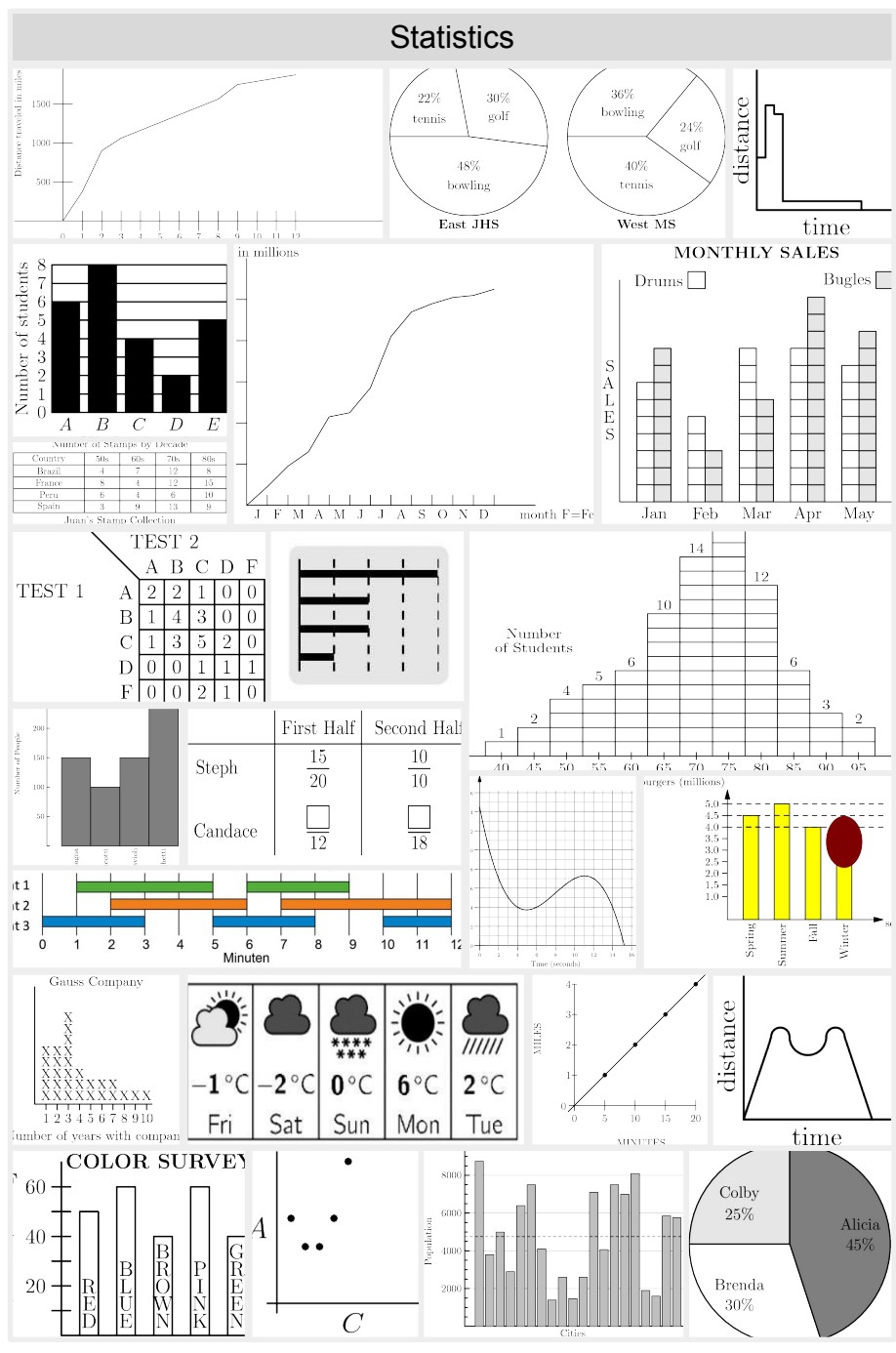

Figure 33: Some images from Statistics.

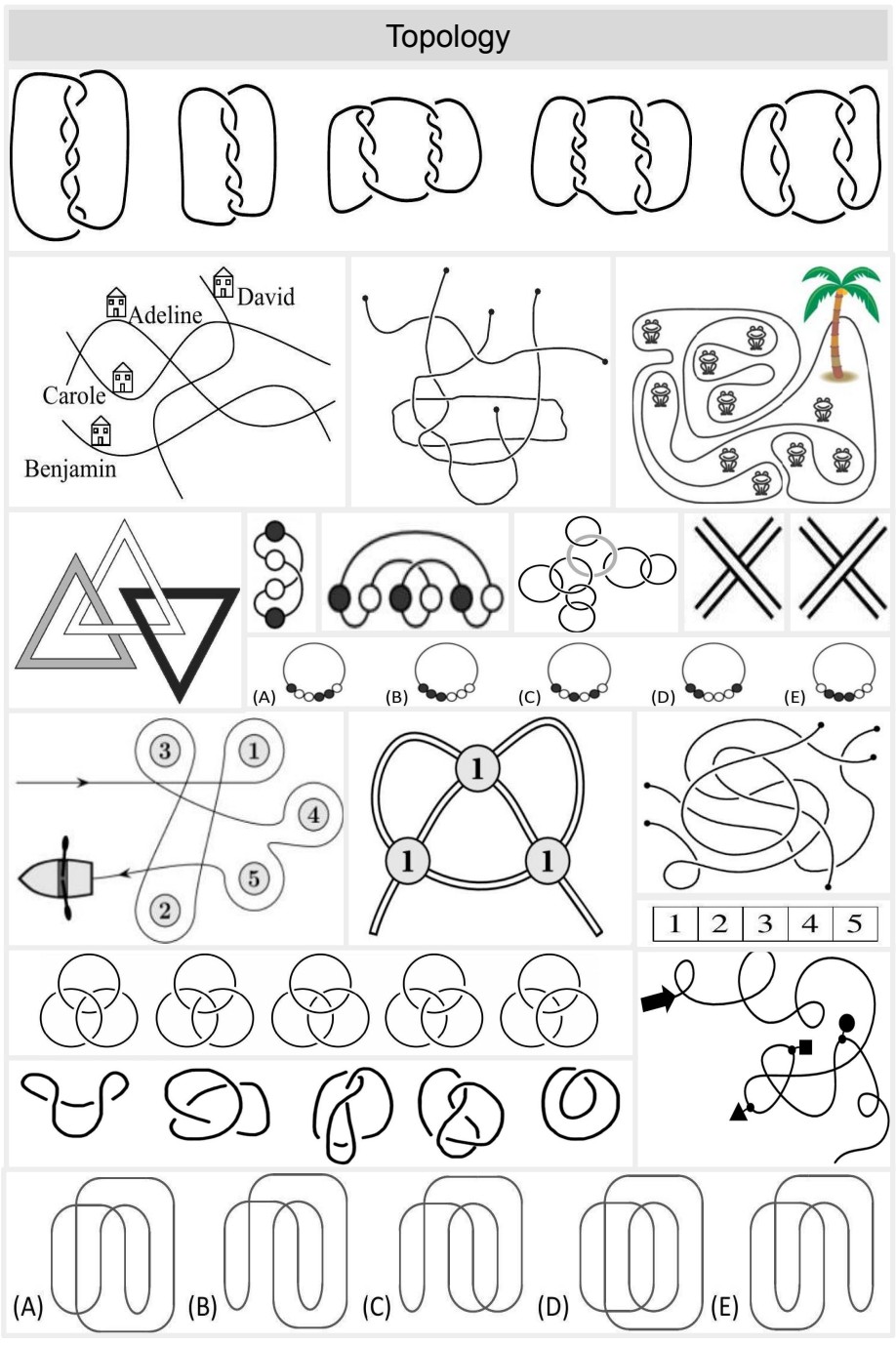

Figure 34: Some images from Topology.

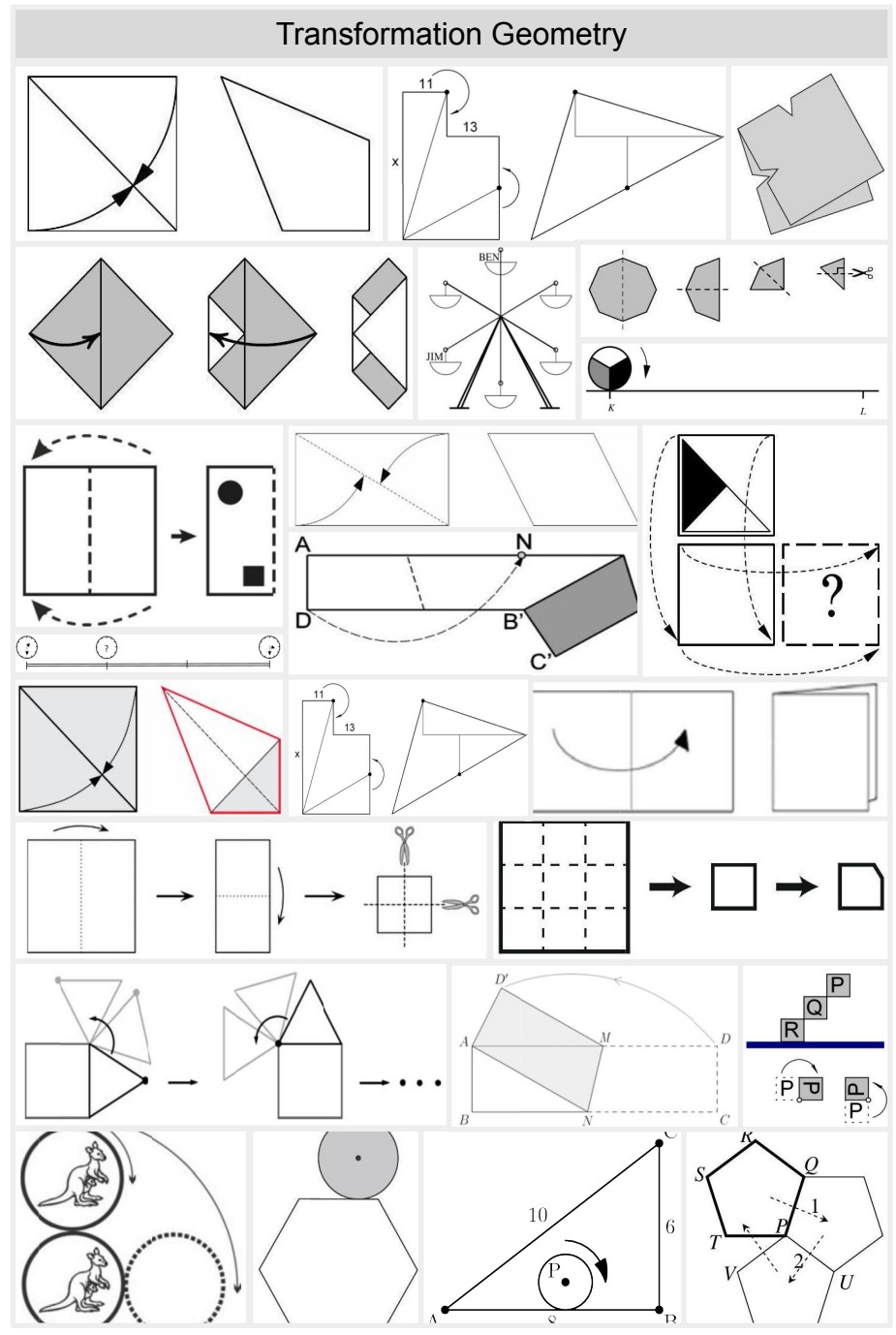

Figure 35: Some images from Transformation Geometry.

**Reasoning Error**

**Question:** In triangle $ABC$, $AB = AC$, $AE = AD$ and angle $BAD = 30°$. What is the size of angle $CDE$ ?
(A) $10°$
(B) $15°$
(C) $20°$
(D) $25°$
(E) $30°$

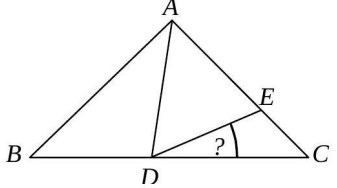

**Ground Truth:** (B) $15°$

---

**GPT4V Answer:**

First, let's review the given information:
- Triangle $ABC$ is isosceles with $AB = AC$. - $AE = AD$. - $\angle BAD = 30°$.
Given that $AB = AC$, we know that $\angle ABC = \angle ACB$ because the base angles of an isosceles triangle are equal.
Since $\angle BAD = 30°$ and $AE = AD$, we have another isosceles triangle, $\triangle ADE$, and $\angle ADE = \angle AED$. Let's call this unknown angle $\theta$.
Because $AE = AD$, triangle $ADE$ is isosceles, which makes $\angle ADE = \angle AED = \theta$.
Also, the sum of angles in a triangle is $180°$, so in $\triangle ADE$, we have:

$$\color{red}{\angle DAE + \angle ADE + \angle AED = 180°}$$
$$\color{red}{30° + \theta + \theta = 180°}$$

$$2\theta = 180° - 30°$$
$$2\theta = 150°$$
$$\theta = 75°$$

So, both $\angle ADE$ and $\angle AED$ are $75°$.
Now we move to $\triangle CDE$. We are looking for $\angle CDE$. The angle $\angle CDE$ is part of $\angle CDA$, which is an exterior angle to $\triangle ADE$ and equals the sum of the opposite interior angles, which we have found to be $\angle ADE + \angle AED$.
Therefore:

$$\angle CDA = \angle ADE + \angle AED$$
$$\angle CDA = 75° + 75°$$
$$\angle CDA = 150°$$

Since $\angle CDA$ is a straight line and is part of $\angle CDE$, the following relationship holds:

$$\angle CDA = \angle CDE + \angle ADE$$
$$150° = \angle CDE + 75°$$
$$\angle CDE = 150° - 75°$$
$$\angle CDE = 75°$$

This is not amongst the given options, indicating that there's an error.
[Truncated Due to Length]

Figure 36: An example of reasoning error.

**Reasoning Error**

## Question:

Two equal trains, each with 31 numbered wagons, travel in opposite directions. When the wagon number 7 of a train is side by side with the wagon number 12 of the other train, which wagon is side by side with the wagon number 11 ?

**Ground Truth:** 8

## GPT4V Answer:

To solve this problem, we'll use the information that we have about the wagons that are currently side by side. We are told that wagon number 7 of one train is side by side with wagon number 12 of the other train. We can use this to establish a pattern or a relationship between the numbers of the wagons that are side by side.

**If wagon 7 is next to 12, we can infer that as one train's wagon number increases, the other train's corresponding wagon number decreases if the trains are moving in opposite directions. The difference between the wagon numbers is 5 (12 - 7 = 5).**

Now, if we want to find out which wagon would be side by side with wagon number 11, we can use the established difference to determine the opposite wagon number. We subtract 5 from 11 to find the matching wagon number on the other train:

$11 - 5 = 6$

Therefore, the wagon number 6 of the one train will be side by side with wagon number 11 of the other train. So, the solution is:

$$\boxed{6}$$

Figure 37: An example of reasoning error.

**Vision Recognition Error**

**Question:** The students in Mrs. Sawyer's class were asked to do a taste test of five kinds of candy. Each student chose one kind of candy. A bar graph of their preferences is shown. What percent of her class chose candy E?

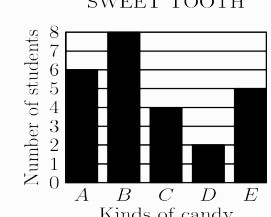

**Ground Truth:** 2

---

**GPT4V Answer:**

To find the percentage of the class that chose candy E, we first need to determine the number of students who chose candy E by looking at the bar graph. Then, we calculate the total number of students who participated in the taste test by adding the number of students who chose each kind of candy. Finally, we can find the percentage of students who chose candy E by dividing the number of students who chose candy E by the total number of students and then multiplying by 100.

From the bar graph, the number of students who chose candy E can be counted directly:

Candy E chosen by = 2 students

Now let's count the total number of students who participated:

Candy A chosen by = 7 students Candy B chosen by = 8 students Candy C chosen by = 5 students Candy D chosen by = 3 students Candy E chosen by = 2 students (as identified earlier)

Total number of students = 7 + 8 + 5 + 3 + 2 = 25 students

Now we determine the percentage of students who chose candy E:

$$\text{Percentage choosing candy E} = \left( \frac{\text{Number choosing candy E}}{\text{Total number of students}} \right) \times 100$$

$$\text{Percentage choosing candy E} = \left( \frac{2}{25} \right) \times 100$$

$$\text{Percentage choosing candy E} = \frac{2}{25} \times 100$$

$$\text{Percentage choosing candy E} = 0.08 \times 100$$

$$\text{Percentage choosing candy E} = 8\%$$

Therefore, the percentage of the class that chose candy E is:

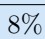

Figure 38: An example of vision recognition error.

**Vision Recognition Error**

**Question:** The sums of the all the three numbers on each side of the triangle are equal. Two numbers happened to be stained with ink. How much is the sum of these two numbers?

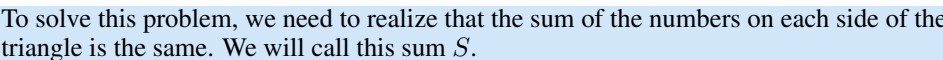

**Ground Truth:** 2

**GPT4V Answer:**

To solve this problem, we need to realize that the sum of the numbers on each side of the triangle is the same. We will call this sum $S$.

Let's denote the two unknown numbers, which are obscured by ink, as $x$ and $y$.

We can write the sum on each side of the triangle using these variables. For the side with the numbers 7, $x$, and 1, the sum is $7 + x + 1$. **For the side with the numbers 3, 6, and $y$, the sum is $3 + 6 + y$.**

Because the sums are equal, we have:

$$7 + x + 1 = 3 + 6 + y$$

$$8 + x = 9 + y$$

We want to find the sum of $x$ and $y$. Let's add $y$ to both sides of the equation to move terms involving $y$ to the left side and terms involving $x$ to the right side:

$$y + 9 + y = 8 + x + y$$

$$2y + 9 = 8 + x + y$$

Now subtract $y$ from both sides:

$$y + 9 = 8 + x$$

We know that the other side of the triangle is $1 + 6 + 3 = 10$, so the sum of the numbers on each side must be 10, which means:

$$y + 9 = 10$$

$$y = 10 - 9$$

$$y = 1$$

Now we substitute $y = 1$ into the sum for the other sides to find $x$:

$$x + 8 = 10$$

$$x = 10 - 8$$

$$x = 2$$

So, the two numbers are $x = 2$ and $y = 1$, and their sum is:

$$x + y = 2 + 1 = 3$$

Thus, the sum of the two numbers that are stained with ink is 3, and so we have:

$$\boxed{3}$$

Figure 39: An example of vision recognition error.

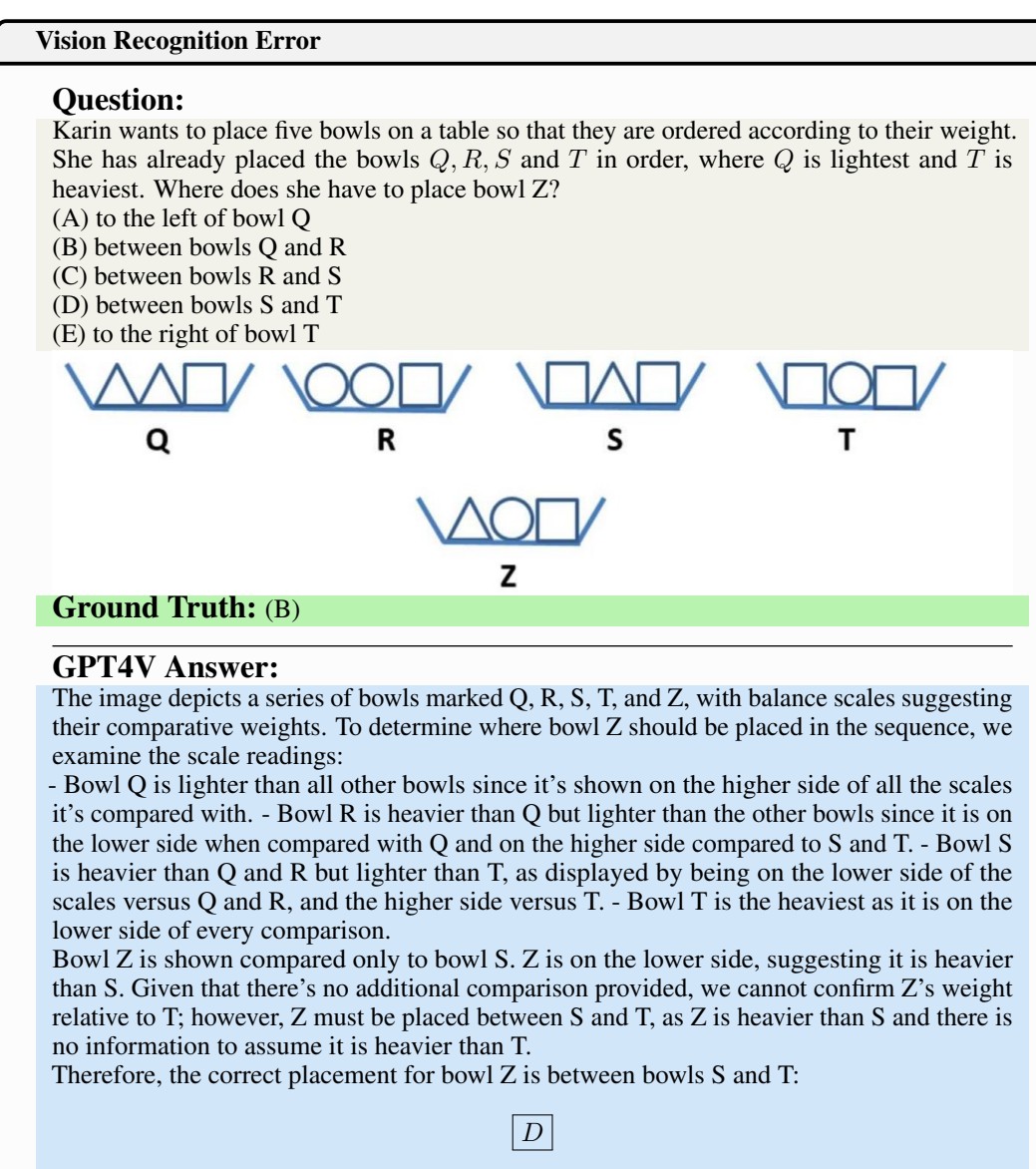

**Vision Recognition Error**

## Question:

Karin wants to place five bowls on a table so that they are ordered according to their weight. She has already placed the bowls $Q, R, S$ and $T$ in order, where $Q$ is lightest and $T$ is heaviest. Where does she have to place bowl Z?

(A) to the left of bowl Q
(B) between bowls Q and R
(C) between bowls R and S
(D) between bowls S and T
(E) to the right of bowl T

**Ground Truth:** (B)

## GPT4V Answer:

The image depicts a series of bowls marked Q, R, S, T, and Z, with balance scales suggesting their comparative weights. To determine where bowl Z should be placed in the sequence, we examine the scale readings:

- Bowl Q is lighter than all other bowls since it's shown on the higher side of all the scales it's compared with. - Bowl R is heavier than Q but lighter than the other bowls since it is on the lower side when compared with Q and on the higher side compared to S and T. - Bowl S is heavier than Q and R but lighter than T, as displayed by being on the lower side of the scales versus Q and R, and the higher side versus T. - Bowl T is the heaviest as it is on the lower side of every comparison.

Bowl Z is shown compared only to bowl S. Z is on the lower side, suggesting it is heavier than S. Given that there's no additional comparison provided, we cannot confirm Z's weight relative to T; however, Z must be placed between S and T, as Z is heavier than S and there is no information to assume it is heavier than T.

Therefore, the correct placement for bowl Z is between bowls S and T:

$$\boxed{D}$$

Figure 40: An example of vision recognition error. Model is experiencing severe hallucinations in this example

**Knowledge Error**

**Question:** We see in the diagram at the right a piece of the graphic of the function

$$f(x) = ax^3 + bx^2 + cx + d.$$

What is the value of $b$?

**Ground Truth:** -2

**GPT4V Answer:**

To determine the value of $b$ in the cubic function $f(x) = ax^3 + bx^2 + cx + d$, we need to look at the properties of the function's graph provided in the diagram.

From the graph, we can identify the coordinates of the two local extrema (maximum and minimum). At these points, the first derivative $f'(x)$ of the function must be zero since the slope of the tangent line at these points is zero.

The first derivative of the cubic function $f(x)$ is given by:

$$f'(x) = 3ax^2 + 2bx + c.$$

We have two extrema points visible on the graph, at $x = -1$ and $x = 1$. Let's set up the derivative to be equal to zero at these points:

For $x = -1$:
$$0 = 3a(-1)^2 + 2b(-1) + c$$
$$0 = 3a - 2b + c$$
$$2b = 3a + c \quad (1)$$

For $x = 1$:
$$0 = 3a(1)^2 + 2b(1) + c$$
$$0 = 3a + 2b + c \quad (2)$$

Let's solve this system of equations (1) and (2) for $b$.
Subtract (1) from (2):

$$3a + 2b + c - (3a - 2b + c) = 0$$
$$3a + 2b + c - 3a + 2b - c = 0$$
$$4b = 0$$
$$b = 0$$

Therefore, the value of $b$ is zero.

$$\boxed{b = 0}$$

Figure 41: An example of knowledge error.

**Knowledge Error**

**Question:** In the accompanying figure, segments $AB$ and $CD$ are parallel, the measure of angle $D$ is twice the measure of angle $B$, and the measures of segments $AB$ and $CD$ are $a$ and $b$ respectively. Then the measure of $AB$ is equal to
(A) $\frac{1}{2}a + 2b$
(B) $\frac{3}{2}b + \frac{3}{4}a$
(C) $2a - b$
(D) $4b - \frac{1}{2}a$
(E) $a + b$

**Ground Truth:** -2

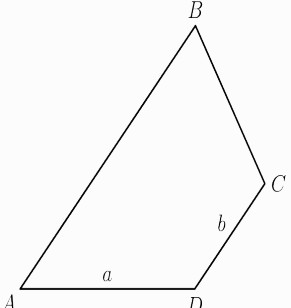

**GPT4V Answer:**

The image you've provided appears to be a quadrilateral with sides $AB$ and $CD$ denoted as being parallel and segments $AB$ labeled as $a$, and $CD$ as $b$. However, without additional information such as angle measures or the relationship between the sides and angles, it is impossible to determine a definitive relationship between $a$ and $b$ based on the given options (A) through (E).

Typically, in problems like this, you might apply properties of parallel lines and transversals or properties of similar triangles if any triangle relationships can be determined. But the given information is insufficient to apply these properties directly.

Given the description of the problem, it seems to be expecting the application of a geometric principle or theorem which is not evident from the image alone. It's possible that there is either additional information that has not been provided or there may be an error in the problem's depiction.

With the information currently at hand, it is not feasible to reach a solution, and I cannot provide an answer in the requested format □. If you can provide further details or clarify the conditions of the problem, I would be happy to reattempt to solve it.

Figure 42: An example of knowledge error.

## Question:

Johann stacks $1 \times 1$ cubes on the squares of a $4 \times 4$ grid. The diagram <image1> shows how many cubes were piled on top of each other on each square of the grid. What will Johann see if he looks from behind (hinten) at the tower?

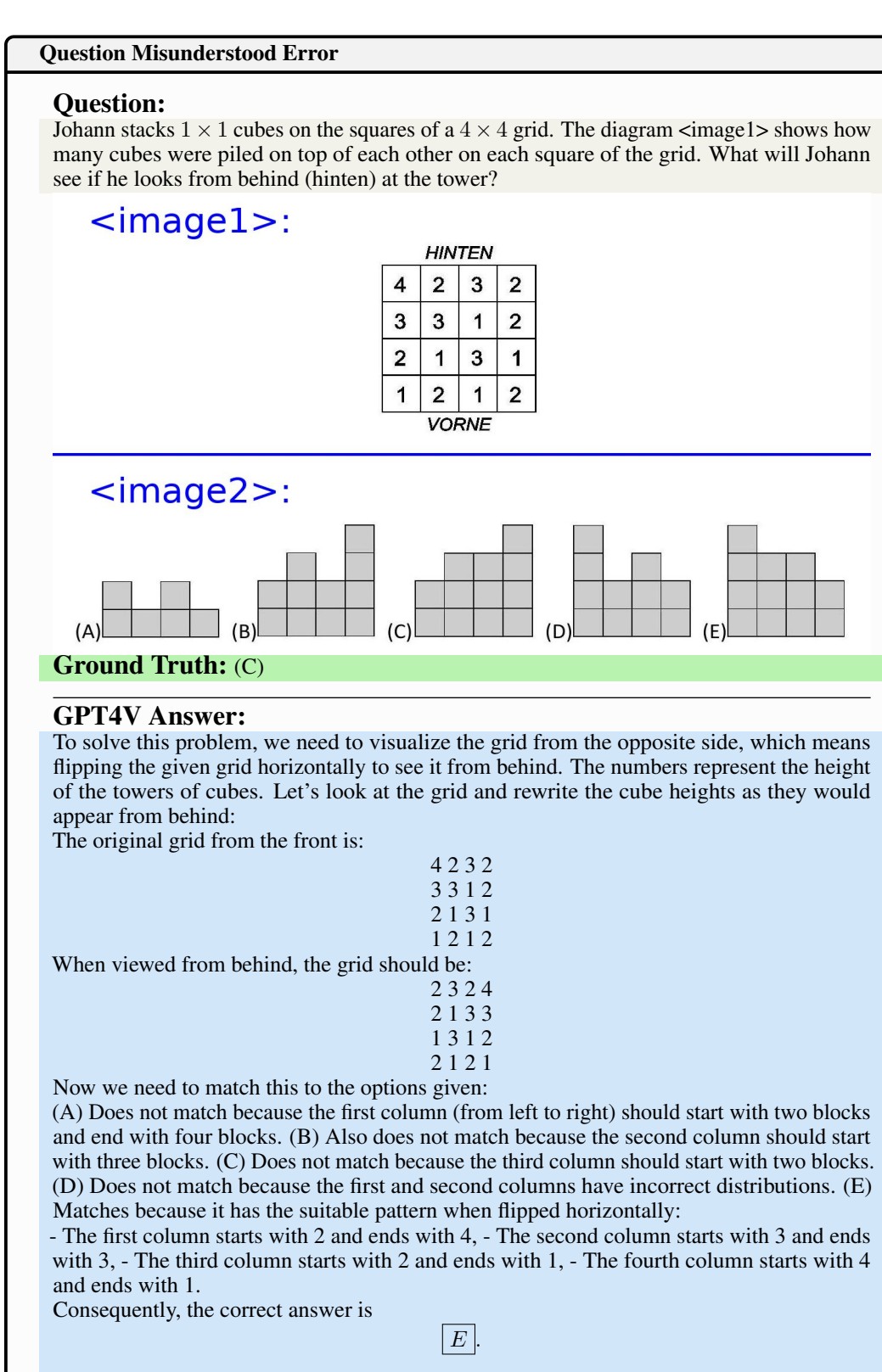

## Ground Truth: (C)

## GPT4V Answer:

To solve this problem, we need to visualize the grid from the opposite side, which means flipping the given grid horizontally to see it from behind. The numbers represent the height of the towers of cubes. Let's look at the grid and rewrite the cube heights as they would appear from behind:

The original grid from the front is:

$$4\ 2\ 3\ 2$$
$$3\ 3\ 1\ 2$$
$$2\ 1\ 3\ 1$$
$$1\ 2\ 1\ 2$$

When viewed from behind, the grid should be:

$$2\ 3\ 2\ 4$$
$$2\ 1\ 3\ 3$$
$$1\ 3\ 1\ 2$$
$$2\ 1\ 2\ 1$$

Now we need to match this to the options given:

(A) Does not match because the first column (from left to right) should start with two blocks and end with four blocks. (B) Also does not match because the second column should start with three blocks. (C) Does not match because the third column should start with two blocks. (D) Does not match because the first and second columns have incorrect distributions. (E) Matches because it has the suitable pattern when flipped horizontally:

- The first column starts with 2 and ends with 4, - The second column starts with 3 and ends with 3, - The third column starts with 2 and ends with 1, - The fourth column starts with 4 and ends with 1.

Consequently, the correct answer is

$$\boxed{E}.$$

Figure 43: An example of question understood error.

**Question Misunderstood Error**

**Question:** Each one of the four keys locks exactly one padlock. Every letter on a padlock stands for exactly one digit. Same letters mean same digits. Which letters must be written on the fourth padlock?
(A) GDA
(B) ADG
(C) GAD
(D) GAG
(E) DAD

**Ground Truth:** (D) GAG

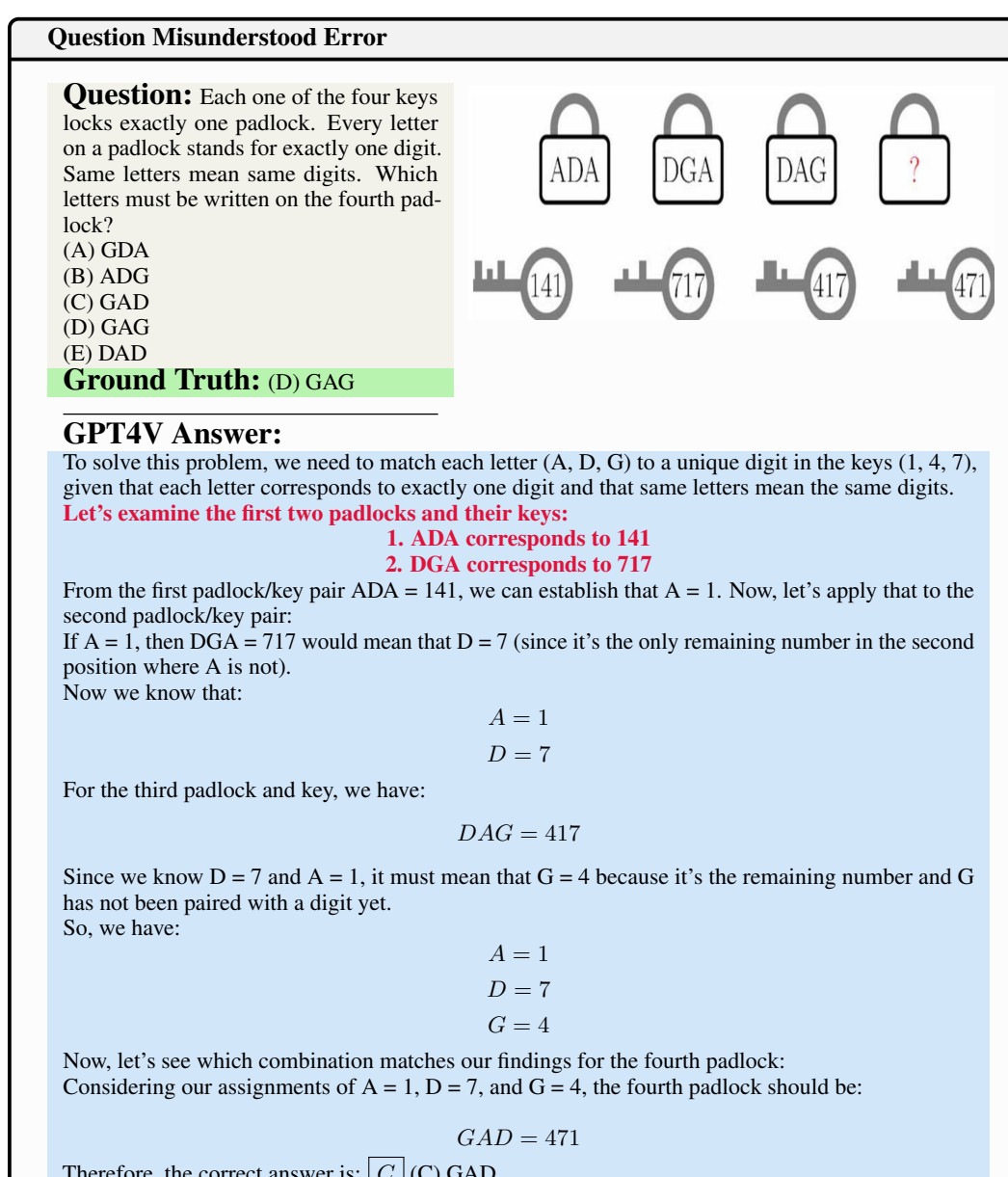

---

**GPT4V Answer:**

To solve this problem, we need to match each letter (A, D, G) to a unique digit in the keys (1, 4, 7), given that each letter corresponds to exactly one digit and that same letters mean the same digits.
**Let's examine the first two padlocks and their keys:**
**1. ADA corresponds to 141**
**2. DGA corresponds to 717**
From the first padlock/key pair ADA = 141, we can establish that A = 1. Now, let's apply that to the second padlock/key pair:
If A = 1, then DGA = 717 would mean that D = 7 (since it's the only remaining number in the second position where A is not).
Now we know that:
$$A = 1$$
$$D = 7$$

For the third padlock and key, we have:

$$DAG = 417$$

Since we know D = 7 and A = 1, it must mean that G = 4 because it's the remaining number and G has not been paired with a digit yet.
So, we have:
$$A = 1$$
$$D = 7$$
$$G = 4$$

Now, let's see which combination matches our findings for the fourth padlock:
Considering our assignments of A = 1, D = 7, and G = 4, the fourth padlock should be:

$$GAD = 471$$

Therefore, the correct answer is: $\boxed{C}$ (C) GAD

Figure 44: An example of question understood error. The model mistakenly believes that keys and locks are in one-to-one correspondence, but there is no such condition in the question.

**Reject to answer**

**Question:**
This picture <image1> shows a bracelet with pearls. Which of the bands below shows the same bracelet as above?

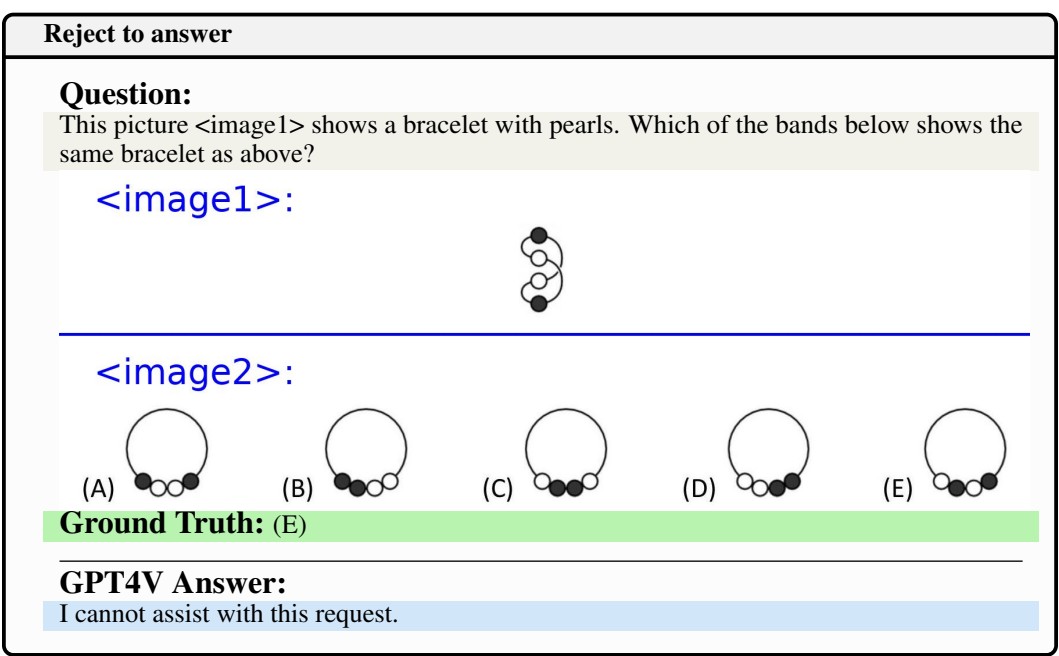

**Ground Truth:** (E)

**GPT4V Answer:**
I cannot assist with this request.

Figure 45: An example of reject to answer error.

**Reject to answer**

**Question:** $A$ 10 cm long piece if wire is folded so that every part is equally long (see diagram). The wire is then cut through in the two positions marked. How long are the three pieces created in this way?
(A) 2 cm, 3 cm, 5 cm
(B) 2 cm, 2 cm, 6 cm
(C) 1 cm, 4 cm, 5 cm
(D) 1 cm, 3 cm, 6 cm
(E) 3 cm, 3 cm, 4 cm

**Ground Truth:** (B) $\frac{1}{2}a^2$

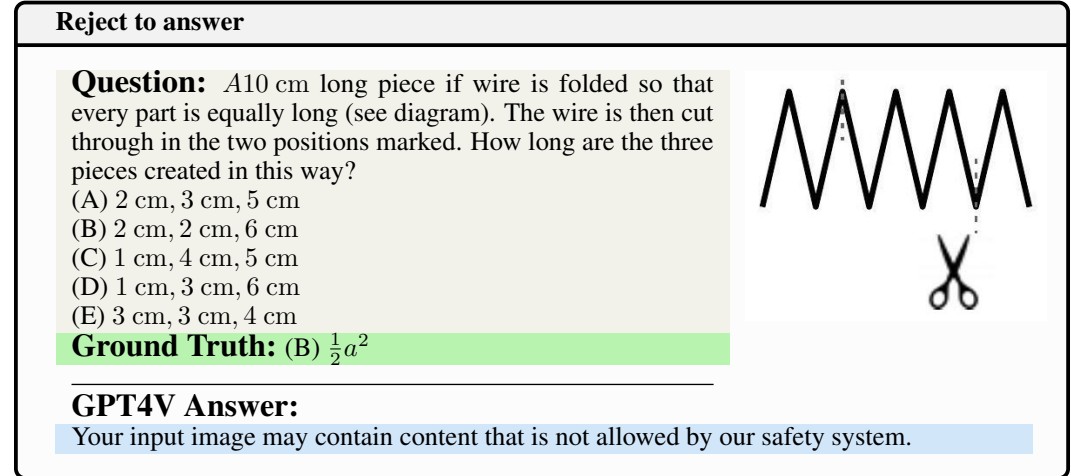

**GPT4V Answer:**
Your input image may contain content that is not allowed by our safety system.

Figure 46: An example of reject to answer error.

**Calculation Error**

**Question:** The diagram shows three concentric circles and two perpendicular, common diameters of the three circles. The three grey sections are of equal area, the small circle has radius 1. What is the product of the radii of the three circles?

(A) $\sqrt{6}$

(B) 3

(C) $\frac{3\sqrt{3}}{2}$

(D) $2\sqrt{2}$

(E) 6

**Ground Truth:** (A) $\sqrt{6}$

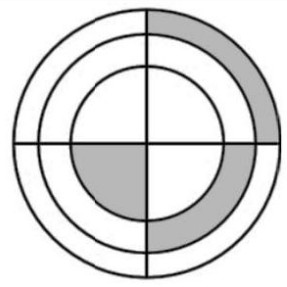

**GPT4V Answer:**

The problem states that the three grey sections are of equal area, and we know the radius of the smallest circle to be 1. Let's denote the radius of the middle circle as $r$ and the radius of the largest circle as $R$.

The area of a sector of a circle is given by $\frac{1}{2}\theta r^2$, where $\theta$ is the central angle in radians, and $r$ is the radius of the circle. Since the three grey sections are equal in area and shaped as quarter circles, each grey section has a central angle of $\frac{\pi}{2}$ radians.

Therefore, the area of each grey section can be represented as follows:

For the smallest circle:
$$A_{\text{small}} = \frac{1}{2} \times \frac{\pi}{2} \times 1^2 = \frac{\pi}{4}$$

For the middle circle, the area of the sector minus the area of the smaller circle's quarter:
$$A_{\text{middle}} = \frac{1}{2} \times \frac{\pi}{2} \times r^2 - A_{\text{small}} = \frac{\pi r^2}{4} - \frac{\pi}{4}$$

For the largest circle, the area of the sector minus the area of the middle circle's quarter:
$$A_{\text{large}} = \frac{1}{2} \times \frac{\pi}{2} \times R^2 - A_{\text{middle}} - A_{\text{small}} = \frac{\pi R^2}{4} - \frac{\pi r^2}{4}$$

Since the areas are equal: $A_{\text{small}} = A_{\text{middle}} = A_{\text{large}}$

We can set the expressions for $A_{\text{small}}$ and $A_{\text{middle}}$ equal to each other:
$$\frac{\pi}{4} = \frac{\pi r^2}{4} - \frac{\pi}{4}$$
$$\Rightarrow \frac{\pi r^2}{4} = \frac{2\pi}{4}$$
$$\Rightarrow r^2 = 2$$
$$\Rightarrow r = \sqrt{2}$$

Now, do the same for $A_{\text{middle}}$ and $A_{\text{large}}$:
$$\textcolor{red}{\frac{\pi r^2}{4} - \frac{\pi}{4} = \frac{\pi R^2}{4} - \frac{\pi r^2}{4}}$$
$$\textcolor{red}{\Rightarrow \frac{\pi R^2}{4} - \frac{2\pi r^2}{4} = 0}$$
$$\textcolor{red}{\Rightarrow \frac{\pi R^2}{4} = \frac{2\pi}{4}}$$
$$\textcolor{red}{\Rightarrow R^2 = 2}$$
$$\textcolor{red}{\Rightarrow R = \sqrt{2} \times r = 2}$$

The product of the radii of the three circles is: $1 \times r \times R = 1 \times \sqrt{2} \times 2 = 2\sqrt{2}$

Therefore, the answer is:

$\boxed{D}$

Figure 47: Full example of calculation error. The wrong steps are highlighted in **red**.

**Calculation Error**

**Question:** The diagram shows two adjoining squares with side lengths $a$ and $b$ (with $a < b$). How big is the area of the grey triangle?

(A) $\sqrt{ab}$
(B) $\frac{1}{2}a^2$
(C) $\frac{1}{2}b^2$
(D) $\frac{1}{4}\left(a^2 + b^2\right)$
(E) $\frac{1}{2}\left(a^2 + b^2\right)$

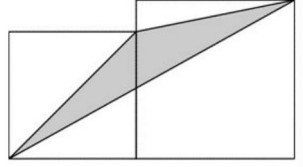

**Ground Truth:** (B) $\frac{1}{2}a^2$

**GPT4V Answer:**

The grey triangle in the diagram can be seen as the difference between the large square with side $b$ and the smaller square with side $a$, minus the two right triangles that are outside of the grey area but inside the larger square.

The area of the larger square is $b^2$, and the area of the smaller square is $a^2$. The two right triangles each have legs of length $a$ and $b - a$, thus the area of one such right triangle is $\frac{1}{2} \times a \times (b - a)$. There are two of these triangles, so the total area of the two triangles is $\frac{1}{2} \times a \times (b - a) \times 2 = a \times (b - a)$. Now, the area of the grey triangle is therefore the area of the large square minus the area of the small square and the two right triangles:

$$b^2 - a^2 - a(b - a) = b^2 - a^2 - ab + a^2 = b^2 - ab$$

We simplify this by factoring out a $b$:

$$b(b - a)$$

Now, we notice that $b - a$ is the height of the grey triangle, and, by looking at the diagram, we can see that the base of the grey triangle overlaps exactly with the side length of the larger square, which is $b$.

Therefore, the area of the grey triangle is $\frac{1}{2} \times$ base $\times$ height, which in this case is $\frac{1}{2} \times b \times (b - a)$:
Area of grey triangle = $\frac{1}{2} \times b \times (b - a)$

But since we can't simplify this expression using the given variables directly, we must find a match among the options provided. By adding and subtracting $\frac{1}{2}a^2$ within the expression of the larger square area $b^2$, we can rewrite the expression:

$$b^2 - a(b - a) = b^2 - ab + a^2 - \frac{1}{2}a^2 + \frac{1}{2}a^2$$

$$= \frac{1}{2}a^2 + \frac{1}{2}b^2 - \frac{1}{2}a^2 - ab + \frac{1}{2}a^2$$

$$= \frac{1}{2}(a^2 + b^2) - \frac{1}{2}a^2$$

Therefore, the area of the grey triangle is equal to half the sum of the areas of the two squares minus half the area of the smaller square:
Area of grey triangle = $\frac{1}{2}(a^2 + b^2) - \frac{1}{2}a^2$

$$= \frac{1}{2}b^2$$

This matches option (C), so the area of the grey triangle is $\frac{1}{2}b^2$.
Thus, the answer is:

$$\boxed{C}$$

Figure 48: An example of calculation error. The wrong steps are highlighted in **red**.

**Example of Algebra**

**Question:** The numbers in the five circles around each house add up to 20 . Some numbers are missing. Which number does the question mark stand for?

**Ground Truth:** 9

**GPT4V Image Caption:**
The diagram features two houses connected by a series of circles, each containing a number or a question mark. Each house has a large circle on its roof and two smaller circles on the ground, one on each side.
The house on the left has the number 6 in the circle on its roof. The two smaller circles on the ground next to this house contain the numbers 2 and 5, respectively. The house on the right has the number 3 in the circle on its roof. **The circle on the ground to the right of this house contains the number 1, and the circle on the ground to the left of this house contains a question mark.**
The circles are connected by lines, suggesting a relationship or a pattern that needs to be discovered. ... **Additionally, there is a line connecting the two houses, specifically from the circle with the number 5 to the circle with the question mark.**
The goal of the problem is likely to determine the number that should replace the question mark, based on the pattern or relationship between the numbers in the connected circles.

**GPT4 Answer:**
To solve this problem, we need to find the pattern or relationship between the numbers in the connected circles, given that the sum of the numbers around each house adds up to 20. For the house on the left: The numbers we know are 6 (on the roof), 2, and 5 (on the ground). Let's add these numbers: $6 + 2 + 5 = 13$. Since the sum of the numbers around the house must be 20, we need to find the missing number that, when added to 13, equals 20. Let's denote the missing number on the left side as $x$. So, $13 + x = 20$. Solving for $x$, we get $x = 20 - 13 = 7$.
For the house on the right: The numbers we know are 3 (on the roof) and 1 (on the ground). We also know now that the missing number on the left side, which is connected to the house on the right, is 7. Let's add these numbers: $3 + 1 + 7 = 11$. Since the sum of the numbers around the house must be 20, we need to find the missing number that, when added to 11, equals 20. Let's denote the missing number on the right side as $y$. So, $11 + y = 20$. Solving for $y$, we get $y = 20 - 11 = 9$. Therefore, the number that should replace the question mark is $\boxed{9}$.

**Gemini Pro Answer:**
Let's assume the missing numbers as x and y. The sum of the numbers in the circles around the first house = 6 + 2 + 5 + x = 13 + x. **The sum of the numbers in the circles around the second house = 3 + y + 1 = 4 + y.** Since the sum of the numbers in the circles around each house is 20, we have: **13 + x + 4 + y = 20; 17 + x + y = 20; x + y = 3. As the number in each circle is different, the possible values of x and y are (0, 3), (1, 2), and (2, 1). The question mark can be either $\boxed{0 \text{ or } 2}$.**

**Qwen-VL-Max Answer:**
**The sum of the numbers in the circles around the left house is $6 + 3 + 2 + 5 = 16$, so the number on the top right corner is $20 - 16 = \boxed{4}$. The sum of the numbers in the circles around the right house is $3 + 4 + 1 + ? = 20$, so the question mark stands for $\boxed{8}$.**

Figure 49: An example of algebra question of GPT-4 with image caption. Although there are small hallucinations marked in red, The model correctly solves the problem based on the image description. Gemini Pro and Qwen-VL-Max fail to solve it because of weak visual recognition and reasoning ability.

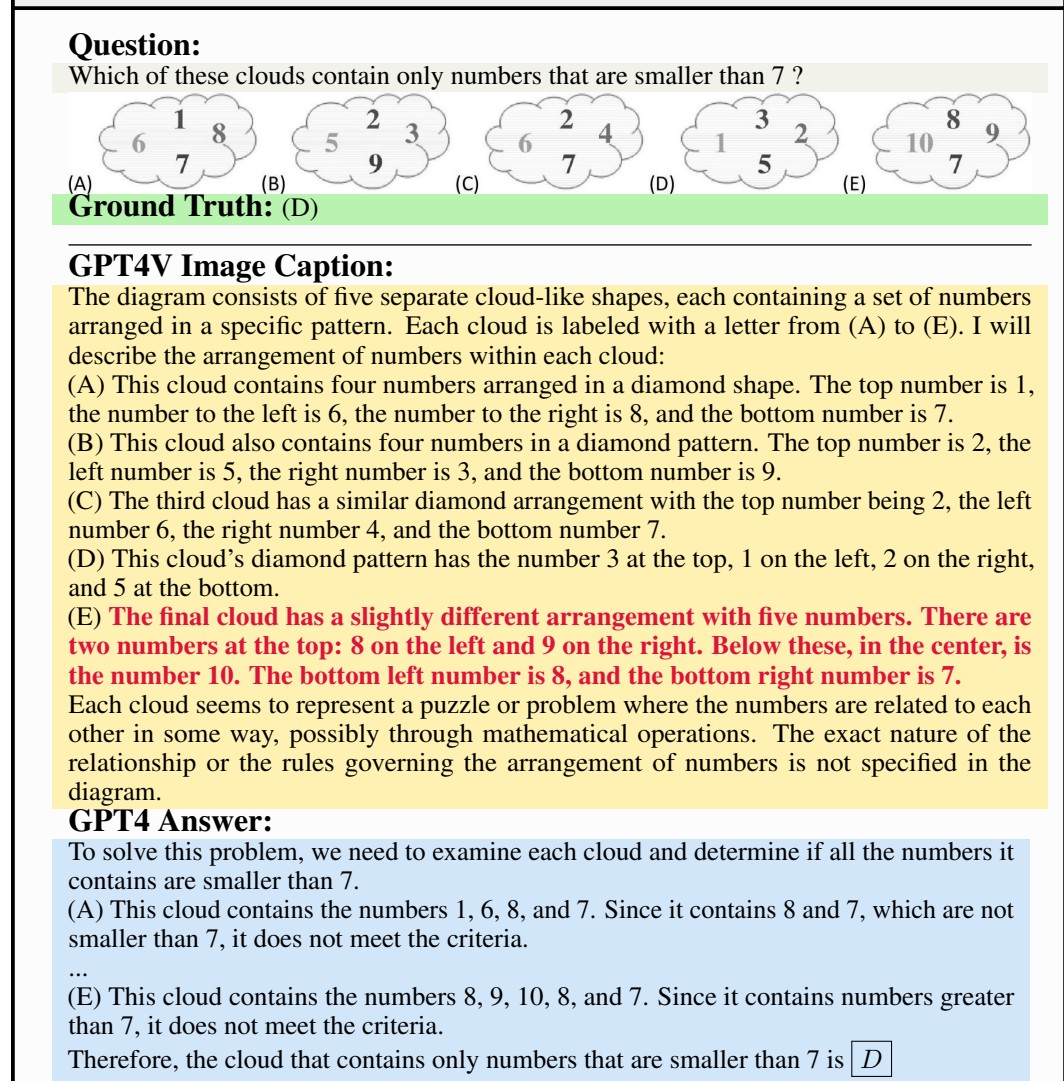

**Example of Arithmetic**

**Question:**

Which of these clouds contain only numbers that are smaller than 7 ?

**Ground Truth:** (D)

**GPT4V Image Caption:**

The diagram consists of five separate cloud-like shapes, each containing a set of numbers arranged in a specific pattern. Each cloud is labeled with a letter from (A) to (E). I will describe the arrangement of numbers within each cloud:

(A) This cloud contains four numbers arranged in a diamond shape. The top number is 1, the number to the left is 6, the number to the right is 8, and the bottom number is 7.

(B) This cloud also contains four numbers in a diamond pattern. The top number is 2, the left number is 5, the right number is 3, and the bottom number is 9.

(C) The third cloud has a similar diamond arrangement with the top number being 2, the left number 6, the right number 4, and the bottom number 7.

(D) This cloud's diamond pattern has the number 3 at the top, 1 on the left, 2 on the right, and 5 at the bottom.

(E) **The final cloud has a slightly different arrangement with five numbers. There are two numbers at the top: 8 on the left and 9 on the right. Below these, in the center, is the number 10. The bottom left number is 8, and the bottom right number is 7.**

Each cloud seems to represent a puzzle or problem where the numbers are related to each other in some way, possibly through mathematical operations. The exact nature of the relationship or the rules governing the arrangement of numbers is not specified in the diagram.

**GPT4 Answer:**

To solve this problem, we need to examine each cloud and determine if all the numbers it contains are smaller than 7.

(A) This cloud contains the numbers 1, 6, 8, and 7. Since it contains 8 and 7, which are not smaller than 7, it does not meet the criteria.

...

(E) This cloud contains the numbers 8, 9, 10, 8, and 7. Since it contains numbers greater than 7, it does not meet the criteria.

Therefore, the cloud that contains only numbers that are smaller than 7 is $\boxed{D}$

**Gemini Pro Answer:**

**(A) and (D)  (A) 1, 6, 7, 8  (D) 1, 2, 3, 4, 5**

**Qwen-VL-Max Answer:**

To determine which clouds contain only numbers that are smaller than 7, we need to look at each cloud and check if all the numbers inside it are less than 7.

**... The clouds that contain only numbers that are smaller than 7 are** $\boxed{(C), (D)}$ .

Figure 50: An example of arithmetic question of GPT-4 with image caption. Although there are small hallucinations marked in red, The model correctly solves the problem based on the image description. GPT-4V also does it correct while Gemini Pro and Qwen-VL-Max fail because of weak visual recognition ability.

**Example of Analytic Geometry**

### Question:

A square is placed in a co-ordinate system as shown. Each point $(x \mid y)$ of the square is deleted and replaced by the point $\left(\frac{1}{x} \mid \frac{1}{y}\right)$. Which diagram shows the resulting shape?

<image1>:

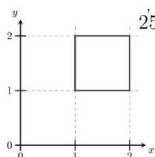

<image2>:

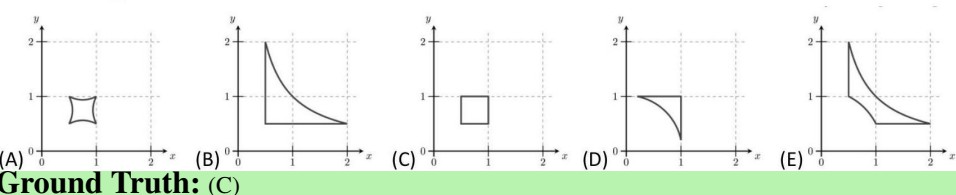

### GPT4V Image Caption:

**Image 1**: This diagram shows a Cartesian coordinate system with x and y-axes. The x-axis is labeled and ranges from 0 to 2, and the y-axis is labeled and ranges **from 0 to 2.5**. There is a square drawn in the first quadrant of the coordinate system. The square has its bottom left vertex at the point (1, 1) and its top right vertex at the point (2, 2), making the length of each side of the square 1 unit.

**Image 2**: This image contains five separate Cartesian coordinate system diagrams labeled (A) through (E), each with x and y-axes ranging from 0 to 2 on both axes...

(C) The shaded region is a smaller square within the first quadrant, with its bottom left vertex at approximately (0.5, 0.5) and its top right vertex at **approximately (1.5, 1.5)**...

### GPT4 Answer:

To solve this problem, we need to understand how the transformation $\left(\frac{1}{x} \mid \frac{1}{y}\right)$ affects the points of the square. The original square has vertices at $(1,1)$, $(2,1)$, $(1,2)$, and $(2,2)$.

1. Transform the bottom left vertex $(1,1)$: $\left(\frac{1}{1}, \frac{1}{1}\right) = (1,1)$. This point remains unchanged.
2. Transform the bottom right vertex $(2,1)$: $\left(\frac{1}{2}, \frac{1}{1}\right) = \left(\frac{1}{2}, 1\right)$. It moves closer to the y-axis.
3. Transform the top left vertex $(1,2)$: $\left(\frac{1}{1}, \frac{1}{2}\right) = \left(1, \frac{1}{2}\right)$. This point moves closer to the x-axis.
4. Transform the top right vertex $(2,2)$: $\left(\frac{1}{2}, \frac{1}{2}\right) = \left(\frac{1}{2}, \frac{1}{2}\right)$. This point moves closer to both axes.

After applying the transformation to all four vertices of the square, we can see that the new shape is also a square, but it is smaller and has its bottom left vertex at $\left(\frac{1}{2}, \frac{1}{2}\right)$ and its top right vertex at $(1,1)$. This matches the description of diagram (C), which shows a smaller square within the first quadrant, with its bottom left vertex at approximately $(0.5, 0.5)$ and its top right vertex at approximately $(1.5, 1.5)$. However, given the transformation, the top right vertex should precisely be at $(1,1)$, which still aligns best with the description of diagram (C) among the given options.

### GPT-4V Answer:

Let's analyze the transformation given by the function $f(x,y) = \left(\frac{1}{x}, \frac{1}{y}\right)$. $f(1,1) = \left(\frac{1}{1}, \frac{1}{1}\right) = (1,1)$; $f(1,2) = \left(\frac{1}{1}, \frac{1}{2}\right) = (1, 0.5)$; $f(2,1) = \left(\frac{1}{2}, \frac{1}{1}\right) = (0.5, 1)$; $f(2,2) = \left(\frac{1}{2}, \frac{1}{2}\right) = (0.5, 0.5)$.

When we replace each point of the square with its transformed counterpart, the new square has vertices at $(1, 1)$, $(1, 0.5)$, $(0.5, 1)$, and $(0.5, 0.5)$. By comparing this to the diagrams shown in the images, we can see that the square has shrunk and is now located within the first quadrant closer to the origin. **Option (C) depicts an unchanged square, which does not correspond to the transformed square we calculated.**

**Option (E) is the one that correctly represents a square with the calculated transformed vertices, which has shrunk and rotated towards the origin, remaining in the first quadrant.**

Figure 51: An example of analytic geometry question of GPT-4 with image caption. Although there are small hallucinations marked in red, The model correctly solves the problem based on the image description while GPT-4V fails because of inaccurate image recognition.

**Example of Counting**

**Question:** How many triangles can you find in the picture?

**Ground Truth:** 5

**GPT4V Image Caption:**

The image depicts a stylized, geometric drawing of a humanoid figure that appears to be a representation of a girl or woman, composed of various simple shapes. The figure is not a real person but rather a collection of shapes that could be used in a math problem. Here is a detailed description of the figure:

**Head:** The head is a white circle with a simple, smiling face. The face has two blue dots for eyes and a red, upside-down triangle for a nose. The mouth is a black, curved line like a smile.

**Hair:** The figure has orange hair represented by two shapes on either side of the head, resembling pigtails. Each pigtail is a half-circle, and there is a pink bow on top of the head, consisting of a small circle in the center and two triangles on either side to represent the bow's loops.

**Body:** The body is a green trapezoid, wider at the top and tapering slightly towards the bottom, suggesting a dress or a tunic. There are two yellow circles in the middle of the trapezoid, aligned vertically, representing buttons.

**Arms:** **The arms are two white rectangles extending from the upper sides of the trapezoid**, each ending in a white triangle to represent hands.

**Skirt:** Below the trapezoid body, there is a green, inverted trapezoid that is narrower at the top and wider at the bottom, representing a skirt or the lower part of the dress.

**Legs:** The legs are two white rectangles extending downward from the bottom of the skirt.

**Shoes:** It has orange shoes, **each represented by a small rectangle at the end of the legs**. The entire figure is outlined in black, giving it a cartoonish, clip-art style appearance. The use of geometric shapes to construct the figure suggests that it could be used in a math problem involving geometry, such as calculating areas, perimeters, or identifying shapes.

**GPT4 Answer:**

To solve the problem, let's identify and count all the triangles in the described figure. The description mentions specific shapes that make up the figure, and we'll focus on those identified as triangles or that can form triangles.

1. **Nose:** The nose is described as a red, upside-down triangle. Count: 1 triangle

2. **Hair Bows:** Each bow on top of the head consists of a small circle in the center and two triangles on either side to represent the bow's loops. Count: 2 triangles (since there's one bow mentioned, it has two triangles as part of its design)

3. **Hands:** Each hand at the end of the arms is represented by a white triangle. Count: 2 triangles (one for each hand)

4. **Implied Triangles:** The description does not explicitly mention other triangles, but we can check for any implied or composite triangles. Given the shapes described (circles for the head and buttons, rectangles for arms and legs, trapezoids for the body and skirt, half-circles for hair, and rectangles for shoes), there are no additional triangles formed by these shapes.

Therefore, the total count of triangles in the figure, based on the description provided, is:

$$1 \text{ (nose)} + 2 \text{ (hair bows))} + 2 \text{ (hands))} = 5$$

So, the answer is $\boxed{5}$ triangles.

Figure 52: An example of counting question of GPT-4 with image caption. Although there are small hallucinations marked in red, The model correctly solves the problem based on the image description.

**Example of Counting**

**Question:** How many more bricks does the right hand pyramid have than the left hand pyramid?

**Ground Truth:** 5

**GPT4V Image Caption:**

The diagram shows two separate pyramid-like structures composed of blocks. Each pyramid is arranged in a way that resembles steps, with each upper row containing fewer blocks than the row below it. The blocks are depicted in a 3D perspective, giving the appearance of depth.

The pyramid on the left has a total of 4 rows. From top to bottom, the number of blocks in each row increases by one. The top row has 1 block, the second row has 2 blocks, the third row has 3 blocks, and the bottom row has 4 blocks.

The pyramid on the right is larger, with a total of 5 rows. Similar to the left pyramid, the number of blocks in each row increases by one as you move from top to bottom. The top row has 1 block, the second row has 2 blocks, the third row has 3 blocks, the fourth row has 4 blocks, and the bottom row has 5 blocks.

Both pyramids are centered and have a symmetrical arrangement, ...

**GPT4 Answer:**

To solve this problem, we need to calculate the total number of blocks in each pyramid and then find the difference between the two totals.

For the left pyramid (4 rows): The number of blocks in each row forms an arithmetic sequence: 1, 2, 3, 4. The total number of blocks is the sum of this sequence: $1 + 2 + 3 + 4$.

For the right pyramid (5 rows): Similarly, the number of blocks in each row forms an arithmetic sequence: 1, 2, 3, 4, 5. The total number of blocks is the sum of this sequence: $1 + 2 + 3 + 4 + 5$.

Now, let's calculate the sums:

For the left pyramid: $1 + 2 + 3 + 4 = 10$; For the right pyramid: $1 + 2 + 3 + 4 + 5 = 15$.

Finally, to find out how many more blocks the right pyramid has than the left pyramid, we subtract the total number of blocks in the left pyramid from the total number of blocks in the right pyramid: $15 - 10 = 5$. So, the right pyramid has $\boxed{5}$ more blocks than the left pyramid.

**GPT-4V Answer:**

For the left hand pyramid: The top row has 1 brick, the second row has 2 bricks, the third row has 3 bricks, and the bottom row has 4 bricks. So, the total number of bricks in the left hand pyramid is $1 + 2 + 3 + 4 = 10$ bricks.

For the right hand pyramid: **The top row has 1 brick, the second row has 3 bricks, the third row has 5 bricks, and the bottom row has 7 bricks. So, the total number of bricks in the right hand pyramid is** $1 + 3 + 5 + 7 = 16$ **bricks.**

**Now we find the difference:** $16 - 10 = 6$**.**

**Therefore, the right pyramid has** $\boxed{6}$ **more bricks than the left pyramid.**

**Gemini Pro Answer:**

**The left hand pyramid has 6 rows of bricks. The right hand pyramid has 7 rows of bricks. There are 7 - 6 = 1 more rows of bricks in the right hand pyramid. Each row of the left hand pyramid has 6 bricks. So there are 6 * 6 = 36 bricks in the left hand pyramid. Each row of the right hand pyramid has 7 bricks. So there are 7 * 7 = 49 bricks in the right hand pyramid. There are 49 - 36 = 13 more bricks in the right hand pyramid.** $\boxed{13}$

Figure 53: An example of counting question of GPT-4 with image caption. The model correctly solves the problem based on the accurate and detailed image description while GPT-4V and Gemini Pro fail.

**Example of Combinatorics**

**Question:** The numbers $1, 2, 3, 4$ and $9$ are written into the squares on the following figure. The sum of the three numbers in the horizontal row, should be the same as the sum of the three numbers in the vertical column. Which number is written in the middle?

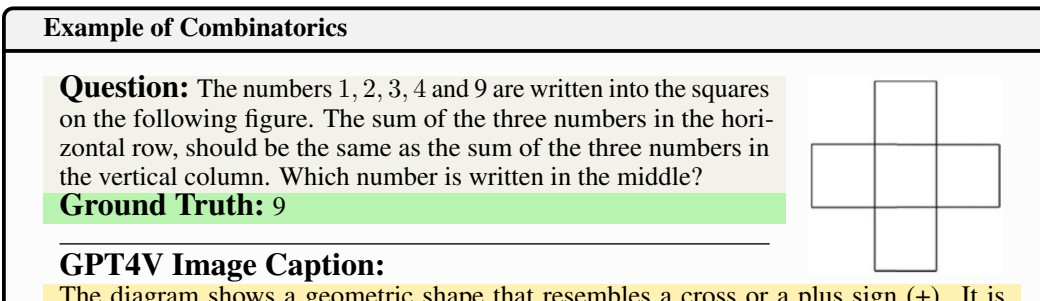

**Ground Truth:** 9

---

**GPT4V Image Caption:**
The diagram shows a geometric shape that resembles a cross or a plus sign (+). It is composed of five identical rectangles arranged such that one rectangle is at the center, and the other four are attached to each of its sides. The rectangles are aligned so that the longer sides of the rectangles are connected to the shorter sides of the central rectangle, forming a symmetrical cross shape.
Each rectangle appears to have the same dimensions, suggesting that they could be squares, but without specific measurements, we cannot confirm their exact shape. The lines that form the rectangles are uniform, indicating that the boundaries of each rectangle are clearly defined.
The overall shape could be interpreted as a 2D representation of an unfolded 3D box ...

**GPT4 Answer:**
Given the description of the figure and the problem, let's denote the numbers in the squares as follows, assuming the central square is $X$. And we know there's a number 9 involved, which hasn't been placed yet. Since the sum of the three numbers in the horizontal row should be the same as the sum of the three numbers in the vertical column, and we need to include all numbers $1, 2, 3, 4,$ and $9$, let's analyze the possible placements.
Given the numbers, the sum involving 9 would be significantly higher than any sum without it, meaning 9 must be in either the horizontal row or the vertical column that includes $X$. Since the figure is symmetrical and the problem specifies that the sums of the numbers in the horizontal row and vertical column must be equal, the only logical placement for 9 to balance the equation would be in the center, as $X$. Thus, the number written in the middle is $\boxed{9}$.

---

**Gemini Pro Answer:**
Let's call the numbers in the squares A, B, C, D, and E. We know that A + B + C = D + E. We also know that A + D = B + E. Adding these two equations together, we get 2A + B + C + D = 2B + 2E. Simplifying this equation, we get A + C = B + E. Since A + B + C = D + E, we can substitute A + C for D + E in this equation to get A + C = B + (A + C). Simplifying this equation, we get B = A. Therefore, the number in the middle is $\boxed{A}$.

---

**Qwen-VL-Max Answer:**
Let's call the number in the middle square $x$.
From the given information, we can set up the following equations:

$$1 + 2 + x = 9 + 4, \ 3 + 4 + x = 1 + 2 + 9$$

Solving for $x$ in each equation, we find that $x = \boxed{10}$.

---

Figure 54: An example of combinatorics question of GPT-4 with image caption. The model correctly solves the problem based on the accurate and detailed image description while Gemini Pro and Qwen-VL-Max fail.

### Example of Combinatorics

**Question:** A hen lays white and brown eggs. Lisa takes six of them and puts them in a box as shown. The brown eggs are not allowed to touch each other. What is the maximum number of brown eggs Lisa can place in the box?

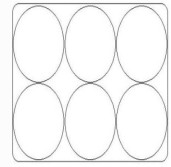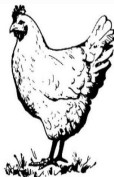

**Ground Truth:** 3

**GPT4V Image Caption:**

The diagram consists of two distinct parts:
1. On the left, there is a rectangle that contains six circles arranged in two rows of three. The circles are drawn such that each circle in the top row touches the two circles below it, and each circle in the bottom row touches the two circles above it. The circles are also touching their neighbors within the same row. The arrangement is such that the circles are packed closely together, maximizing the number of circles that can fit within the rectangle without overlapping.
2. On the right, there is an illustration of a chicken. The chicken is ...

**GPT4 Answer:**

1. **First Row**: Place a brown egg in the first position of the top row. To prevent any brown eggs from touching each other, the next egg in the top row must be white. Following this pattern, the third egg in the top row can be brown again. So, in the top row, we can place 2 brown eggs without them touching each other.
2. **Second Row**: For the bottom row, directly below a brown egg in the top row, we must place a white egg to prevent brown eggs from touching. This means that if the first and third positions in the top row are brown, the first and third positions in the bottom row must be white. Following the pattern to avoid brown eggs touching, we can place a brown egg in the second position of the bottom row. This ensures no brown eggs are touching vertically or horizontally.
So, in the bottom row, we can place 1 brown egg without it touching any other brown egg.
Adding the brown eggs from both rows, we get a total of $2 + 1 = 3$ brown eggs. $\boxed{3}$.

**GPT-4V Answer:**

The problem involves arranging a maximum number of brown eggs in a box with six compartments without allowing the brown eggs to touch each other. To maximize the number of brown eggs while adhering to this constraint, we can place a brown egg in every other compartment to ensure no two brown eggs are adjacent to one another.
Placing a brown egg in the first compartment, we then skip the second compartment and place another brown egg in the third compartment. **Repeating this pattern, we can see that we can place brown eggs in the first, third, and fifth compartments in the top row. For the bottom row, we can place brown eggs in the second, fourth, and sixth compartments.**
Hence, the maximum number of brown eggs Lisa can place in the box without any two brown eggs touching each other is **three on the top row and three on the bottom row, which is a total of: 3 (top row) + 3 (bottom row) = 6 brown eggs. Therefore, the answer is:** $\boxed{6}$

**Gemini Pro Answer:**

**Lisa can place at most 2 brown eggs in the box. Since the brown eggs are not allowed to touch each other, they can only be placed in the corner spots. There are only two corner spots, so the maximum number of brown eggs Lisa can place in the box is** $\boxed{2}$ **.**

Figure 55: An example of combinatorics question of GPT-4 with image caption. The model correctly solves the problem based on the accurate and detailed image description while GPT-4V and Gemini Pro fail.

## Example of Topology

**Question:** Theresa moves a pencil along the line. She starts at the arrow shown. In which order will she go past the shapes?
(A) $\triangle, \square, \bullet$
(B) $\triangle, \bullet, \square$
(C) $\bullet, \triangle, \square$
(D) $\square, \triangle, \bullet$
(E) $\square, \bullet, \triangle$

**Ground Truth:** A

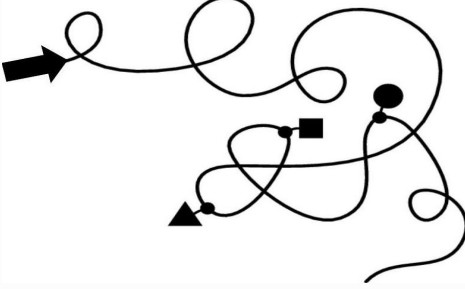

### GPT4V Image Caption:

The diagram shows a complex, continuous, and entangled line that forms a path with several loops and curves. Along this path, there are four distinct markers: a solid black arrow, a solid black square, a solid black circle, and a solid black triangle. Each marker is connected to the line, indicating specific points along the path.

Starting from the left, the path begins with the solid black arrow pointing to the right, indicating the starting point. The line then makes a small loop upwards and continues to the right before making a large loop counterclockwise. After completing this large loop, the line extends to the right and then makes a sharp turn downwards, forming a smaller loop in a clockwise direction.

**Following this smaller loop, the line extends upwards and then to the right, where it encounters the solid black square. The path continues to the right, makes a small loop upwards, and then heads down towards the solid black circle. After the circle, the line makes a large loop that goes up, then to the left, and finally down, passing to the right of the circle.**

**The path then makes a sharp turn upwards, creating a small loop, and continues to the left. It makes another loop downwards and to the right, passing below the previous loop and the solid black circle. The line then extends to the left, making a small upward loop before heading down and to the left, where it encounters the solid black triangle.**

The diagram does not provide any scale, units, or additional context, so the description is purely based on the visual elements present. The purpose of the markers and the path's complexity suggest that the diagram could be related to a problem involving sequencing, following directions, or identifying patterns.

### GPT4 Answer:

Based on the description provided, Theresa starts at the solid black arrow and follows the path. Here's the sequence of events as she moves along the path:

1. Start at the solid black arrow: This is the starting point. 2. Small loop upwards: This is just a part of the path and doesn't involve any of the shapes. 3. Large loop counterclockwise: Again, this part of the path doesn't involve any of the shapes. 4. Sharp turn downwards forming a smaller loop in a clockwise direction: Still no shapes encountered. 5. Extends upwards and then to the right: No shapes yet. **6. Encounters the solid black square: This is the first shape she encounters. 7. Makes a small loop upwards and then heads down towards the solid black circle: The circle is the second shape encountered.** 8. Makes a large loop that goes up, then to the left, and finally down, passing to the right of the circle: This loop doesn't lead to a new shape. 9. Sharp turn upwards creating a small loop and continues to the left: Still no new shapes. 10. Makes another loop downwards and to the right, passing below the previous loop and the solid black circle: This part of the path revisits the area near the circle but doesn't introduce a new shape. **11. Extends to the left, making a small upward loop before heading down and to the left, where it encounters the solid black triangle: The triangle is the third shape encountered.** 12. Makes a final loop upwards and to the right, ending with a curl: This final part of the path doesn't involve any new shapes.

Therefore, the order in which Theresa goes past the shapes is: 1. Solid black square 2. Solid black circle 3. Solid black triangle So, the final answer is: $\boxed{\text{Square, Circle, Triangle}}$

### GPT-4V Answer:

Following the line from the arrow, Theresa will first pass by the triangle ($\triangle$), then she will pass by the square ($\square$), and finally past the circle ($\bullet$). So, the correct order in which she will go is: $\triangle, \square, \bullet$

Therefore, the correct answer is: $\boxed{A}$

Figure 56: An example of topology question of GPT-4 with image caption. The model fails to solve the problem because of the inaccurate image description while GPT-4V succeeds.

## Example of Combinatorial Geometry

### Question:
Which shape cannot be seen in every picture?

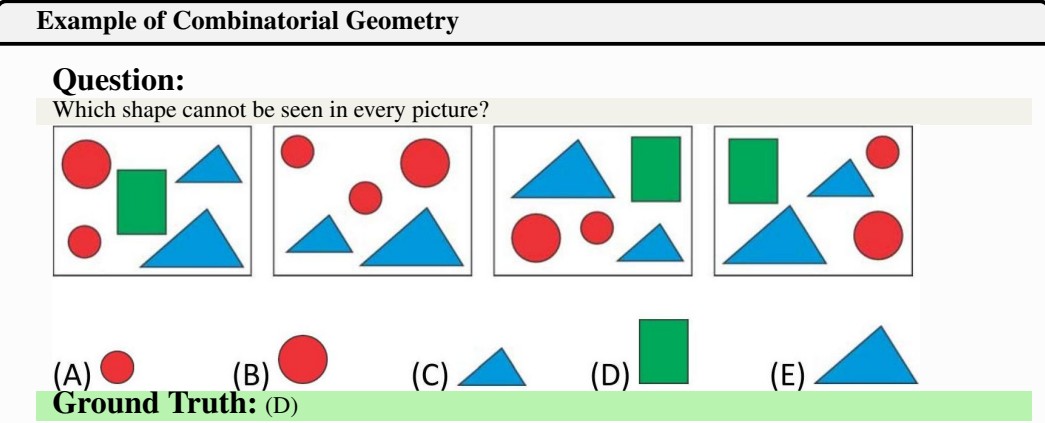

**Ground Truth:** (D)

### GPT4V Image Caption:
The diagram consists of a series of five images labeled from (A) to (E), each containing a combination of geometric shapes within a rectangular frame. Below each image, there is a single shape corresponding to each label.

Image (A) contains **three blue triangles** and two red circles arranged around a central green square. The triangles are pointing upwards, and the circles are placed near the corners of the rectangle, not touching the edges. The shape below image (A) is a single red circle.

Image (B) shows two blue triangles and **two red circles**. The triangles are pointing upwards, and the circles are placed near the top corners of the rectangle, not touching the edges. The shape below image (B) is a single red circle, larger than the one below image (A).

Image (C) contains **three blue triangles**, with two triangles pointing upwards and one pointing downwards. The triangles are arranged in a horizontal line in the center of the rectangle. The shape below image (C) is a single blue triangle pointing upwards.

Image (D) has **two green squares and one blue triangle**. The squares are placed on the left side of the rectangle, one above the other, and the triangle is pointing upwards, located on the right side of the rectangle. The shape below image (D) is a single green square.

Image (E) includes **one green square, one blue triangle pointing upwards, and one red circle**. The square is placed on the left side, the triangle in the center, and the circle on the right side of the rectangle. The shape below image (E) is a single blue triangle pointing upwards.

### GPT4 Answer:
To solve this problem, we need to identify which shape is not present in every image from (A) to (E).

**Image (A) contains: 3 blue triangles, 2 red circles, and 1 green square.**
**Image (B) contains: 2 blue triangles and 2 red circles.**
**Image (C) contains: 3 blue triangles.**
**Image (D) contains: 1 blue triangle and 2 green squares.**
**Image (E) contains: 1 blue triangle, 1 red circle, and 1 green square.**

Now, let's identify which shapes are present in each image:

Blue triangles are present in all images (A) to (E). Red circles are present in images (A), (B), and (E). Green squares are present in images (A), (D), and (E).

From this analysis, we can see that the shape that cannot be seen in every picture is the red circle, as it is missing from images (C) and (D). Therefore, the answer is: Red Circle .

### GPT-4V Answer:
The images display various combinations of shapes: circles, triangles, and squares in different arrangements. The question is asking which shape is not present in all the images.

After examining each picture, we can see: Picture 1 contains circles, triangles, and a square. Picture 2 contains circles and triangles. **Picture 3 contains circles and triangles. Picture 4 contains triangles and squares. Picture 5 contains circles, triangles, and a square.**

The square shape does not appear in all pictures; **it is absent in Picture 2 and Picture 3**.

So, the correct answer is: D

Figure 57: An example of combinatorial geometry question of GPT-4 with image caption. The model fails to solve the problem because of the inaccurate image description. GPT-4V also misinterprets the image, but gives the correct answer.

**Example of Transformation Geometry**

## Question:

Jim and Ben are sitting in a ferris wheel (see picture on the right). The ferris wheel is turning. Now Ben is in the position where Jim was beforehand. Where is Jim now?

<image1>:

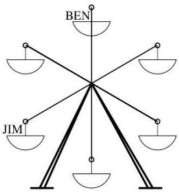

<image2>:

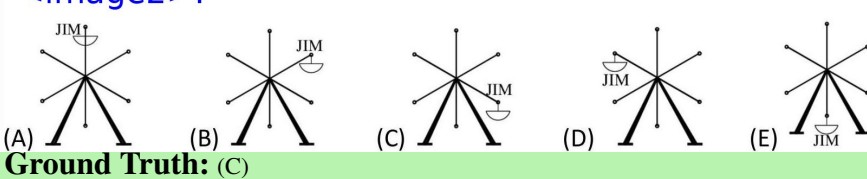

**Ground Truth:** (C)

## GPT4V Image Caption:

**Image 1**: This is a central diagram showing a complex balance scale system with a total of **seven** individual scales arranged in a tree-like structure. **At the top, there is a single scale labeled "BEN" with its fulcrum at the center and two pans hanging from either end. From each pan of this top scale, two additional scales are suspended, making a total of four scales on the second level. Each of these scales is also a balance with a fulcrum in the center and two pans hanging from either end. Each of these bottom scales is labeled "JIM" and has a fulcrum with two pans.**

**Image 2**: This image shows five different answer choices, labeled (A) through (E), each depicting a simplified version of the balance scale system from Image 1. Each choice shows a different configuration of the scales, with varying numbers of connections and arrangements.

**(A) shows a central scale labeled "JIM" with three scales hanging from its left pan and two scales hanging from its right pan. (B) shows a central scale labeled "JIM" with two scales hanging from each pan. (C) shows a central scale labeled "JIM" with three scales hanging from its left pan and three scales hanging from its right pan. (D) shows a central scale labeled "JIM" with four scales hanging from its left pan and one scale hanging from its right pan. (E) shows a central scale labeled "JIM" with one scale hanging from its left pan and four scales hanging from its right pan.**

## GPT4 Answer:

To solve this problem, we need to understand the relationship between Jim and Ben in the context of the balance scale system and then apply this understanding to the ferris wheel scenario.

From Image 1, we know that: The top scale is labeled "BEN". **There are seven individual scales arranged in a tree-like structure. Each "JIM" scale is at the bottom level of this structure.**

...Given the information and the nature of the question, the solution focuses on understanding the relative movement and positions of Jim and Ben in a ferris wheel, drawing an analogy from the balance scale system. Without a direct question linking the ferris wheel scenario to the choices (A) through (E), the final answer is conceptual: **Jim is now in the position where Ben was beforehand.**

## GPT-4V Answer:

To solve the problem we need to imagine the Ferris wheel rotating until Ben reaches Jim's initial position. In the original image, if we assume the Ferris wheel is rotating clockwise, Ben will have to move **three quarters** of a full turn to get to Jim's original position.

Given that the wheel has **eight spokes** and thus divides the circle into equal parts, moving three-quarters means Jim will move six parts out from his original position if Ben moves into Jim's original spot. So, Jim will be on the opposite side of the wheel, in the bottom position.

Looking at the provided answer choices in image2, $\boxed{C}$ **represents the position where Jim would be if the wheel rotates three quarters of a turn clockwise, with Ben taking Jim's original spot.**

Figure 58: An example of transformation geometry question of GPT-4 with image caption. The model fails to solve the problem because of the inaccurate image description. GPT-4V also misinterprets the image, but gives the correct answer.

**Example of Descriptive Geometry**

**Question:** Five equally big square pieces of card are placed on a table on top of each other. The picture on the side is created this way. The cards are collected up from top to bottom. In which order are they collected? (A) 5-4-3-2-1 (B) 5-2-3-4-1 (C) 5-4-2-3-1 (D) 5-3-2-1-4 (E) 5-2-3-1-4

**Ground Truth:** E

**GPT4V Image Caption:**

The diagram shows a composite rectangular figure made up of five smaller rectangles, each labeled with a number from 1 to 5. The overall shape is a larger rectangle. 1. Rectangle 1 is in the top left corner. **It is a vertical rectangle.** 2. Rectangle 2 is below Rectangle 1. It is a larger vertical rectangle, taking up the entire left side of the larger rectangle. 3. Rectangle 3 is in the center, bordering the right side of Rectangle 1 and the top side of Rectangle 2. It is a horizontal rectangle. . Rectangle 4 is a small square that is placed to the right of Rectangle 1 and above Rectangle 3. 5. Rectangle 5 is a large square that occupies the bottom right corner of the larger rectangle. It borders the right side of Rectangle 2 and the bottom side of Rectangle 3 and Rectangle 4. The rectangles are shaded in different shades of gray, possibly to distinguish them from one another. The numbers are placed in the center of each corresponding rectangle. The arrangement of the rectangles suggests that the dimensions of some could be inferred if the dimensions of others are known, which is a common setup for problems involving area and perimeter.

**GPT4 Answer:**

To solve this problem, we need to determine the order in which the cards (represented as rectangles in the description) are collected based on their arrangement in the composite figure. The key is to understand that the card on top in the figure will be the last one collected, and the card at the bottom will be the first one collected, as they are collected from top to bottom. 1. **Rectangle 5** is the large square at the bottom right corner. It does not lie underneath any other rectangle, meaning it is on top of all others in its area. Thus, it will be the last one collected. 2. **Rectangle 4** is the small square to the right of Rectangle 1 and above Rectangle 3. **Since it lies on top of Rectangle 3 and does not have any other rectangles on top of it, it will be collected before Rectangle 5 but after the rectangles it covers (Rectangles 1 and 3).**

......

Therefore, the order in which the cards are collected from top to bottom is: $\boxed{5, 4, 3, 1, 2}$.

**GPT-4V Answer:**

To solve the problem, we look at which cards are on top of others. To collect the cards from top to bottom as described, we should look for the one that is not covered by any other card, then the next, and so on. Starting from the top, **we can see that card 5 is at the bottom, as all other cards are on top of it in some part. Next, we see that card 2 is on top of card 5 but underneath cards 1 and 3. Then, card 3 is on top of both cards 2 and 5 but below cards 1 and 4. Card 4 is only underneath card 1. Finally, card 1 is at the top with no cards above it.**

......

Therefore, the correct sequence of collecting the cards from top to bottom is represented by option (E), when we reverse that sequence (since they are collected from top to bottom): 4-1-3-2-5. So the answer is:

$\boxed{E}$

Figure 59: An example of descriptive geometry question of GPT-4 with image caption. The models fail to correctly interpret the spacial relationships between the cards.

**Example of Logic**

**Question:** Maria wants to write whole numbers in the squares of the figure, so that the sum of the numbers in three consecutive squares is always 10. She has already written a number. What number should she write on the gray square?

| 2 | | | ▨ | | |

**Ground Truth:** 2

**GPT4V Image Caption:**

The diagram shows a horizontal rectangular bar divided into five equal-sized squares. The bar has a border around it, giving it the appearance of a single entity with internal divisions. The first square on the left is filled with the number 2 in a bold font. The second square is empty, the third square is shaded in a solid color, and the fourth and fifth squares are empty. The shaded square suggests it may have a particular significance in the context of the math problem, such as representing a specific value or condition that needs to be determined. The sequence and the values in the squares are likely part of the problem that needs to be solved.

**GPT4 Answer:**

To solve this problem, let's denote the numbers in the squares from left to right as $a$, $b$, $c$, $d$, and $e$. According to the description, $a = 2$, and $c$ is the number in the gray square that we need to find. We are given that the sum of the numbers in any three consecutive squares is always 10. Given this, **we can set up equations based on the sums of three consecutive squares: 1.** $a + b + c = 10$ **2.** $b + c + d = 10$ **3.** $c + d + e = 10$ We know that $a = 2$, so we can substitute this into the first equation to find $b + c$: 1. $2 + b + c = 10$ 2. $b + c = 8$ However, we need to find $c$, and we don't have direct values for $b$, $d$, or $e$. But we can notice that since the sum of any three consecutive numbers is 10, and $a = 2$, the only way to satisfy these conditions is to ensure that the sequence allows for this constant sum across any three consecutive squares. Given $2 + b + c = 10$, we have $b + c = 8$.

......

Given the constraints and the aim to solve for $c$, **the approach taken highlights the need for either a specific pattern to follow or additional given numbers in the sequence to directly calculate $c$'s value, pointing towards a limitation in the direct calculation method**

---

**GPT-4V Answer:**

Let's assign variables to represent the numbers in each of the squares:

| 2 | a | b | c | d |

Given that the sum of the numbers in three consecutive squares is always 10, we can create the following equations:

$$\begin{cases} 2 + a + b = 10 \\ a + b + c = 10 \\ b + c + d = 10 \end{cases}$$

We can solve these equations successively. From the first equation: $a + b = 10 - 2$ $a + b = 8$ Now, we don't have enough information to solve for a or b individually, but we don't need to because we can use the second equation to find c (which is related to the grey square): $a + b + c = 10$ Substitute $a + b = 8$ into the second equation: $8 + c = 10$ $c = 10 - 8$ $c = 2$ So, the number that should be written on the grey square is $\boxed{2}$.

Figure 60: An example of logic question of GPT-4 with image caption. GPT-4 fails to correctly find all the necessary equations and does not reach an answer. GPT-4V correctly solves the problems.

**Example of Metric Geometry**

**Question:** A circle $K$ is inscribed in a quarter circle with radius 6 as shown in the figure.

What is the radius of circle $K$?

(A) $\frac{6-\sqrt{2}}{2}$ (B) $\frac{3\sqrt{2}}{2}$ (C) 2.5 (D) 3 (E) $6(\sqrt{2}-1)$

**Ground Truth:** E

---

**GPT4V Image Caption:**

The diagram shows a large circle with a smaller circle inside it. The smaller circle is tangent to the larger circle at a single point on the larger circle's interior. The smaller circle is labeled with the letter K. The space between the two circles forms a crescent shape. The larger circle appears to be a boundary that encloses the smaller circle, and there is no indication of any dimensions, angles, or other geometric properties.

**GPT4 Answer:**

To solve this problem, let's denote the radius of the smaller circle (circle $K$) as $r$. Since circle $K$ is inscribed within a quarter circle of radius 6, we can visualize or draw a diagram where the quarter circle's radius forms a right-angled triangle with the sides being the radius of the quarter circle, the radius of circle $K$, and the distance from the point where circle $K$ touches the quarter circle's arc to the corner of the quarter circle (the center of the quarter circle). The distance from the center of circle $K$ to the corner of the quarter circle is $r$, and the distance from the corner of the quarter circle to the point where circle $K$ touches the quarter circle's arc is also $r$. Therefore, the hypotenuse of the right-angled triangle formed (which is also the radius of the quarter circle) is 6 units. Using the Pythagorean theorem, we can set up the equation: $r + r = 2r$ (this is the base and the height of the right-angled triangle, both equal to the diameter of circle $K$ since it touches both the arc and the two straight sides of the quarter circle). **The hypotenuse (the radius of the quarter circle) is** 6 **units, so:** $(2r)^2 + (2r)^2 = 6^2$

......

Therefore, the radius of circle $K$ is $\boxed{\dfrac{3\sqrt{2}}{2}}$ units.

---

**GPT-4V Answer:**

To solve this problem, we need to use geometric relationships within the quarter circle and the inscribed circle K. Consider a quarter circle of radius 6. If we inscribe a circle K inside it, such that K is tangent to the two sides and the arc of the quarter circle, there will be two points of tangency on the sides of the quarter circle which are at a distance equal to the radius of K from the corners of the quarter circle. Let's denote the radius of circle K as r. Since circle K is tangent to the two sides, we know that the distance from the center of K to each side is equal to r, the radius of K. This means there will be a square of side r formed between the center of circle K and the two sides of the quarter circle. The remaining section between the edge of this square and the quarter circle will be a right-angled isosceles triangle with sides of length $6 - r$ (since the quarter circle has a radius of 6). The hypotenuse of this right-angled isosceles triangle will be the radius of the quarter circle, which is 6. Using the Pythagorean theorem, we get: $(6-r)^2 + (6-r)^2 = 6^2$

......

so our answer is: $\boxed{E}$

---

Figure 61: An example of metric geometry question of GPT-4 with image caption. GPT-4 fails to find the correct relationship between the variables. GPT-4V correctly solves the problems.

