**▷ Function symmetry, etc.**

**Image:**

**Question1:** Is this an odd function?
(A) Yes (B) No
**Question2:** How many zeros does this function have?
**Question3:** What is the value of f(0)?
**Question4:** Is this function differentiable at each point?
**Question5:** f(2) is ____ zero.
**Question6:** The derivative of this function at x=0 is ____
**Question7:** What is this function most likely be?
(A) a polynomial
(B) a trigonometric function
(C) an exponential function
(D) a logarithmic function

**▷ Function monotonicity, etc.**

**Image:**

**Question1:** Which function is monotonic in range [0, pi]?
(A) the red one
(B) the blue one
(C) both
(D) none of them
**Question2:** At x=1, the value of the red curve is ____ that of the blue curve.
**Question3:** Which function is even?
**Question4:** Which function is odd?
**Question5:** Which function is periodic?
**Question6:** What is the value of the red curve at x=0?
**Question7:** What is the value of the blue curve at x=0?

**▷ Function roots, etc.**

**Image:**

**Question1:** How many zeros does this function have?
**Question2:** What is the zero of this function?
**Question3:** The derivative of y at x=6 is ____ that at x=8.
**Question4:** What is the value of y at x=2?
**Question5:** What is the value of y at x=1?
**Question6:** If this function invertible?
**Question7:** What is this function most likely be?
A a polynomial
B a trigonometric function
C an exponential function
D a logarithmic function