# OpenReview forum: "Measuring Multimodal Mathematical Reasoning with MATH-Vision Dataset"
_NeurIPS.cc/2024/Datasets_and_Benchmarks_Track — NeurIPS 2024 Track Datasets and Benchmarks Poster_

### Official Review · Reviewer_mPoH · 2024-06-15
**Towards Comprehensive Error Profiling in Multimodal Math Reasoning: Exploring Error Type Interactions and Automated Evaluation Frameworks**

**Rating:** 7
**Confidence:** 4
**Correctness:** N/A
**Clarity:** The paper is well written.

**Review:**

Please see strengths and limitations.

**Strengths:**

MATH-V stands out as a large-scale benchmark dataset, comprising 3,040 carefully curated mathematical problems sourced from real math competitions. The dataset covers an impressive range of 16 mathematical disciplines, including 8 newly introduced subjects such as analytic geometry, combinatorial geometry, and topology. This breadth of coverage ensures that the benchmark provides a comprehensive evaluation of LMMs' mathematical reasoning capabilities across various domains. Furthermore, the problems are graded into 5 levels of difficulty, enabling a nuanced assessment of models' performance at different complexity levels.

**Additional Feedback:**

N/A

**Documentation:**

Please see above in "Opportunities for Improvement".

**Limitations:**

1. Regarding dataset construction: MATH-V has many highlights in terms of data scale, quality control, and collection methods, setting a new benchmark for this field. However, the description of the difficulty level division of the dataset in the paper is not detailed enough, only mentioning in general that it refers to the grade requirements of the original competitions. To better guide subsequent research, future work could consider providing clearer and more operational criteria for difficulty division, such as setting explicit evaluation rules based on the mathematical concepts involved in the problems, reasoning steps, computational complexity, etc. This would not only help researchers better understand and apply this dataset but also provide useful references for the construction of other similar datasets.

2. Regarding error analysis: The authors conducted a thorough qualitative analysis of the error cases of GPT-4V, summarizing six major error types and their distribution patterns. This undoubtedly provides important insights and guidance for subsequent research. However, the paper's discussion of the potential associations or dependencies between different error types is not deep enough. Analyzing the interactions between errors can help us better understand the error patterns of the model and identify the sources and propagation paths of errors. For example, visual recognition errors may directly lead to problem understanding errors, which in turn trigger a series of reasoning errors. By analyzing the co-occurrence patterns, causal chains, conditional probabilities, etc. of these error types, we can more accurately locate the weak links of the model and carry out targeted optimization and improvement. Of course, such error interaction analysis also faces many methodological challenges, such as how to control confounding factors and how to distinguish correlation from causation, requiring researchers to be particularly cautious in experimental design and data analysis.

3. Regarding evaluation methods: The authors used manual annotation and multi-person voting to analyze the error cases of GPT-4V, which is to some extent constrained by scalability and subjectivity. As new models continue to emerge, this manual evaluation approach may struggle to keep up with the rapid iteration pace. To achieve larger-scale and more objective error analysis, future work could consider developing an automated multimodal mathematical reasoning evaluation framework. Specifically, the mathematical reasoning process could be deconstructed into several key subtasks, such as visual understanding, problem understanding, knowledge retrieval, logical reasoning, etc., and dedicated evaluation schemes and metric systems could be designed for each subtask. By systematically evaluating the model's performance on these subtasks, we can characterize its strengths and weaknesses in a more comprehensive and fine-grained manner. This fine-grained evaluation approach not only helps avoid the subjectivity of human judgment but also enables the automation and scalability of the evaluation process, significantly improving evaluation efficiency and the credibility of evaluation results. More importantly, this subtask-based evaluation system also lays an important foundation for realizing interpretable, modular multimodal mathematical reasoning models.

**Opportunities For Improvement:**

1. Dataset Construction: An opportunity for improvement lies in providing more explicit and operational criteria for dividing the dataset into difficulty levels. Instead of only referencing the grade requirements of the original competitions, future work could set clear evaluation rules based on factors like the mathematical concepts involved, reasoning steps required, and computational complexity. Establishing such criteria would enhance researchers' understanding and application of the dataset while also serving as a useful reference for constructing similar datasets.

2. Error Analysis: While the authors conducted a thorough qualitative analysis of error types, an opportunity for improvement is to delve deeper into the potential associations or dependencies between different error types. Analyzing the interactions, co-occurrence patterns, causal chains, and conditional probabilities of errors could lead to a more comprehensive understanding of the model's error patterns, error sources, and error propagation paths. For instance, visual recognition errors may directly cause problem understanding errors, subsequently triggering a cascade of reasoning errors.

3. Evaluation Methods: The manual annotation and multi-person voting approach used for error analysis has limitations in scalability and objectivity, presenting an opportunity for improvement. As new models emerge rapidly, an automated multimodal mathematical reasoning evaluation framework could facilitate larger-scale and more objective error analysis. This framework could deconstruct the reasoning process into subtasks (e.g., visual understanding, problem understanding, knowledge retrieval, logical reasoning) and design dedicated evaluation schemes and metric systems for each subtask. Systematically evaluating model performance on these subtasks would provide a comprehensive and fine-grained characterization of strengths and weaknesses. Moreover, this fine-grained, subtask-based evaluation system could lay the foundation for interpretable, modular multimodal mathematical reasoning models.

**Relation To Prior Work:**

Yes.

**Summary And Contributions:**

The paper "MATH-Vision (MATH-V): A Large-Scale Benchmark for Multimodal Mathematical Reasoning" introduces a new dataset consisting of 3,040 high-quality mathematical problems with visual contexts, sourced from real math competitions. The dataset spans 16 distinct mathematical disciplines and is graded across 5 levels of difficulty, providing a comprehensive and diverse set of challenges for evaluating the mathematical reasoning abilities of large multimodal models (LMMs). Through extensive experimentation, the authors uncover a performance gap between current LMMs and human performance on MATH-V, highlighting the need for further advancements in LMMs. Moreover, the detailed categorization of problems allows for a thorough error analysis of LMMs.

---

> ### Author Rebuttal · Authors · 2024-08-18
>
> **Q1:** Providing more explicit and operational criteria for dividing the dataset into difficulty levels
>
> **A1:** Thank you for your suggestion. First of all, the competitions from which our data comes are all targeted at different grades. This is a natural division of difficulty, because solving problems for higher grades requires more knowledge than solving problems for lower grades. The specific details are detailed in Appendix H.2. Secondly, we do plan to add multi-level labels of the mathematical concepts involved to each question in the future work, so that the questions in our dataset form an interconnected knowledge network.
>
> **Q2:** Delve deeper into the potential associations or dependencies between different error types.
>
> **A2:** Thank you for your insightful suggestion, this is indeed a good direction for future research.
>
> **Q3:** An automated multimodal mathematical reasoning evaluation framework could facilitate larger-scale and more objective error analysis. This framework could deconstruct the reasoning process into subtasks (e.g., visual understanding, problem understanding, knowledge retrieval, logical reasoning) and design dedicated evaluation schemes and metric systems for each subtask.
>
> **A3:** Yes, we totally agree with your suggestion. The reason why we use human annotation is to ensure the reliability of our error analysis. As the model develops, we can now use models such as GPT4-o or Llama-3-70B for error analysis because the general level of these models is high enough. In fact, we are working with the team of automated evaluation to achieve this.

---

### Official Review · Reviewer_QvcB · 2024-07-24

**Rating:** 8
**Confidence:** 4
**Correctness:** They seem correct to me.
**Clarity:** The paper is well written and easy to…

**Review:**

R1- [Major] By eyeballing through some examples in the dataset, I noticed that there are quite a number of questions where the image is totally unnecessary for solving the problem. Scanning the first 20 examples from the test set on huggingface, I found examples 10, 12, 14, 15, 17 and 18 to be of this kind. For example, question 15 is "Emily celebrated her birthday on Thursday, and her sister Liepa 8 days earlier. Which weekday was that?" and the image is simply an image of a birthday cake. The question can be solved without needing to look at the image at all. I'm not sure about the extent of this problem: my subsample shows 6/20 of the problems have this issue, but that could just be an unlucky subsample. One could argue that this might be a feature than a bug as it tests the ability of the models to reason in presence of distractor images. But in any case, the extent of such problems and their effect on the results should be made clear.

R2- [Minor] I found some of the comparisons to the previous work not super convincing. For example, it is highlighted several times in the paper that the average length of the questions in MATH-V is bigger than that of MATH-Vista. But question length alone doesn't really mean much, unless it is accompanied with more information on what the extra length is bringing.

Q1: What is the size of the category (from your 16 categories) with the minimum number of questions?

Q2: Line 84 claims the questions are "unseen" for LLMs. Is there any evidence for this, besides the filtering explained on line 158?

**Strengths:**

* Visual mathematical reasoning is an important area of research.
* The dataset comes from previous national and international competitions so the quality of the questions is expected to be high. I looked at a few of the problems and some of them indeed look quite nice.
* The dataset comes with difficulty levels making it amenable to various interesting future analysis, similar to the textual MATH dataset.
* Human performance is reported for the dataset and there seems to be quite a good amount of headroom for current models.
* Various models are evaluated on the benchmark.

**Additional Feedback:**

See review and areas for improvement.

**Documentation:**

There is a good amount of documentation.

**Ethics:**

No ethics concerns

**Limitations:**

No specific limitation besides those pointed out in the review.

**Opportunities For Improvement:**

For the problems with irrelevant images, I would suggest doing one of the following:
* Either remove them from the dataset,
* Or quantify the extent of such problems and try to reduce it so it only corresponds to a small portion of the entire the dataset (maybe 2-3%),
* Or create separate test sets, one containing only problems with relevant images and one containing only problems with irrelevant images. Then, future users of the dataset can decide which set they would like to evaluate on.

**Relation To Prior Work:**

[Minor] Some of the "major update" claims starting in line 60 are also not quite novel. For example, visual graph theory problems have been already benchmarked in Das et al. "Which modality should I use - Text, Motif, or Image?" (although the graph theory problems in MATH-V seem to differ) and dividing plane geometry problems into angle, area and side has been already done in GeomVerse. Would be nice to properly credit these related works.

**Summary And Contributions:**

This work creates a multi-modal mathematical reasoning benchmark with 3040 questions. The problems come from national and international competitions, are graded with 5 difficulty levels, and are grouped into 16 categories. Several open and closed LLMs are evaluated on this dataset and the results are compared against human performance showing a wide gap.

---

> ### Author Rebuttal · Authors · 2024-08-18
>
> **Q1: [Major]** There are quite a number of questions where the image is totally unnecessary for solving the problem.
>
> **A1:** Thank you for your detailed observation and valuable feedback. We are also concerned about the issue. Firstly, we make sure that all images in our dataset are relevant to their corresponding questions. Upon reviewing all 3,040 questions, we found that only 2.89% of the questions did not need to have images in the questions. Among these, 55.7% would require the additional construction of a visual representation to solve the problem if the image were not provided, such as in plane geometry problems. Secondly, the reason why there are so many pictures of questions in your sample seems unnecessary is that the first 20 questions are all level 1 and level 2 questions. These questions correspond to the lower grades of elementary school. For these children, these pictures can be helpful for understanding the questions. For example, in question 10, without a diagram, there might be ambiguity as to whether two people can sit on the same side of the table at the same time. Thirdly, we show the results of GPT-4 without image (only 8.16%), which is actually close to the score of random answers. This also shows that the questions that can be answered without pictures only account for a small part of our dataset.
>
> **Q2: [Minor]** Question length alone doesn't really mean much, unless it is accompanied with more information on what the extra length is bringing.
>
> **A2:** Thank you for your feedback, and we apologize for any confusion regarding the significance of question length in our comparisons. We agree that, in isolation, question length might not directly indicate the quality or difficulty of the questions. Our intention in highlighting the difference in question length was to emphasize the advantages of our holistic data collection approach. As mentioned in line 65 of our paper, "all data are jointly collected from real-world math competitions, which eliminates the need for additional annotation and avoids monotonous question patterns." This approach naturally leads to a **wider variety of question types and lengths**, reflecting the diversity and complexity inherent in real-world math problems.In Appendix G.1, we further analyzed that a significant number of questions in MathVista were generated from images and often followed repetitive templates. For example, questions containing the phrase "What is the age gap between" appear 310 times. This templating results in shorter, more straightforward questions that are easier to generate but may lack the depth and variety of real-world competition problems. To better illustrate this point, we provided qualitative comparisons in Figures 9 through 13. These figures show that **shorter questions in MathVista tend to be template-driven, whereas the longer questions in MATH-V involve richer context** and require more intricate visual recognition and mathematical reasoning.We hope this clarification addresses your concern and demonstrates that the longer question length in our dataset is a reflection of its greater complexity and diversity.
>
> **Q3:** What is the size of the category (from your 16 categories) with the minimum number of questions?
>
> **A3:** The smallest category is Topology, which contains 23 questions.
>
> **Q4:** Line 84 claims the questions are "unseen" for LLMs. Is there any evidence for this, besides the filtering explained on line 158?
>
> **A4:** Thank you for your question. Firstly, the three main closed-source models, GPT-4V, Gemini 1.5 and Claude 3, all evaluate their models using several contests in the technical report, including AMC, which is a primary source of our problems. We therefore assumed that GPT-4V, Gemini 1.5, Claude 3 follow the common practice of not utilizing leaked data for training. However, since they are closed-source models, this is our best guess. Secondly, among open-source models, the most widely used base model is LLaVA, which utilizes a subset of the CC-3M dataset for pretraining and LAION/CC/SBU BLIP captions for instruction tuning. We show the overlap between MATH-Vision and three representative LMM training datasets. We find that the overlap is less that 1% when n=6 and almost negligible when n=8 (used in Llama3) and n=13 (used in GPT-3).
>
>
>
> |                          | 4-grams | 5-grams | 6-grams | 7-grams | 8-grams | 13-grams |
> | :----------------------- | :-----: | :-----: | :-----: | :-----: | :-----: | :------: |
> | sharegpt4v_mix665k       | 12.89%  |  3.68%  |  0.98%  |  0.24%  |  0.06%  |    0     |
> | blip_laion_cc_sbu_558k   |  2.74%  |  0.54%  |  0.09%  |  0.02%  |    0    |    0     |
> | LLaVA-CC3M-Pretrain-595K |  3.45%  |  0.66%  |  0.10%  |  0.01%  |    0    |    0     |
>
>
>
> **Q5: [Minor]** Some of the "major update" claims starting in line 60 are also not quite novel. For example, ... . Would be nice to properly credit these related works.
>
> **A5:** Sorry for the oversight. Our statement here is mainly based on the comparison with MathVista and MMMU-Math. We will revise our statement and cite these papers appropriately. Thank you for pointing out our oversight.

---

> > ### Comment · Reviewer_QvcB · 2024-08-26
> > **Reviewer Response**
> >
> > Thanks for the response and for verifying the extent of the "no image needed" problem I raised. My concern is now resolved and I have increased my score.

---

> > > ### Author Response · Authors · 2024-08-28
> > > **Thanks for your feedback!**
> > >
> > > We are glad to have received your message! Thank you for recognizing our paper. Your advice and comments have truly contributed to the improvement of our work!

---

### Official Review · Reviewer_ZxmA · 2024-07-25

**Rating:** 5
**Confidence:** 4

**Review:**

The authors proposed a more diverse and challenging multimodal math benchmark, but there are several issues in correctness and clarity, as well as language errors.

**Strengths:**

The proposed dataset is more diverse and challenging compared to previous multimodal math benchmarks.

**Additional Feedback:**

Typo: “GPT-4turbo”, “​​on several prominent, to determine”, “Failure CoT prompting”, “SPHNIX”
What is the performance without images?

**Clarity:**

It is not clear whether the annotators are qualified to check the dataset samples, as they are from science background but the questions are competition-level math.
It is not guaranteed whether the questions really need visual inputs to be answered. The authors should conduct a more thorough analysis on this aspect of the data.

**Correctness:**

There is a contradiction where the authors state they use zero-shot setting, but include chain-of-thought prompting, which is few-shot based on the cited paper (Wei et al).
While the authors try to avoid data leakage by selection of sources, they should conduct a more rigorous data contamination analysis (such as n-gram testing which is common)

**Documentation:**

Yes

**Ethics:**

Yes, the licensing and copyright permissions of the data source exams need to be double-checked.

**Limitations:**

The authors should discuss the limitation of whether the annotators are qualified to check the highly challenging competition math questions.

**Opportunities For Improvement:**

There are several areas for improvement regarding correctness and clarity, and the main motivation of more diversity and breadth in math questions is not very strong. There are also numerous typos.

**Relation To Prior Work:**

Yes

**Summary And Contributions:**

To address the limitations in diversity of previous multimodal math benchmarks, the authors propose a high-quality dataset from math competitions. The experiments reveal significant gaps between model and human performance, and provide insights to improve future models.

---

> ### Author Rebuttal · Authors · 2024-08-18
>
> **Q1: The main motivation of more diversity and breadth in math questions is not very strong.**
>
> A1: Thank you for your comment. The aim of proposing this benchmark is to complement existing multimodal reasoning datasets by incorporating additional mathematical categories and presenting high-quality problems embedded within visual contexts. Our objective is to more accurately highlight the disparities between current LMMs and human performance that are **overlooked by current benchmarks**. There are many problems that are easy for humans but difficult for LMMs which are not covered in current benchmarks but are important for evaluating the ability to solve problems related to engineering, architecture, and design, etc. Consequently, we believe our dataset fills a crucial gap, providing insights into the areas where LMMs still lag behind humans.
>
>
>
>
> **Q2: There is a contradiction where the authors state they use zero-shot setting, but include chain-of-thought prompting, which is few-shot based on the cited paper (Wei et al).**
>
> A2: Sorry for the confusion. When CoT prompting was first proposed, it was indeed a few-shot method, but as the model capabilities increase, using zero-shot prompts only (like “Let’s think step by step”) can also disclose the model's step-by-step reasoning ability. We will cite the following work using zero-shot CoT to dispel this ambiguity.
>
> *Large Language Models are Zero-Shot Reasoners: https://arxiv.org/abs/2205.11916.*
>
> **Q3: While the authors try to avoid data leakage by selection of sources, they should conduct a more rigorous data contamination analysis (such as n-gram testing which is common)**
>
> A3: Thank you for your suggestion. We show the overlap between MATH-Vision and three representative LMM training datasets. We find that the overlap is less that 1% when n=6 and almost negligible when n=8 (used in Llama3) and n=13 (used in GPT-3). We will put the results in the revised version of Appendix D.
>
> |                          | 4-grams | 5-grams | 6-grams | 7-grams | 8-grams | 13-grams |
> | :----------------------- | :-----: | :-----: | :-----: | :-----: | :-----: | :------: |
> | sharegpt4v_mix665k       | 12.89%  |  3.68%  |  0.98%  |  0.24%  |  0.06%  |    0     |
> | blip_laion_cc_sbu_558k   |  2.74%  |  0.54%  |  0.09%  |  0.02%  |    0    |    0     |
> | LLaVA-CC3M-Pretrain-595K |  3.45%  |  0.66%  |  0.10%  |  0.01%  |    0    |    0     |
>
>
>
> **Q4: Whether the annotators are qualified to check the dataset samples, as they are from science background but the questions are competition-level math.**
>
> A4: Thank you for the question. Our annotators are college students with science backgrounds with proven track records (either with high GPAs in Math or as finalists in mathematical competitions) that possess a strong proficiency in comprehending and resolving high school-level and lower math problems, therefore they are fully qualified for the described tasks since our dataset only contains questions at the **high school level and below** (grade 1 to 12), and we did not select overly difficult competitions. In Line 153, “Competitions like the International Mathematical Olympiad (IMO) are excluded due to the excessive difficulty …”.
>
>
>
> **Q5: Whether the questions really need visual inputs to be answered. What is the performance without images?**
>
> A5: Thank you for your question. Firstly, we make sure that all images in our dataset are relevant to their corresponding questions. Upon reviewing all 3,040 questions, we found that only 2.89% of the questions did not need to have images in the questions. Among these, 55.7% would require the additional construction of a visual representation to solve the problem if the image were not provided, such as in plane geometry problems. Secondly, we show the results of GPT-4 without image (only 8.16%), which is actually close to the score of random answers. This also shows that the questions that can be answered without pictures only account for a small part of our dataset.
>
>
>
> **Q6: The licensing and copyright permissions of the data source exams need to be double-checked.**
>
> A6: Thank you for the valuable suggestion. We have checked and further discussed the licensing and copyright permissions of the data sources (KMA, UKMT and MAA) in the ethic reviews. Please refer to the relevant sections for more details.
>
> **Q7: Typos.**
>
> A7: We sincerely apologize for the language errors noted in the review, and greatly appreciate the reviewer’s attention to these details. We will carefully revise our paper to correct these errors to improve the overall quality of our submission.

---

> > ### Author Response · Authors · 2024-08-24
> >
> > Dear Reviewer,
> >
> > We sincerely hope to address any concerns you may have and appreciate the opportunity to engage with you. As the discussion period is coming to an end, we kindly expect your feedback on our response.
> >
> > Thank you for your time and suggestions.
> >
> > Best regards,
> > Authors

---

> ### Author Response · Authors · 2024-08-28
> **Looking forward to your reply!**
>
> Dear Reviewer,
>
> Thanks again for your great efforts in reviewing this paper! With the discussion period drawing to a close, we expect your feedback and thoughts on our reply. We sincerely hope you can consider our reply in your assessment. We look forward to hearing from you, and we can further address unclear explanations and remaining concerns if any.
>
> Best,
>
> Authors

---

> ### Author Response · Authors · 2024-08-31
> **Sincere Request for Review of Our Responses**
>
> Dear Reviewer ZxmA,
>
> We greatly appreciate the time and effort you have invested in reviewing our work. Your initial comments are very insightful and helpful in enhancing the quality of our paper.
>
> In response to your suggestions, we have made the following revisions:
> 1. Conducted n-gram testing to improve data contamination analysis.
> 2. Checked "no image needed" problem.
> 3. Secured and documented permissions from our data sources.
> 4. Expanded our explanations to clarify some points.
>
> **We sincerely hope that these revisions have adequately addressed your feedback and will encourage you to reconsider the rating.** We are eager to hear any further feedback you might have and are available to discuss any points in more detail before the discussion deadline. Your expertise is highly valued and has been crucial in refining our work.
>
>
> Best regards,
>
> The Authors

---

### Official Review · Reviewer_YHoM · 2024-07-25
**A Visual math benchmark for evaluating mathematical reasoning capabilities of models**

**Rating:** 8
**Confidence:** 4
**Clarity:** The paper is well-written and easy to…

**Review:**

Benchmarking the mathematical reasoning of top AI models is an important area, and this paper provides a solid improvement compared to existing benchmarks.

pros:
- The benchmark consists of real math problems from competitions and exams.
- The paper is well-written and provides thorough details of data collection and evaluation. The process is careful and well-designed.
- The evaluations showcase the limitations in current models in the visual math reasoning tasks.

cons:
- Given the relatively small size of the samples, the MATH-Vision is a benchmark for evaluation rather than a dataset to train/improve models.

**Strengths:**

The dataset is collected in a careful way. Specifically, considering the possibility of data contamination and taking measures against it is notable. Moreover, the paper has clear merit compared to existing benchmarks, having real problems, and a more neate selection.

**Additional Feedback:**

Do the authors mean Kangaroo of Mathematics in "Austria", or "Australia"? If this is not a mistake and they used problems from Kangaroo of Mathematics in Austria, did they translate the problems from German to English?

**Correctness:**

The methods for creating the dataset and evaluating models seem sound. The authors took care to ensure high-quality problems and annotations. The evaluation process and error analysis appear to be done correctly.

**Documentation:**

The paper provides good details on how the dataset was created and organized. They mention plans to release the dataset publicly. A website hosted for the benchmark, including the details and use manuals, alongside more specifics on long-term maintenance plans, would be helpful.

**Ethics:**

There don't seem to be major ethical concerns with this paper.

**Limitations:**

The authors discuss limitations. Math problem datasets generally have a low risk of negative societal impact.

**Opportunities For Improvement:**

- Provide model performances in each difficulty level.
- Expand the sample size.

**Relation To Prior Work:**

The authors provide a very good comparison against existing benchmarks, specifically Mathvista. The provided details and discussions show clear improvements.

**Summary And Contributions:**

This paper introduces a new benchmark called MATH-Vision (MATH-V) for testing how well AI models can do math reasoning with both images and text. The dataset has more than 3,000 math problems from real math competitions, covering different math topics and difficulty levels. The authors' evaluation of several AI models on this benchmark shows the limitations of current models in visual math reasoning.

---

> ### Author Rebuttal · Authors · 2024-08-18
>
> **Q1: Given the relatively small size of the samples, the MATH-Vision is a benchmark for evaluation rather than a dataset to train/improve models.**
>
> A1: Yes, we built MATH-V to better evaluate the mathematical reasoning ability of multimodal models. We plan to explore how to improve the mathematical ability of LMMs in future work.
>
> **Q2: Provide model performances in each difficulty level.**
>
> A2: Thank you for the suggestion. We put the model performances in each difficulty level at Appendix H.2 due to page limit. Here are the results of GPT-4 and Gemini.
>
>
> |                | Level 1 | Level 2 | Level 3 | Level 4 | Level 5 |
> | :------------- | :-----: | :-----: | :-----: | :-----: | :-----: |
> | GPT4turbo      |   38    |  35.1   |  31.8   |  29.4   |  22.2   |
> | GPT4o          |  37.6   |  35.5   |  31.6   |  28.4   |  23.8   |
> | Gemini-1.5 Pro |  33.1   |  32.1   |  22.1   |  17.4   |  15.5   |
> | Gemini Pro     |  19.4   |  19.2   |  18.9   |  16.3   |  16.1   |
>
> **Q3: Expand the sample size.**
>
> A3: Thank you for your suggestion! We will maintain our dataset for a long time. Most of the competitions that our datasets are sourced from are held annually, and we will regularly add these new questions, which will not only expand the dataset, but also improve the reliability and fairness of our evaluation results.
>
> **Q4: Do the authors mean Kangaroo of Mathematics in "Austria", or "Australia"? If this is not a mistake and they used problems from Kangaroo of Mathematics in Austria, did they translate the problems from German to English?**
>
> A4: This is not a mistake and we do not translate these questions, Kangaroo of Mathematics in Austria provides both German and English versions of the questions.

---

### Comment · Area_Chair_CnkB · 2024-08-30
**Urgent: Reminder to Review Author Rebuttals and Engage in Discussion**

Dear Reviewers,

We are just two days away from the end of the discussion period. The authors have provided their rebuttals to the reviews. It is very important that you review the rebuttal and other reviews, and engage in the conversation or respond if the rebuttal addresses your concerns.

Thank you.

---

### Author Response · Authors · 2024-09-01
**Summary of the Author-Reviewer Discussion Period**

Dear Chairs,

We sincerely appreciate the efforts of all reviewers in reviewing our paper and providing insightful suggestions for its improvement. Despite our efforts in addressing and submitting responses to all reviewers' comments two weeks ago, only **reviewer QvcB** has responded thus far.

We have carefully addressed all the concerns and suggestions of the reviewers, and **R3-QvcB** raised the score after reviewing our rebuttal.

We are delighted that all reviewers have emphasized the significant contributions of our work.
**R1-YHoM** and **R3-QvcB** note that our paper studies an important area, and **all reviewers** agree that our dataset provides insights to improve future models **(including R2-ZxmA)**. The comprehensive evaluations conducted under various settings with various models **(R3-QvcB)** and the contribution of disclosing a wide gap between current models and human performance **(all reviewers)** have also been appreciated.

Regarding **R2-ZxmA**'s concerns, we want to stress the following:

- Our dataset reveals significant gaps between current LMMs and human performance that are relevant but not known to the community, making it a valuable addition to existing benchmarks.
- Results of n-gram testing show that our dataset doesn't suffer from data leakage issues.
- Our dataset only contains questions at the high school level and below and our annotators are qualified to check the dataset samples.
- The analysis of the "no image needed" problem once again demonstrates the high quality of our dataset. **R3-QvcB has the same concern and raised the score after our rebuttal.** https://openreview.net/forum?id=QWTCcxMpPA&noteId=GKRiyutStu

Regarding ethics reviews:
- Association Kangourou sans Frontières (AKSF) which owns that copyright for all problems we collected from KMA and UKMT **granted us all necessary rights** to utilize their data for our benchmark.

- For problems collected from MMA, we did not receive any negative feedback from board members. Our licensing terms are based on those of the MATH dataset (https://arxiv.org/pdf/2103.03874, NeurIPS DB 2021), which was constructed using problems from the same origin. MAA possesses the ability to address inappropriate use of intellectual property generated through Association activities.

These updates will be consistently incorporated into future versions of the paper. Thank you for your consideration and timely attention.

Best regards,

The Authors

---

### Decision · Program_Chairs · 2024-09-26

**Decision:**

Accept (Poster)

**Comment:**

The following recommendation does not include an ethics review and is contingent upon the outcome of the ethics review.

This paper addresses the lack of diverse benchmarks for mathematical visual reasoning problems. It introduces a new benchmark, MATH-V, which includes 3,040 mathematical problems drawn from real math competitions across 16 disciplines and 5 difficulty levels. Furthermore, the paper conducts extensive experiments on this benchmark and demonstrates a significant gap between human performance and that of state-of-the-art LMMs.

Three reviewers recommended accepting the paper, while one leaned towards rejecting it. The reviewers: (i) found the paper to make a significant contribution by introducing an important new benchmark for mathematical visual reasoning, (ii) considered the dataset diverse and comprehensive, containing real math competition questions from various disciplines and difficulty levels, and (iii) appreciated the experiments on multiple recent models, which highlighted important findings on the limitations of state-of-the-art methods. During rebuttal, the authors provided analyses on the fraction of questions requiring visual context, the overlap of the new benchmark with existing ones, and the models' performance with respect to difficulty levels, addressing the raised concerns.

AC agrees that the community would significantly benefit from the introduced benchmark and presented findings, and that it would spark further research in the area. Therefore, AC recommends accepting the paper.